# Dietary restriction transforms the mammalian protein persulfidome in a tissue-specific and cystathionine γ-lyase-dependent manner

Nazmin Bithi[1], Christopher Link[1], Yoko O. Henderson[1], Suzie Kim[1], Jie Yang[1], Ling Li[2], Rui Wang[3], Belinda Willard[2] & Christopher Hine [1✉]

Hydrogen sulfide ($H_2S$) is a cytoprotective redox-active metabolite that signals through protein persulfidation (R-SS$_n$H). Despite the known importance of persulfidation, tissue-specific persulfidome profiles and their associated functions are not well characterized, specifically under conditions and interventions known to modulate $H_2S$ production. We hypothesize that dietary restriction (DR), which increases lifespan and can boost $H_2S$ production, expands tissue-specific persulfidomes. Here, we find protein persulfidation enriched in liver, kidney, muscle, and brain but decreased in heart of young and aged male mice under two forms of DR, with DR promoting persulfidation in numerous metabolic and aging-related pathways. Mice lacking cystathionine γ-lyase (CGL) have overall decreased tissue protein persulfidation numbers and fail to functionally augment persulfidomes in response to DR, predominantly in kidney, muscle, and brain. Here, we define tissue- and CGL-dependent persulfidomes and how diet transforms their makeup, underscoring the breadth for DR and $H_2S$ to impact biological processes and organismal health.

[1] Department of Cardiovascular and Metabolic Sciences, Cleveland Clinic Lerner Research Institute, Cleveland, OH, USA. [2] Proteomics and Metabolomics Core Laboratory, Cleveland Clinic Lerner Research Institute, Cleveland, OH, USA. [3] Department of Biology, Faculty of Science, York University, Toronto, ON, Canada. ✉email: hinec@ccf.org

Established interventions and models to extend both lifespan and healthspan across evolutionary boundaries include dietary restriction (DR)[1] and disruption of the growth hormone (GH)/insulin-like growth factor 1 (IGF-1) axis[2]. Largely defined as reduced nutrient intake without malnutrition, DR encompasses decreased total daily caloric intake, the removal of specific macronutrients such as amino acids, and/or intermittent fasting. In addition to defending against aging-related diseases[3–5], DR also provides stress resistance[6] and metabolic fitness[7,8]. Similar to DR, mice lacking adequate GH production[9] or GH receptor (GHRKO) signaling[10] have increased lifespans and metabolic fitness[11]. It is hypothesized that common molecular pathways enriched or affected by DR and GH/IGF-1 axis disruption are central to improved aging[12]. Identifying these shared pathways and elucidating their mechanism of action will shed light onto the underpinnings of aging and usher targeted diagnostics and therapies into the clinic.

A common molecular phenomenon amongst several dietary and endocrine models of longevity is the altered metabolism of sulfur-containing amino acids methionine and cysteine, which entails flux through transsulfuration[13–15] and increased production capacity and/or bioavailability of hydrogen sulfide ($H_2S$) gas[16]. $H_2S$ and its $HS^-$ anion and $S^{2-}$ ion, herein referred simply as $H_2S$, is historically classified as an environmental and occupational hazard[17–19]. However, in the later part of the 20th and early part of the 21st centuries, functional and beneficial production of $H_2S$ via enzymatic catalysis of sulfur-containing amino acids was discovered in brain, aorta, and vascular smooth muscle cells[20–22]. Exogenously provided $H_2S$ increases lifespan in model organisms[23], prevents ischemia-reperfusion injury[24,25], improves cardiovascular health[26,27], and is recognized as a potential therapeutic against aging-associated diseases[28]. Conversely, deficiencies in endogenous $H_2S$ levels or production capacity correlate with and/or are causative of hypertension[27] and neurodegeneration[29]. Thus, enhancing $H_2S$ production serves as a therapeutic target for numerous clinically relevant endpoints.

Due to the multiphasic dose response of $H_2S$, life evolved with mechanisms for its controlled production and utilization. In mammals, $H_2S$ is enzymatically produced during the catabolism of sulfur-containing amino acids, primarily cysteine and homocysteine, via the activities of cystathionine β-synthase (CBS), cystathionine γ-lyase (CGL), and 3-mercaptopyruvate sulfurtransferase (3-MST)[30]. CGL, CBS, and 3-MST are differentially expressed and active in tissue-specific manners, with CGL contributing the majority of enzymatic $H_2S$ production in the kidney, liver, and endothelium[31–33]. Lack of CGL expression and/or activity is attributed to hypertension[27], neurodegeneration[34], and the inability to properly respond to dietary and endocrine cues[25,33,35–37]. Remarkably, increased CGL expression in the liver is a hallmark of numerous dietary, genetic, hormonal, and pharmaceutical models of an extended lifespan and may serve as a molecular biomarker associated with longevity[38].

Downstream mechanisms to how CGL and subsequent $H_2S$ production impart cellular and systemic benefits rest on the redox versatility of sulfur[39]. $H_2S$ performs several non-mutually exclusive biochemical reactions, including mitochondrial electron transfer, alterations of iron–sulfur and heme centers, antioxidant activity, and protein posttranslational modification via persulfidation—aka sulfhydration—of reactive cysteine residues to form persulfide tails ($R-SS_nH$)[40]. While the benefits of a diffusible antioxidant to counter oxidative damage along with improvement in bioenergetics to delay the onset of aging-related disorders are easily identifiable, the act of persulfidation and its extent to improve fitness and lifespan is less understood.

Like other posttranslational modifications, persulfidation potentially alters a protein's structure, function, stability, and/or macromolecular interactions[39,41,42]. The persulfidation process generally involves: (1) oxidation such as sulfenylation of the cysteine residue(s) on the targeted protein followed by nucleophilic attack of the oxidized thiol and/or disulfide by $H_2S$, or (2) reaction between $H_2S$ derived polysulfides ($H_2S_n$) with a reduced or glutathione-attached protein thiol[41,43–45]. It is estimated that 10 to 25% of proteins in the rodent liver are persulfidated[42] and this modification typically increases the reactivity of the modified cysteine[46]. Studies examining biochemical functions of persulfidation are primarily constructed on cell or purified protein models in vitro. Few studies have examined the total number and identity of persulfidated proteins, or persulfidome—aka sulfhydrome—of an organism in vivo. One such study was completed in the plant Arabidopsis thaliana, with the discovery of 2015 proteins, or ~5% of the entire proteome, as persulfidated[47]. Recently, it was elegantly shown using a dimedone switch method that aging attenuates and long-term DR enhances overall liver persulfidation abundance in mice[44]. However, a detailed analysis of mammalian multi-organ persulfidomes, the individual proteins and their cysteine sites of modification contained within, and the enrichment of specific biological pathways with these persulfidated proteins have not been described, particularly under anti-aging and CGL-perturbed conditions known to impact endogenous $H_2S$ production.

Taking advantage of the relative specificity for maleimide to alkylate protein thiols and persulfides[48–50], particularly in whole tissue preparations[51], we utilized biotin-conjugated maleimide in a modified Biotin-Thiol-Assay (BTA) developed by Gao, et al.[43,45], to isolate and identify persulfidated proteins and peptides from mammalian tissues. We set out to test the hypothesis that interventions known to increase lifespan, improve metabolic and stress resiliency, and boost endogenous $H_2S$ production would essentially expand and/or alter tissue-specific persulfidome profiles in vivo. To address this hypothesis, we employed two dietary interventions: (A) Short-term 1-week 50% caloric restriction in young adult (6 and 12 month old) male CGL wildtype (WT) and total body CGL knockout (KO) mice, and (B) Long-term 2.5-month every other day (EOD) fasting in aged adult (20 month old) C57BL/6 male mice. In the 50% caloric restriction scenario, we examined for diet-, tissue-, and genotype-specific changes in $H_2S$ production (i), protein persulfidation (ii), and biological pathways impacted by persulfidation (iii) (Fig. 1a). This type and duration of dietary intervention was chosen as it was previously shown to induce CGL expression and functional CGL-dependent hepatic $H_2S$ production in WT mice[25].

In this work, we identify persulfidated proteins and their associated biological pathways in the liver, kidney, heart, muscle (quadriceps), brain, and plasma. DR enriches the number of persulfidated proteins in the liver, kidney, muscle, and brain while it decreases these in the heart and has minimal impact on the plasma in WT mice. CGL KO mice have an overall decrease in persulfidated proteins identified in all tissues and fail to appropriately augment their persulfidome in response to DR relative to WT mice, chiefly in kidney, muscle, and brain. Likewise, long-term EOD fasting in aged mice enriches tissue-specific persulfidomes to a similar extent as 1-week 50% caloric restriction in young adult mice. These findings suggest a transformation of the protein persulfidome may be one mechanism as to how DR imparts its multi-tissue benefits to improve organismal health.

## Results

**Diet and CGL status impact tissue-specific $H_2S$ production.** Ad libitum (AL) fed mice were provided 24-h access to food and the 50% DR (DR) mice were fed their calculated allotment near the start of their dark phase at 7 pm to avoid disturbances in circadian

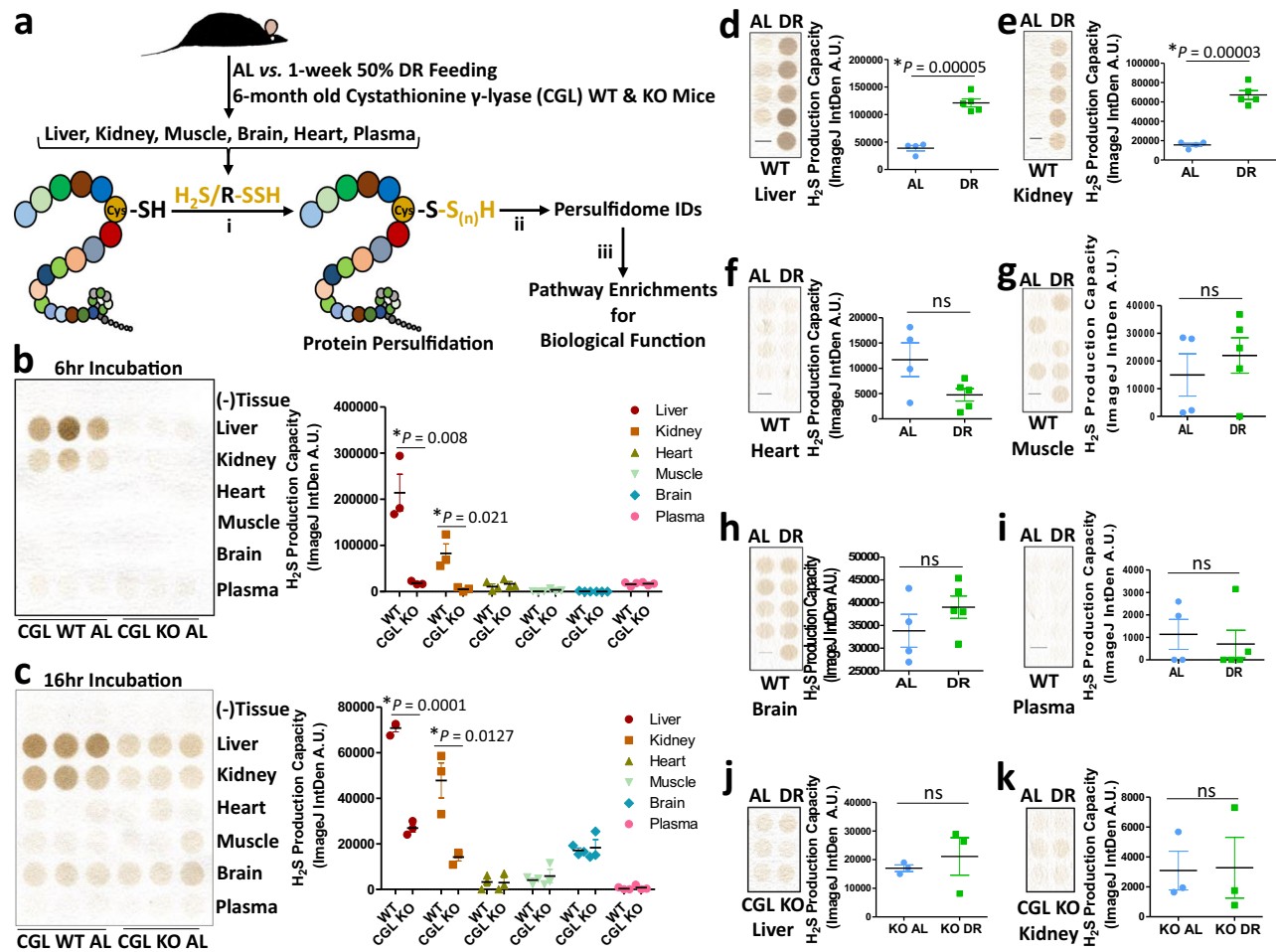

**Fig. 1 Diet and cystathionine γ-lyase (CGL) status impact H₂S production in a tissue specific manner. a** Graphical presentation of the overarching experimental setup. Six-month-old male CGL wildtype (WT) and total body knock out (KO) mice were placed on ad libitum (AL) or 50% dietary restriction (DR) diets for 1 week prior to tissue harvest. Tissues were analyzed for (**i**) H₂S production capacity via the lead acetate/lead sulfide method, (**ii**) protein persulfidation profiles via the biotin thiol (BTA) assay, and (**iii**) biological pathway enrichment/function of the identified persulfidated proteins. **b, c** H₂S production capacity in tissues in AL fed CGL WT (n = 3 mice/group) and CGL KO (n = 3 mice/group) mice at short 6 h (**b**) and long 16 h (**c**) exposures of the same assay plate. Means are shown and error bars are ± SEM. The asterisk indicates the significance of the difference in the average tissue-specific H₂S production capacity between CGL WT and KO mice after subtracting the (–) Tissue negative controls as background; with P values obtained from two-sided Student's t test. **d–i** H₂S production capacity in tissues harvested from CGL WT mice after 1 week of AL (n = 4 mice/group) or 50% DR (n = 5 mice/group) feeding. **j, k** H₂S production capacity in liver (**j**) and kidney (**k**) in CGL KO mice after 1 week of AL (n = 3 mice/group) or 50% DR (n = 3 mice/group) feeding. Means are shown and error bars are ± SEM. The asterisks indicate the significance of the difference between the dietary treatment groups; with P values obtained from two-sided Student's t test. ns not significant. Negative values after subtracting background have been converted to zero. The dash mark in the lead acetate/lead sulfide images in (**d–i**) indicates no sample loaded in that location. Source data are provided as a Source Data file. See also Supplementary Fig. 1.

rhythms and feeding patterns between the two groups[52,53]. Average daily food intake monitored over the 1-week intervention showed the AL group consumed 0.2 g food/g mouse, or ~12 kcal/mouse, per day while the DR group consumed 0.1 g food/g mouse, or ~6 kcal/mouse, per day (Supplementary Fig. 1a, b). DR for 7 days reduced body mass similarly in both CGL WT and KO mice by 5–10% of initial weight (Supplementary Fig. 1c, d).

After the 1-week dietary intervention, liver, kidney, heart, muscle (quadriceps), brain, and plasma were collected for downstream H₂S production capacity analysis via the lead acetate/lead sulfide endpoint method[54] and for persulfidation profiling (Fig. 1a). Similar to previous results obtained in our laboratory regarding the tissue specificity of CGL, CBS, and 3MST protein expression, H₂S production, and the dependence on CGL for H₂S production[32], and in line with unbiased

ENCODE transcriptome data of H₂S producing and consuming genes obtained from NCBI Gene Database[55] (Supplementary Fig. 1e–h), we show here H₂S production capacity in AL fed mice is highest in liver and kidney, and this is dependent on CGL (Fig. 1b, c). Heart, muscle, brain, and plasma showed low H₂S production capacities with little to no dependence on CGL (Fig. 1b, c). DR increased H₂S production capacity in the liver (Fig. 1d) and kidney (Fig. 1e), but had little to no impact in heart, muscle, brain, or plasma (Fig. 1f–i). As CGL status is important for liver and kidney H₂S production, we next focused on the impact of DR on H₂S production in these two organs in CGL KO mice. DR failed to increase detectable H₂S production capacity in the liver and kidney of CGL KO mice (Fig. 1j, k). Thus, H₂S production capacity is highest and malleable via DR in a CGL-dependent manner in the liver and kidney, while the other

whole-tissue homogenates tested have relatively low $H_2S$ production capacity independent of CGL status or diet.

**DR enriches the persulfidome in the liver, kidney, muscle, and brain.** To verify the ability of the modified BTA[43] (Supplementary Fig. 2a) to detect persulfidated proteins from tissues, we ran Western blot analysis on the liver BTA elution $-/+$ DTT for α-Tubulin and GAPDH, two hepatic proteins previously shown to be persulfidated[42] (Supplementary Fig. 2b). α-Tubulin and GAPDH were detected in the persulfidation-specific +DTT elution[43] from WT mice but not CGL KO mice nor in the –DTT elution lanes (Supplementary Fig. 2b). Pre-treatment of 1 mM NaHS on WT livers ex vivo prior to performing BTA resulted in decreased detection. This is indicative that ex vivo addition of 1 mM NaHS may act as a reducing equivalent, similar to DTT, and remove the in vivo formed persulfidation modifications prior to the BTA (Supplementary Fig. 2b). Thus, utilizing this modified BTA protein level isolation technique followed by mass spectrometry-based proteomics analysis for identification and label-free quantification (Supplementary Fig. 2a), we set out to examine tissue-, dietary-, and CGL-dependent persulfidome profiles. Two types of label-free mass spectrometry quantification analysis approaches were used for this study: peptide spectral counting and MS1 intensity. While the scientific narrative outlined in these Figures is based on the spectral counting method due to it affording relatively simple and fast data analysis in addition to its ability to quantify relative protein abundance over a dynamic range of 2 to 60 orders of magnitude[56–58], we have also supplied all MS1 intensity-based analyses in parallel. The presentation of intensity-based analyses is included in the Supplementary Data files as the second tab in the Excel sheets, with the first tab containing the respective spectral counting data. In these sheets are the quantitative and qualitative data with corresponding visual graphs and plots for both spectral counting and intensity analyses. Including data in this format has two benefits: (1) Ease of data access and visualization from both approaches, and (2) Ability to compare and utilize intermediate and end results from the two approaches, which ultimately show similar outcomes and emphasize the reported results in the narrative as robust and rigorously obtained. Additionally, side-by-side comparisons of these two approaches are described in the Peer Review File.

Of the six tissues tested in WT mice, four had enriched persulfidome profiles in response to DR when compared to AL feeding (Fig. 2a–d, Supplementary Fig. 2c–f, Supplementary Data 1–4). These tissues included liver (Fig. 2a, Supplementary Fig. 2c, Supplementary Data 1), kidney (Fig. 2b, Supplementary Fig. 2d, Supplementary Data 2), muscle (Fig. 2c, Supplementary Fig. 2e, Supplementary Data 3), and brain (Fig. 2d, Supplementary Fig. 2f, Supplementary Data 4). Total persulfidated proteins identified in each tissue included 977 in liver, 1086 in kidney, 431 in muscle, and 884 in brain. Identified proteins enriched under DR, displayed as green dots, included 30 in liver, 16 in kidney, 34 in muscle, and 369 in brain that met a biological- and statistical-significance threshold of at least a twofold increase in DR:AL spectral count ratio with a $P$ value $< 0.05$, respectively (Fig. 2a–d). Likewise, proteins enriched under AL feeding in these four tissues, displayed as blue dots, included 16 in liver, 5 in kidney, 0 in muscle, and 42 in brain that met a biological- and statistical-significance threshold of at least a twofold increase in AL:DR spectral count ratio with a $P$ value $< 0.05$, respectively (Fig. 2a–d). In total, 72%, 76%, 86%, and 87% of persulfidated proteins were skewed for enrichment under DR feeding in liver, kidney, muscle, and brain, respectively (Fig. 2a–d).

**Numerous pathways enriched with persulfidated proteins dependent on diet and organ.** In the four organs positively enriched for persulfidated proteins after DR, 1854 individual proteins were identified, with 209, or 11.3%, shared amongst all four tissues (Fig. 3a, Supplementary Data 5). Biological function and pathway enrichment of these 209 via g:Profiler analysis[59] utilizing the KEGG database[60] revealed 15 pathways, with carbon metabolism, proteasome, and valine, leucine, and isoleucine degradation as the top three hits (Fig. 3b). Of the 1854 proteins, 63 unique proteins were enriched under AL feeding (Fig. 3c) and 429 unique proteins under DR feeding (Fig. 3d), with none of these, shared amongst all four tissues.

As no common persulfidated proteins were enriched as a function of diet in all four tissues, we next examined tissue-specific pathway enrichment. The liver contained two pathways enriched under AL; selenocompound metabolism and terpenoid backbone biosynthesis, and one pathway enriched under DR; metabolic pathways (Fig. 3e, Supplementary Data 6). Kidney contained one pathway enriched under AL; glycine, serine, and threonine metabolism, and one pathway under DR; nitrogen metabolism (Fig. 3f, Supplementary Data 7). No pathways were enriched from the diet-specific persulfidated proteins in the muscle (Supplementary Data 8). Brain contained 12 pathways enriched under AL and 13 enriched under DR feeding (Fig. 3g, Supplementary Data 9). The top three pathways enriched under AL feeding in the brain included valine, leucine, and isoleucine degradation, fatty acid degradation, and metabolic pathways, while the top three pathways enriched under DR feeding in the brain included carbon metabolism, biosynthesis of amino acids, and glycolysis/gluconeogenesis.

Persulfidated proteins not significantly impacted by diet, which primarily compose the majority of the persulfidomes in liver, kidney, muscle, and brain, still trended for enrichment under DR (Fig. 2a–d; gray dots). Function and pathway enrichment of these proteins revealed 34, 41, 15, and 13 pathways in liver, kidney, muscle, and brain, respectively (Supplementary Fig. 3a–d). Ten of the pathways identified were shared amongst all four tissues, and include metabolic processes, carbon metabolism, valine, leucine, and isoleucine degradation, biosynthesis of amino acids, glyoxylate and dicarboxylate metabolism, cysteine and methionine metabolism, pyruvate metabolism, proteasome, citrate cycle (TCA cycle), and glycolysis/gluconeogenesis. While not shared amongst all tissues, pathways related to aging-related neurodegenerative Parkinson and Huntington diseases were enriched with persulfidated proteins in both kidney and muscle (Supplementary Fig. 3b, c). Thus, these four major metabolic organs respond to DR by augmenting protein persulfidation, with many of the proteins falling into specific biological pathways and functions.

**DR decreases heart and negligibly impacts plasma persulfidome profiles.** While DR enriched the persulfidomes of liver, kidney, muscle, and brain, it failed to do so in heart and plasma. In heart, 459 persulfidated proteins were identified with 45 enriched under AL feeding and 17 enriched under DR feeding (Fig. 4a, Supplementary Fig. 4a, and Supplementary Data 10). In plasma, 160 persulfidated proteins were identified with three enriched under AL feeding and zero enriched under DR feeding (Fig. 4b, Supplementary Fig. 4b, and Supplementary Data 11). Between heart and plasma, 78 common persulfidated proteins were identified (Fig. 4c, Supplementary Data 12), of which four biological functions/pathways were enriched, and include complement and coagulation cascades, staphylococcus aureus infection, HIF-1 signaling, and glycolysis/gluconeogenesis (Fig. 4d).

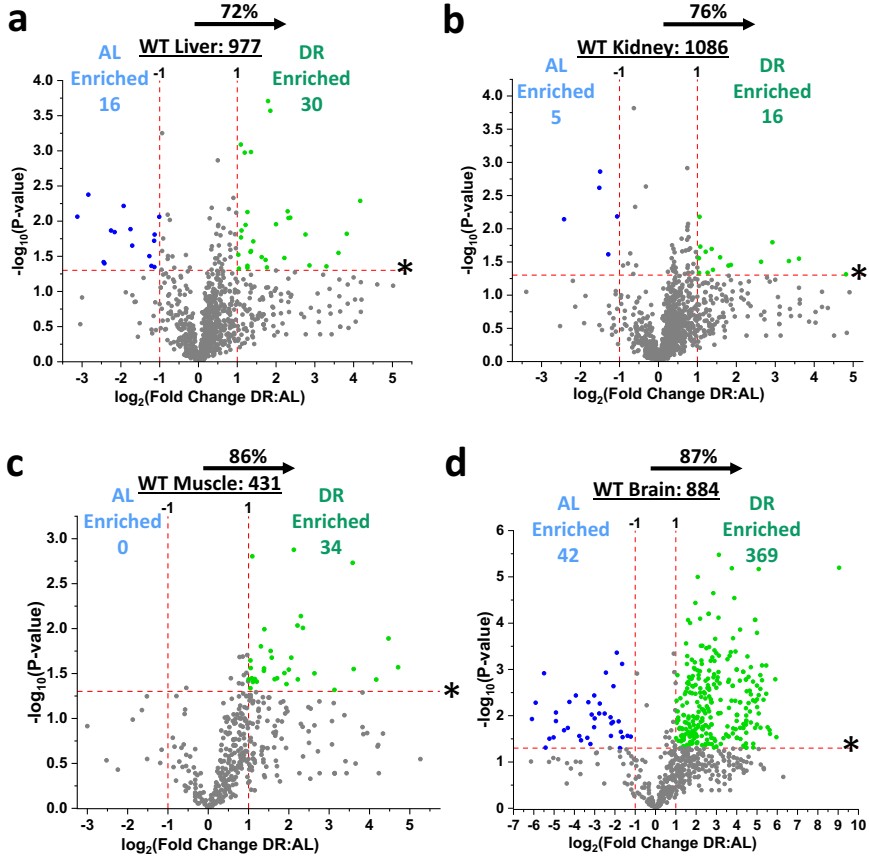

**Fig. 2 Dietary restriction enriches the persulfidation profiles in liver, kidney, muscle, and brain. a–d** Volcano plots showing differentially abundant persulfidated proteins in liver (**a**), kidney (**b**), quadriceps muscle (**c**), and whole brain (**d**) from ad libitum (AL; $n = 4$ mice/group) versus 50% dietary restriction (DR; $n = 5$ mice/group) fed cystathionine γ-lyase (CGL) wildtype (WT) mice. The log$_2$(Fold Change DR:AL) X-axis displays the average fold change in spectral counts for each identified persulfidated protein while the −log$_{10}$ Y-axis displays the calculated $P$ value when comparing the individual spectral count values for each identified persulfidated protein in a specific tissue from AL versus DR fed mice via a two-sided Student's $t$ test. The non-axial red dotted vertical lines highlight the biological significance threshold of + / − twofold change in spectral counts between DR versus AL, while the non-axial red dotted horizontal line with asterisk highlights our statistical significance threshold of $P < 0.05$. The number of total persulfidated proteins identified in each tissue are given next to the tissue name, while blue (AL enriched) and green (DR enriched) colored dots and text indicate persulfidated proteins reaching both biological- and statistical-thresholds. Gray color dots indicate persulfidated proteins not reaching the criteria for both biological and statistical significance for enrichment under either diet. The percentage of proteins skewed toward DR is provided above the tissue label. See also Supplementary Fig. 2.

Of the persulfidated proteins enriched under AL feeding in heart and plasma, zero were common between the two tissues (Fig. 4e). 

Examination of tissue-specific pathway enrichment revealed heart contained 12 pathways under AL with the top three being carbon metabolism, valine, leucine, and isoleucine degradation, and citrate/TCA cycle, and three pathways under DR; complement and coagulation cascades, ferroptosis, and staphylococcus aureus infection (Fig. 4f, Supplementary Data 13). Plasma had no diet-specific pathway enrichment (Supplementary Data 14). Pathway analysis of the persulfidated proteins not significantly impacted by diet revealed 27 and 6 pathways enriched in the heart and plasma, respectively. In heart, the top three pathways enriched were carbon metabolism, metabolic pathways, and pyruvate metabolism (Supplementary Fig. 4c). In plasma, the top three pathways enriched were complement and coagulation cascades, staphylococcus aureus infection, and prion diseases (Supplementary Fig. 4d).

To examine shared persulfidated proteins amongst all six tissues, we compared the 209 proteins identified as shared in liver, kidney, muscle, and brain with the 78 proteins identified as shared in heart and plasma. A total of 28 proteins were identified

as common in all six tissues (Fig. 4g, Supplementary Data 15). These 28 were involved in two blood-centric biological functions/ pathways, which included complement and coagulation cascades, and platelet activation (Fig. 4h), indicative of potential blood/ plasma carryover in the non-perfused tissues and thus suggestive that little to no common persulfidated proteins are inherent to all tissues.

**Late-life initiated long-term EOD fasting transforms tissue-specific persulfidomes**. While 1 week of 50% DR tested in the young adult mice is sufficient to temporarily enhance acute tissue-specific H$_2$S production capacity and stress resistance[25], it is most likely not long enough to extend longevity nor represent a tool for probing resiliency and adaptation at older ages. The impact of DR on longevity parameters sometimes arises more latently to the effects of reduced calories and/or timed feeding. Previously, Zivanovic, et al. showed that persulfidation in rat brain and heart tissues decreased with advanced age at 12–24 months, and that long-term daily 30% DR in mice between 2 to 20 months of age rescued aging-related persulfidation loss in

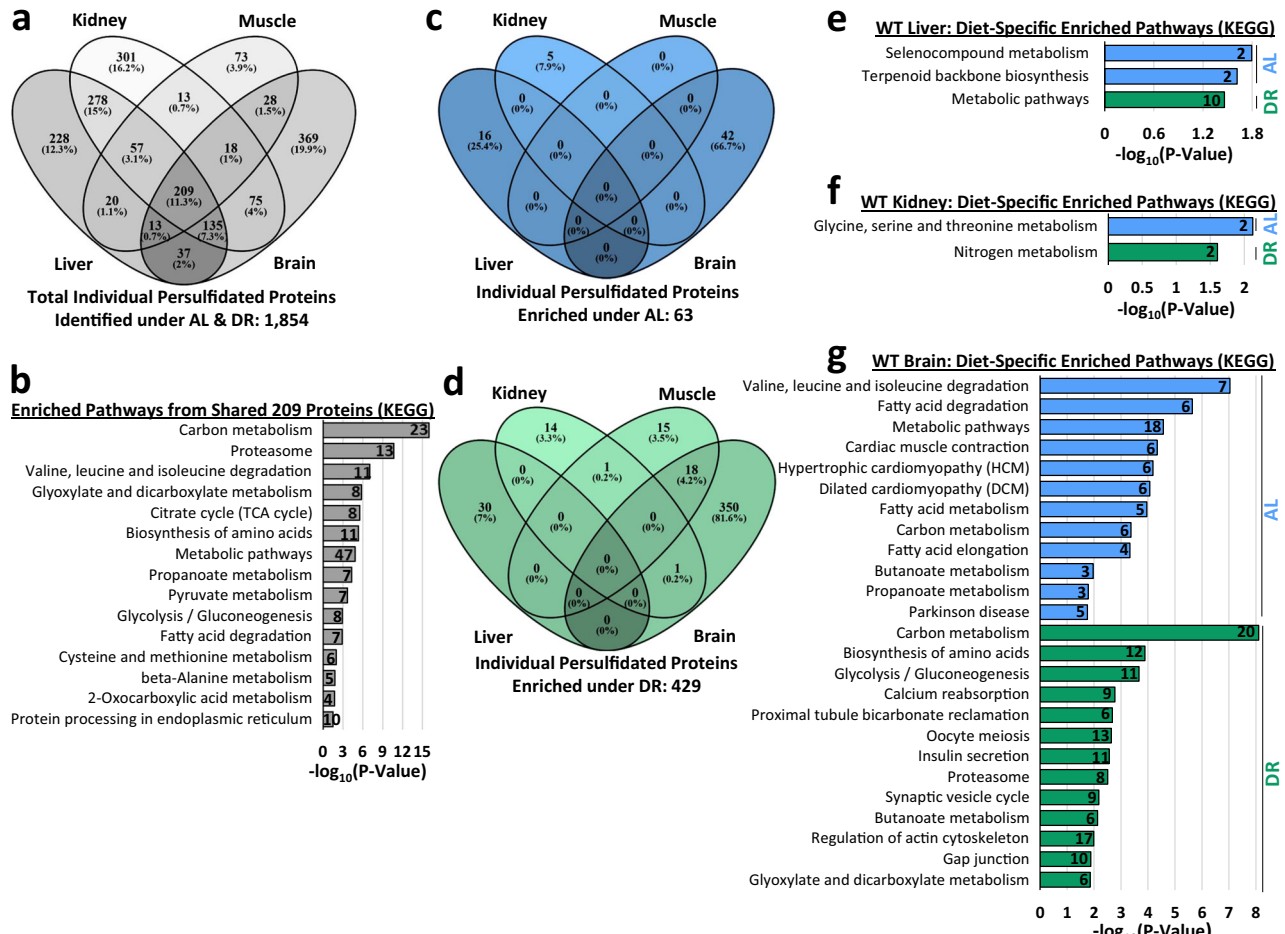

**Fig. 3 Multiple pathways enriched with persulfidated proteins dependent on diet and organ. a** Four-way Venn diagram showing the number of shared and non-shared persulfidated proteins identified in liver, kidney, muscle, and brain in cystathionine γ-lyase (CGL) wildtype (WT) mice. A total of 209 persulfidated proteins were shared amongst all four organs. $n = 9$ mice total; 4/AL group and 5/DR group. **b** KEGG biological function and pathway enrichment of the shared 209 proteins found in liver, kidney, muscle, and brain via g:Profiler analysis. **c, d** Four-way Venn diagrams showing shared and non-shared persulfidated proteins enriched under AL (**c**) and DR (**d**) feeding. **e–g** KEGG biological function and pathway enrichment of AL enriched (blue bars) or DR enriched (green bars) persulfidated proteins in liver (**e**), kidney (**f**), and brain (**g**). The numbers within the bars indicate individual persulfidated proteins identified for that specific pathway. Statistical significance for pathway enrichment plotted as the adjusted $-\log_{10}$ (P value) and was auto-calculated via the g:Profiler g:SCS algorithm for KEGG database that utilizes multiple testing corrections. See also Supplementary Fig. 3.

the liver[44]. However, studies examining the effects of long-term daily DR on older humans are somewhat difficult to execute due to patient adherence[61,62] and need for medical oversight to ensure caloric intake guidelines are safe followed[63–67]. Given these limitations, the relatively implementable yet effective DR regimen of intermittent fasting has gained substantial attention as an alternative method to continuous, chronic DR[68,69]. There are several types of intermittent fasting regimens, with every-other-day (EOD) fasting being one of the most popular[70,71]. In EOD fasting, animals and/or participants abstain from food for a day (i.e., 24 h) followed by food intake ad libitum (AL) for the next 24 h. This method is feasible in older adult humans[72], and in aged rodents augments lifespan[73–77] and improves metabolic flexibility[74].

Therefore, we next utilized late-life initiated EOD fasting for 2.5 months in 20-month-old male C57BL/6 mice to test if this more translatable form of DR is capable of altering tissue-specific persulfidomes in aged subjects (Fig. 5a). Similar to our findings in young adult mice, EOD fasting augmented protein persulfidation in liver (Fig. 5b, Supplementary Fig. 5a, Supplementary Data 16), kidney (Fig. 5c, Supplementary Fig. 5b, Supplementary Data 17), muscle (Fig. 5d, Supplementary Fig. 5c, Supplementary Data 18), and brain (Fig. 5e, Supplementary Fig. 5d, Supplementary

Data 19), while it decreased persulfidation in the heart (Fig. 5f, Supplementary Fig. 5e, Supplementary Data 20). Overall numbers of persulfidated proteins identified in each tissue from these aged mice were on the same order to what was found in the young adult mice. While the shift in protein persulfidation enrichment under 2.5 months EOD fasting in aged mice for kidney (Fig. 5c) and muscle (Fig. 5d) was not as robust as that found under 1-week 50% DR in young adult mice, a greater number of proteins meeting the biological- and statistical-thresholds were found in the EOD fasting group. Thus, despite being completely different forms and durations of DR, in addition to being tested on mice of different ages and strain background/source, long-term EOD fasting and short-term 50% DR alter tissue-specific persulfidomes in an analogous manner.

**CGL and regulation of its enzymatic activity as targets for persulfidation.** H$_2$S generating enzymes CGL and CBS were identified as targets for persulfidation in liver (Supplementary Data 1 and 16) and kidney (Supplementary Data 2 and 17) from young adult and aged mice in this current study, and in liver in a previous study by Mustafa, et al.[42]. However, functional consequences for CGL persulfidation have not been established.

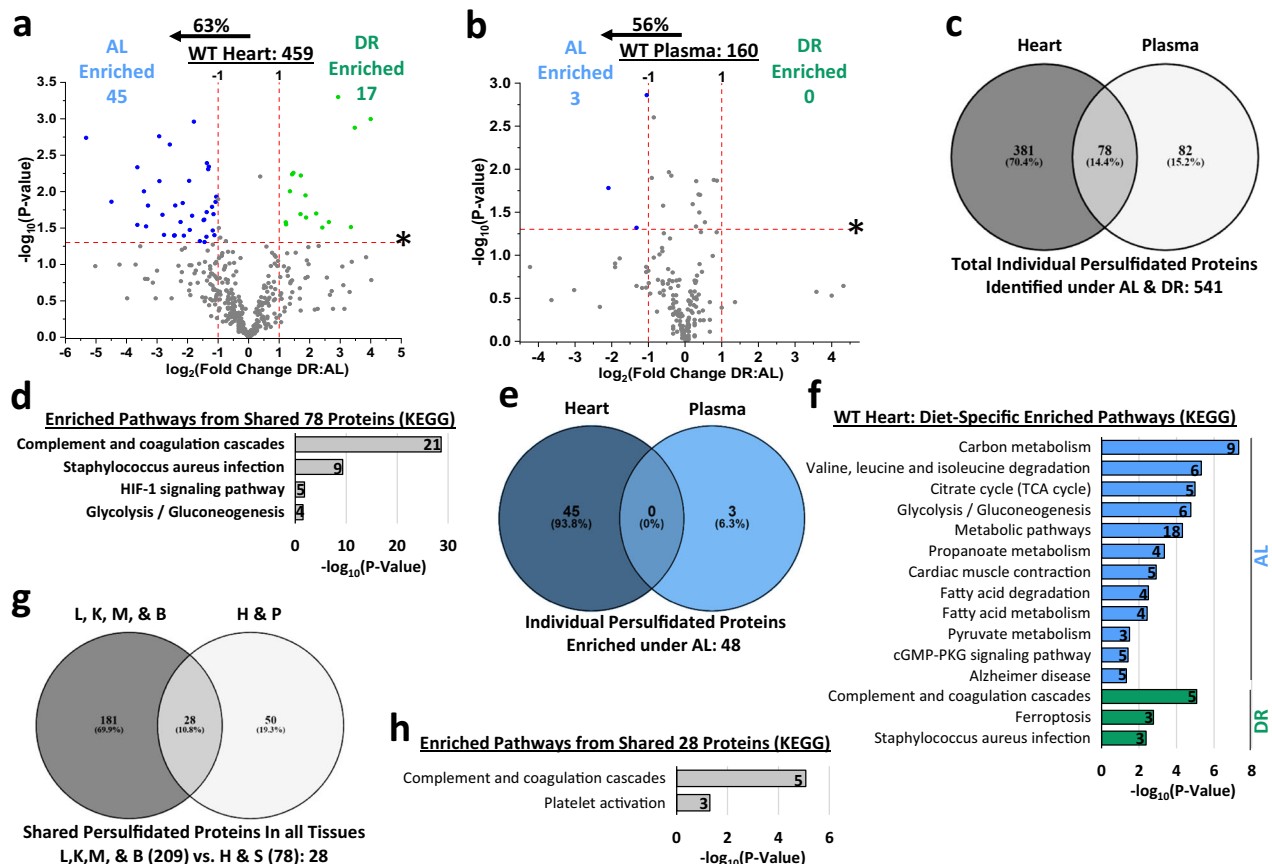

**Fig. 4 Persulfidomes of heart and plasma respond differently to diet compared to those of liver, kidney, muscle, and brain. a, b** Volcano plots showing differentially abundant persulfidated proteins in heart (**a**) and plasma (**b**) from AL ($n = 4$ mice/group) versus DR ($n = 5$ mice/group) cystathionine γ-lyase (CGL) wildtype (WT) mice. The $\log_2$(Fold Change DR:AL) X-axis displays the average fold change in spectral counts for each identified persulfidated protein while the $-\log_{10}$ Y-axis displays the calculated P value when comparing the individual spectral count values for each identified persulfidated protein in a specific tissue from AL versus DR fed mice with P values obtained from two-sided Student's t test. The number of total persulfidated proteins identified in each tissue are given next to the tissue name, while blue (AL enriched) and green (DR enriched) colored dots and text indicate persulfidated proteins reaching both biological- and statistical-thresholds. Gray color dots indicate persulfidated proteins not reaching the criteria for both biological and statistical significance for enrichment under either diet. The percentage and direction of proteins skewed toward AL is provided above the tissue label. **c** Venn diagram of shared and non-shared persulfidated proteins in heart and plasma. **d** KEGG biological pathway enrichment of the shared 78 proteins found in heart and plasma via g:Profiler analysis. **e** Overlap of AL enriched persulfidated proteins in heart and plasma. **f** KEGG biological function and pathway enrichment of AL enriched (blue bars) or DR enriched (green bars) persulfidated proteins in heart. **g** Venn diagram of shared and unshared persulfidated proteins amongst all 6 tissues. **h** KEGG biological function and pathway enrichment of the shared 28 proteins found in all six tissues via g:Profiler analysis. The numbers in the bars indicate individual persulfidated proteins identified for that specific pathway. Statistical significance for pathway enrichment plotted as the adjusted $-\log_{10}$ (P value) and was auto-calculated via the g:Profiler g:SCS algorithm for KEGG database that utilizes multiple testing corrections. See also Supplementary Fig. 4.

Given that reduction of redox-active cysteine residues in CBS increases $H_2S$ production capacity[78], we set out to determine if treatment of CGL with sodium hydrosulfide (NaHS) would impact its enzymatic $H_2S$ production capacity. To do so, we first adapted an in vitro lead acetate/lead sulfide headspace assay[54] to measure $H_2S$ production capacity from purified recombinant CGL protein. There was a linear increase in $H_2S$ production as a function of CGL concentration between 1–10 μg after a 2 h incubation at 37 °C (Fig. 6a, b). Next, we added NaHS between 0–1 mM to reaction mixtures with and without 1.125 μg purified CGL prior to incubation at 37 °C under normal room oxygen to examine for dose-dependent increases in enzymatic $H_2S$ production (Fig. 6c, d and Supplementary Fig. 6a–c). As NaHS itself gives an immediate positive lead sulfide result as it quickly releases $HS^-$ upon placement in aqueous solutions at physiological pH, two additional controls were utilized to diminish this

noise and allow for accurate detection of CGL-derived $H_2S$. First, multiple successive incubation periods each with a fresh lead acetate embedded detection paper were used to allow the NaHS to off-gas $HS^-$ during the early 1 h incubation period (Supplementary Fig. 6a) so as to not interfere with detection of enzymatic production during the subsequent 2.5, 5, and 6.5 h incubation periods (Fig. 6c, d and Supplementary Fig. 6b, c). Second, the (−) CGL lead sulfide integrated density values were subtracted from their dose-corresponding (+)CGL lead sulfide integrated density values. Taking these controls into account, we determined NaHS pretreatment of CGL prior to incubation at 37 °C resulted in dose-dependent increases up to 10–20% in enzymatic $H_2S$ production capacity (Fig. 6c, d and Supplementary Fig. 6b, c). Thus, the persulfidation of CGL detected in vivo combined with these in vitro results suggest a potential positive feedback loop regulating CGL activity and downstream $H_2S$ production.

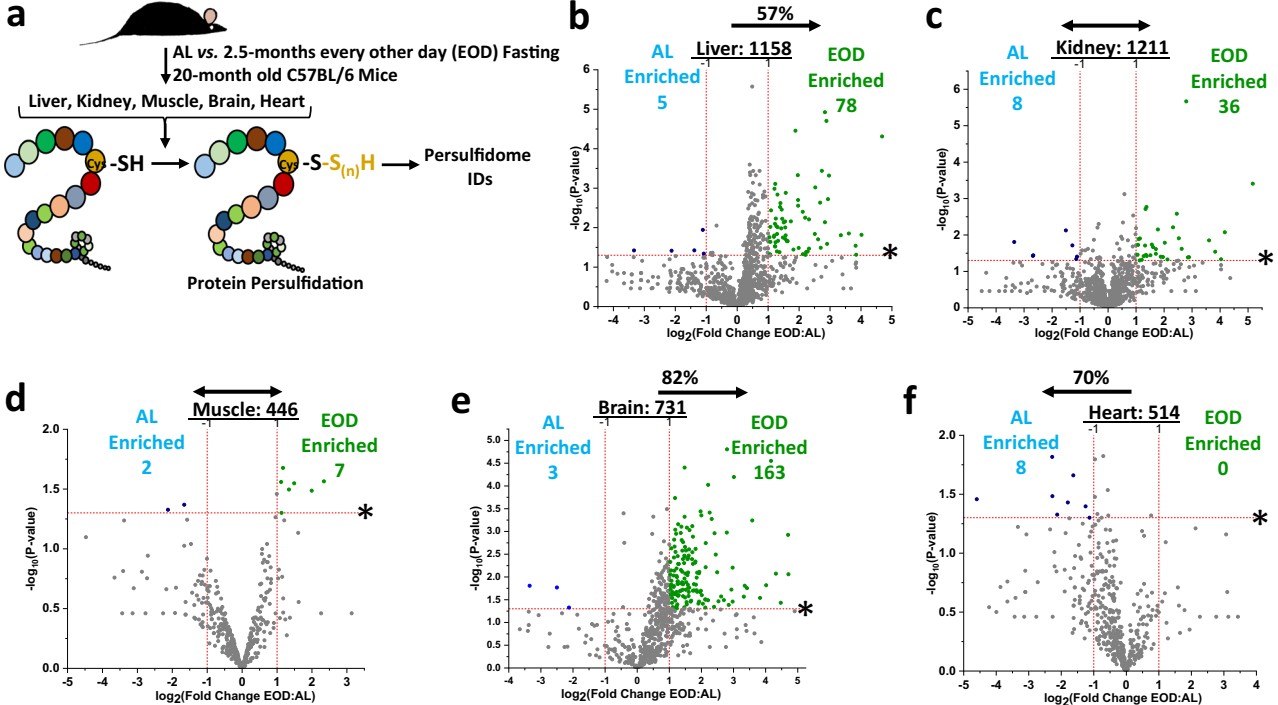

**Fig. 5 Late-life initiated every other day (EOD) fasting for 2.5 months transforms tissue-specific persulfidomes similar to short-term DR in young adult mice. a** Graphical presentation of the overarching experimental setup. Twenty-month-old male C57BL/6 mice were placed on ad libitum (AL) or Every Other Day (EOD) fasting for 2.5 months prior to tissue collection. Tissues were analyzed for protein persulfidation via the biotin thiol (BTA) assay. **b–f** Volcano plots showing differentially abundant persulfidated proteins in liver (**b**), kidney (**c**), quadriceps muscle (**d**), whole brain (**e**), and heart (**f**) from AL fed ($n = 5$ mice/group) versus 2.5 months of EOD fasting ($n = 5$ mice/group) 20-month-old C57BL/6 male mice. The $\log_2$(Fold Change DR:AL) X-axis displays the average fold change in spectral counts for each identified persulfidated protein while the $-\log_{10}$ Y-axis displays the calculated P value from a two-sided Student's t test when comparing the individual spectral count values for each identified persulfidated protein in a specific tissue from AL versus EOD fed mice. The non-axial red dotted vertical lines highlight the biological significance threshold of $+/-$ twofold change in spectral counts between EOD versus AL, while the non-axial red dotted horizontal line with asterisk highlights our statistical significance threshold of $P < 0.05$. The number of total persulfidated proteins identified in each tissue are given next to the tissue name, while blue (AL enriched) and green (EOD enriched) colored dots and text indicate persulfidated proteins reaching both biological- and statistical-thresholds. Gray color dots indicate persulfidated proteins not reaching the criteria for both biological and statistical significance for enrichment under either diet. The percentage and direction of proteins skewed toward one diet type are provided above the tissue label. See also Supplementary Fig. 5.

**CGL deficiency limits the persulfidome under AL and/or DR feeding in a tissue-dependent manner.** We next tested the requirement for CGL in diet-induced persulfidome changes. As basal and enhanced diet-induced $H_2S$ production in liver and kidney were primarily CGL driven, we first examined persulfidation in these two tissues from 6-month-old CGL KO mice under AL and 1 week 50% DR feeding. In the liver, 698 persulfidated proteins were detected; ~30% reduction compared to CGL WT mice (Fig. 7a, Supplementary Fig. 7a, and Supplementary Data 21). Despite this overall reduction, DR still enriched the liver persulfidome in CGL KO mice (61% skewed toward DR; Fig. 7a) similarly to WT mice (72% skewed toward DR; Fig. 2a). However, in comparing total persulfidated proteins of WT and CGL KO mice, we discovered ~33% unique to WT mice, ~6% unique to KO mice, and ~62% shared (Fig. 7b). Pathway enrichment for the CGL-dependent proteins (Supplementary Data 22) included metabolic pathways, oxidative phosphorylation, Parkinson's disease, Huntington's disease, non-alcoholic fatty liver disease, lysosome, Alzheimer's disease, and thermogenesis (Fig. 7c). In comparing DR-enriched persulfidated proteins in WT and CGL KO mice, we found 55% unique to WT mice, 46% unique to KO mice, and zero shared (Fig. 7d). The DR-enriched proteins unique to the KO livers failed to account for any pathway enrichment. Thus, despite CGL being dispensable for DR-induced overall

enrichment of the liver persulfidome (suggesting a potential involvement of CBS and 3MST), it may be required for targeted protein-specific persulfidation.

In the kidney, 869 persulfidated proteins were detected in CGL KO mice; ~20% reduction compared to WT mice (Fig. 7e, Supplementary Fig. 7b, and Supplementary Data 23). Loss of CGL resulted in failure to enrich the persulfidome under DR, with 74% of persulfidated proteins skewed toward AL feeding (Fig. 7e), which was an equal and opposite response detected in WT mice, which had 76% skewed toward DR (Fig. 2b). In comparing total kidney persulfidated proteins of WT and CGL KO mice, ~28% were unique to WT mice, ~10% unique to KO mice, and ~63% shared (Fig. 7f). Pathway enrichment for the CGL-dependent proteins (Supplementary Data 24) included metabolic pathways and valine, leucine, and isoleucine degradation (Fig. 7g). In comparing DR-enriched persulfidated proteins in WT and CGL KO kidneys, ~76% were unique to WT mice, ~24% unique to KO mice, and zero shared (Fig. 7h), with the DR-enriched proteins unique to KO kidneys failing to account for any pathway enrichment. Thus, CGL is required for DR-induced persulfidation enrichment in the kidney.

We next examined tissues with DR-induced shifts in persulfidome profiles yet no discernable dependence for CGL or diet in their $H_2S$ production capacities. Muscle, brain, and heart

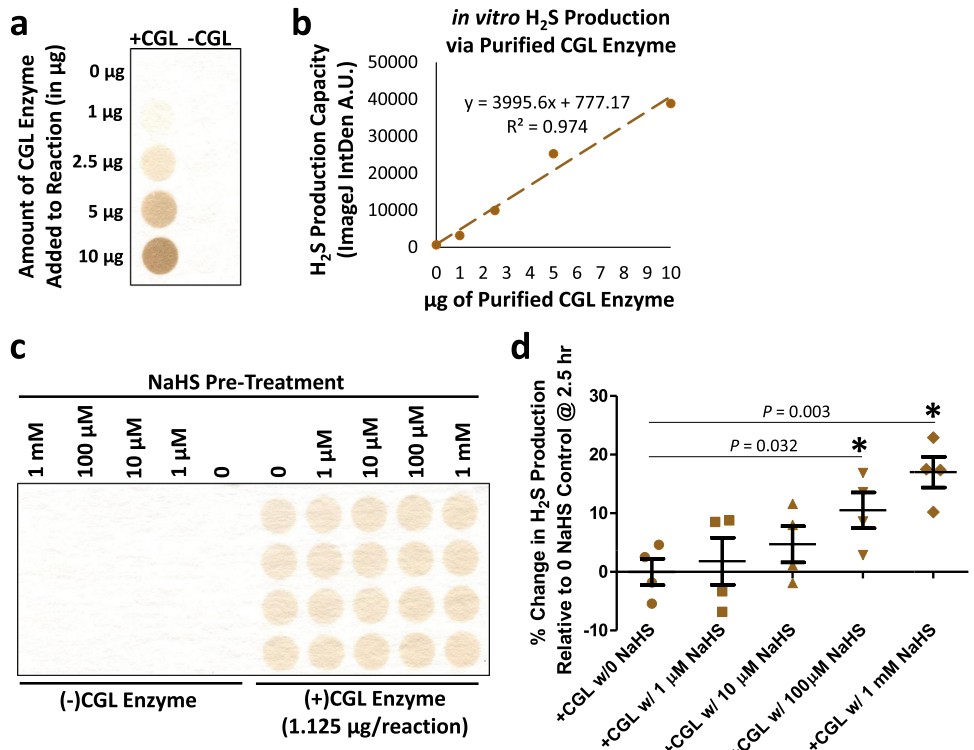

**Fig. 6 NaHS pretreatment augments H$_2$S production capacity of CGL. a**, **b** Detection via the lead acetate/lead sulfide method (**a**) and quantification via integrated densitometry (**b**) of CGL-derived H$_2$S production in reaction mixtures containing L-cysteine and pyridoxal phosphate after 2 h incubation at 37 °C. **c**, **d** Pretreatment with NaHS prior to incubation at 37 °C dose dependently increases CGL-derived H$_2$S production capacity after correcting for detection of non-enzymatic H$_2$S off-gassed in the corresponding (−)CGL wells (**c**). Plot of the percent change in H$_2$S production relative to the (+)CGL w/0 μM NaHS control. Error bars are ± SEM. The given individual $P$ values are calculated from two-sided $t$ test without multiple comparison test, while asterisks indicate $P < 0.05$ calculated from one-way ANOVA with Dunnet's Multiple Comparison Test. $n = 4$/reaction condition. Source data are provided as a Source Data file. See also Supplementary Fig. 6.

from CGL KO mice had decreased numbers of total persulfidated proteins compared to WT mice by 56, 39, and 59%, respectively (Fig. 8a–c, Supplementary Fig. 8a–c, and Supplementary Data 25, 26, and 27). DR failed to enrich the muscle persulfidome in CGL KO mice (Fig. 8a), however, DR still enriched brain (Fig. 8b) and attenuated heart (Fig. 8c) persulfidomes in CGL KO mice. In comparing total persulfidated proteins from these three tissues, we discovered: (1) 472 in muscle with 284 (~60%) unique to WT mice, 41 (~9%) unique to KO mice, and 147 (~31%) shared (Fig. 8d), (2) 998 in the brain with 459 (~46%) unique to WT mice, 114 (~11%) unique to KO mice, and 425 (~43%) shared (Fig. 8e), (3) 479 in the heart with 292 (~61%) unique to WT mice, 20 (~4%) unique to KO mice, and 167 (~35%) shared (Fig. 8f). The top ten biological pathways enriched by CGL-dependent persulfidated proteins in the muscle (Supplementary Data 28), brain (Supplementary Data 29), and heart (Supplementary Data 30) are shown in Fig. 8g–i, respectively, and include carbon metabolism, oxidative phosphorylation, and neurodegenerative-related pathways.

Further examining the extent for CGL in shaping the persulfidome under DR, we compiled individual and shared persulfidated proteins from the five solid tissues in WT (Fig. 8j) and CGL KO mice (Fig. 8k). A total of 3837 proteins were plotted in WT and 2481 proteins in KO, resulting in an ~35% decrease due to loss of CGL. While the fold difference between the numbers of DR-enriched and AL-enriched persulfidated proteins meeting the statistical and biological thresholds were similar in WT (4.3-fold) and KO (4.2-fold), the proteins in CGL KO mice failed to fall under specific functional pathway enrichments. Proteins not meeting statistical and biological thresholds are

prominently shifted towards enrichment under DR in WT mice, with 71% +/−20% with a DR:AL spectral count ratio above 1 (Fig. 8j, Supplementary Data 31), while this same subset of persulfidated proteins is shifted towards enrichment under AL in CGL KO mice, with only 42% +/−30% having a DR:AL spectral count ratio above 1 (Fig. 8k, Supplementary Data 31). Importantly, similar results were obtained via MS1 intensity analysis of this subset of persulfidated proteins from the five solid tissues (Fig. 8l), with 72% +/−25% having a DR:AL intensity ratio above 1 in CGL WT mice (Supplementary Data 31), while in CGL KO mice only 38% +/−34% had a DR:AL intensity ratio above 1 (Supplementary Data 31). Thus, both label-free quantification mass spectrometry analysis techniques utilized ultimately produced similar results and highlighted the tissue specificity and dependence on CGL for transforming persulfidomes under DR.

**Orthogonal protein and labeled peptide approaches show CGL dependence for DR-induced persulfidation shifts.** Persulfidome profiles generated thus far were performed on protein level pulldown and subsequent label-free quantification (LFQ). While suitable for protein persulfidation identification and showing relative shifts between groups, these methods are not without their drawbacks. Mainly, protein-level pulldown lends to potential false positive identification due to intermolecular disulfide bonds, thus allowing a non-persulfidated protein to be co-captured with a persulfidated protein and eluted during the final reducing step[39,79]. Conversely, false negatives are possible due to NM-Biotin bound non-persulfidated intramolecular thiols (RS-NM-Biotin) on the same protein as a labeled persulfidated cysteine residue (RS-S$_n$-NM-Biotin) preventing final elution with

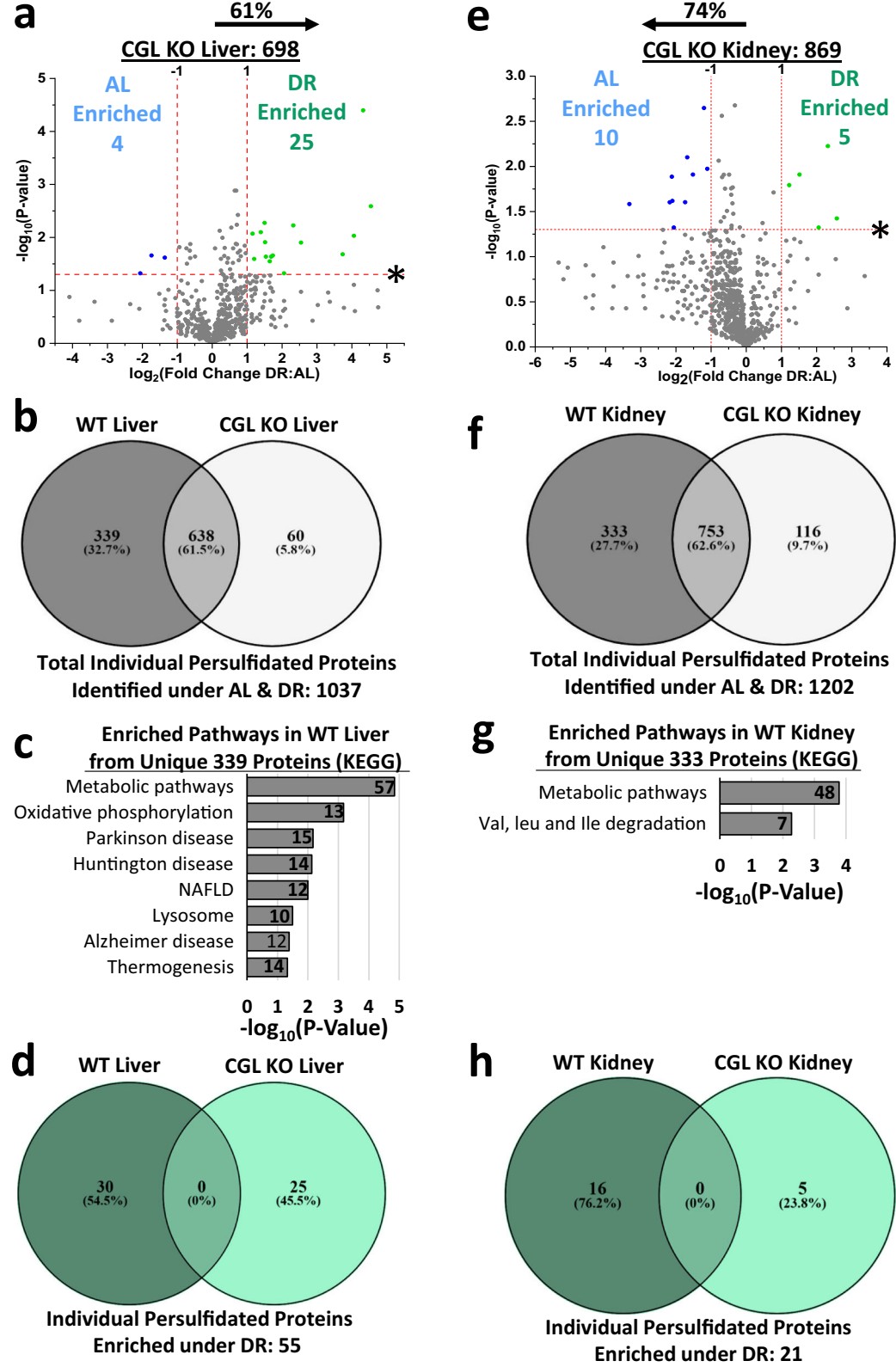

a, e Volcano plots; b, f Venn diagrams of total individual persulfidated proteins identified under AL & DR; c, g enriched pathways in WT from unique proteins (KEGG); d, h Venn diagrams of individual persulfidated proteins enriched under DR.

a reducing agent[79]. While these caveats are valid, it has been hypothesized that concentrations of disulfide bonds in intracellular proteins are low and not expected to be a major source of false positives in the approach utilized[39]. Nonetheless, we pursued an orthogonal approach (Fig. 9a and Supplementary Fig. 9a) to avoid these limitations and further validate and expand our

findings utilizing kidney and brain obtained from an independent cohort of 12-month-old CGL WT and KO mice fed 1 week 50% DR.

Direct comparison between WT and KO kidney utilizing protein level BTA revealed the persulfidome enriched in WT with 87% of all proteins having a higher LFQ intensity in WT over KO

**Fig. 7 Diet induced persulfidation changes in the liver and kidney of cystathionine γ-lyase (CGL) knockout (KO) mice. a** Volcano plot showing differentially abundant persulfidated proteins in liver from AL ($n = 3$ mice/group) versus DR ($n = 3$ mice/group) fed CGL KO mice. The $\log_2$(Fold Change DR:AL) X-axis displays the average fold change in spectral counts for each identified persulfidated protein while the $-\log_{10}$ Y-axis displays the calculated $P$ value from a two-sided Student's $t$ test when comparing the individual spectral count values for each identified persulfidated protein in liver from AL versus DR fed mice. Blue (AL enriched) and green (DR enriched) dots and text indicate persulfidated proteins reaching both biological (twofold enrichment)- and statistical ($P < 0.05$)- thresholds. Gray dots indicate persulfidated proteins not reaching the criteria for biological and statistical significance under either diet. The percentage and direction of proteins skewed toward one diet type is provided above the tissue label. **b** Venn diagram of shared and unshared persulfidated proteins between CGL WT and KO livers. **c** Enriched function/pathway analysis of the 339 persulfidated proteins unique to CGL WT liver. **d** Venn diagram to examine shared and unshared DR enriched persulfidated proteins in WT and CGL KO livers. **e–h** are similar to **a–d**, but analysis was performed in kidneys. **e** Volcano plot showing differentially abundant persulfidated proteins in kidney from AL ($n = 3$ mice/group) versus DR ($n = 3$ mice/group) fed CGL KO mice. **f** Shared and unshared persulfidated proteins between CGL WT and KO kidneys. **g** Enriched function/pathway analysis of the 333 persulfidated proteins unique to CGL WT kidney. **h** Shared and unshared DR enriched persulfidated proteins in WT and CGL KO kidneys. Significance for pathway enrichment plotted as the adjusted $-\log_{10}$ ($P$ value) and was auto-calculated via the g:Profiler g:SCS algorithm for KEGG database that utilizes multiple testing corrections. See also Supplementary Fig. 7.

(Fig. 9b, Supplementary Fig. 9b, and Supplementary Data 32). Similarly, 123 proteins in WT met the biological- and statistical-significance LFQ thresholds, versus only seven meeting similar thresholds in KO (Fig. 9b). Direct comparison between WT and KO brains utilizing protein level BTA resulted in a near-even split in total persulfidated proteins, with 55% skewed toward WT (Fig. 9c, Supplementary Fig. 9c, and Supplementary Data 33). These results are not surprising given: (1) The equal but opposite direction the kidney persulfidome is shifted under DR in 6-month-old CGL WT (76% toward DR; Fig. 2b) and KO mice (74% toward AL; Fig. 7e), and (2) The comparable brain persulfidome enrichments in 6-month-old CGL WT (87% toward DR; Fig. 2d) and KO mice (86% toward DR; Fig. 8b).

We next examined CGL dependence for DR-induced shifts utilizing peptide level BTA (Fig. 9a). Tissue lysates underwent trypsinization to digest proteins and produce smaller peptides prior to affinity purification, thus overcoming potential inter-molecular and intramolecular cysteine-related issues experienced in protein level BTA[79]. In kidney, 70% of peptides and resultant protein ID's had higher LFQ intensities in WT over KO (Fig. 9d, Supplementary Fig. 9b, and Supplementary Data 34). Seven proteins in WT met the biological- and statistical-significance LFQ thresholds, versus only two meeting similar thresholds in KO (Fig. 9d). In brain, 74% of the peptides and resultant protein ID's were skewed toward WT (Fig. 9e, Supplementary Fig. 9c, and Supplementary Data 35). Six proteins in WT met the biological- and statistical-significance LFQ thresholds, versus only one meeting similar thresholds in KO (Fig. 9e). Thus, peptide level pulldown showed CGL-dependent kidney persulfidation enrichment under DR similar to protein level pulldown, while it revealed a CGL dependence for brain persulfidation enrichment under DR that was not detected by protein level BTA analyses.

Advancing the peptide level BTA, we applied multiplex iodoacetyl isobaric tandem mass tag (iodoTMT) labeling of sulfhydryl groups after final tris(2-carboxyethyl)phosphine (TCEP) elution of persulfidated (Fig. 9a and Supplementary Fig. 9a). These steps and approaches advance upon non-labeled peptide BTA as TCEP reduction and subsequent iodoTMT labeling provide quantitative identification of persulfidated proteins and peptide site-level analysis for the modified cysteine residues[45,80,81]. In both kidneys (Fig. 9f and Supplementary Data 36) and brain (Fig. 9g Supplementary Data 37) total, persulfidated peptides and resultant protein ID's had higher iodoTMT intensities in WT over KO, with 100% enrichment in WT kidney and 80% enrichment in WT brain. Twelve proteins in WT kidney and two proteins in WT brain met the biological- and statistical-significance for WT:KO iodoTMT intensity ratio, while no proteins reached similar thresholds in KO tissues (Fig. 9f, g). Interestingly, when examining kidneys from AL fed WT and KO

mice, the shift in iodoTMT intensities favored the KO group (68%; Supplementary Fig. 9d and Supplementary Data 38), despite decreased $H_2S$ production capacity in KO kidneys (Fig. 1b, c). This suggests: (1) Basal persulfidation activity under AL feeding or unstressed conditions is independent of CGL and/ or CGL derived $H_2S$, and (2) Enhanced persulfidation in select tissues relies not only on CGL derived $H_2S$ but also its coordination with other cysteine- and thiol-redox modifications induced by DR[82,83] or other stressors[45,84]. Site-level analysis for the modified cysteine residues from peptide iodoTMT labeling is provided in the second tabs of Supplementary Data 36 for DR kidney, Supplementary Data 37 for DR brain, and Supplementary Data 38 for AL kidney.

## Discussion

Enhanced $H_2S$ production and signaling, particularly via CGL activity, has only recently been recognized as a common phenomenon and proposed mechanisms of action in models of longevity[16,36,38]. However, the mechanisms underlying the pleiotropic cellular, physiological, and systemic benefits of $H_2S$ have remained unclear. As aging is a multifaceted and complex decline in numerous physiological and metabolic functions, the role $H_2S$ plays in combating these declines theoretically should not be limited to one pathway or mechanism. Most likely, it is through $H_2S$ performing several non-mutually exclusive functions related to antioxidant and redox homeostasis[85], mitochondrial electron transfer[86], and protein modification and signaling via persulfidation[39,42,43]. In the current study, we focused on the latter mechanism, as little has been revealed regarding the entirety and scope of persulfidation in mammals across several organ systems while under an anti-aging intervention.

Here, we found CGL dependent alterations in $H_2S$ production capacity and persulfidome enrichment induced by DR to vary based on tissue, summarized in Fig. 10. Utilizing three independent experimental mouse cohorts differing in age, diet type, diet duration, and/or strain background along with several variations of the BTA, we found DR enhanced the persulfidomes of liver, kidney, muscle, and brain while contracted those of heart. Dependence for CGL in diet-induced persulfidome expansion was most prominent in kidney and muscle, and to a certain extent in brain. While this is not the first study to recognize relative changes in mammalian protein persulfidation as a function of CGL or diet[44,87], it is the first to reveal individual identities and enrichment shifts of these proteins, their biological pathway involvement, sites of modification, and the interdependence of tissue-type, diet, and CGL to impact persulfidation profiles.

While the results obtained in liver and kidney were predictable based upon their diet-malleable $H_2S$ production capacities, we did

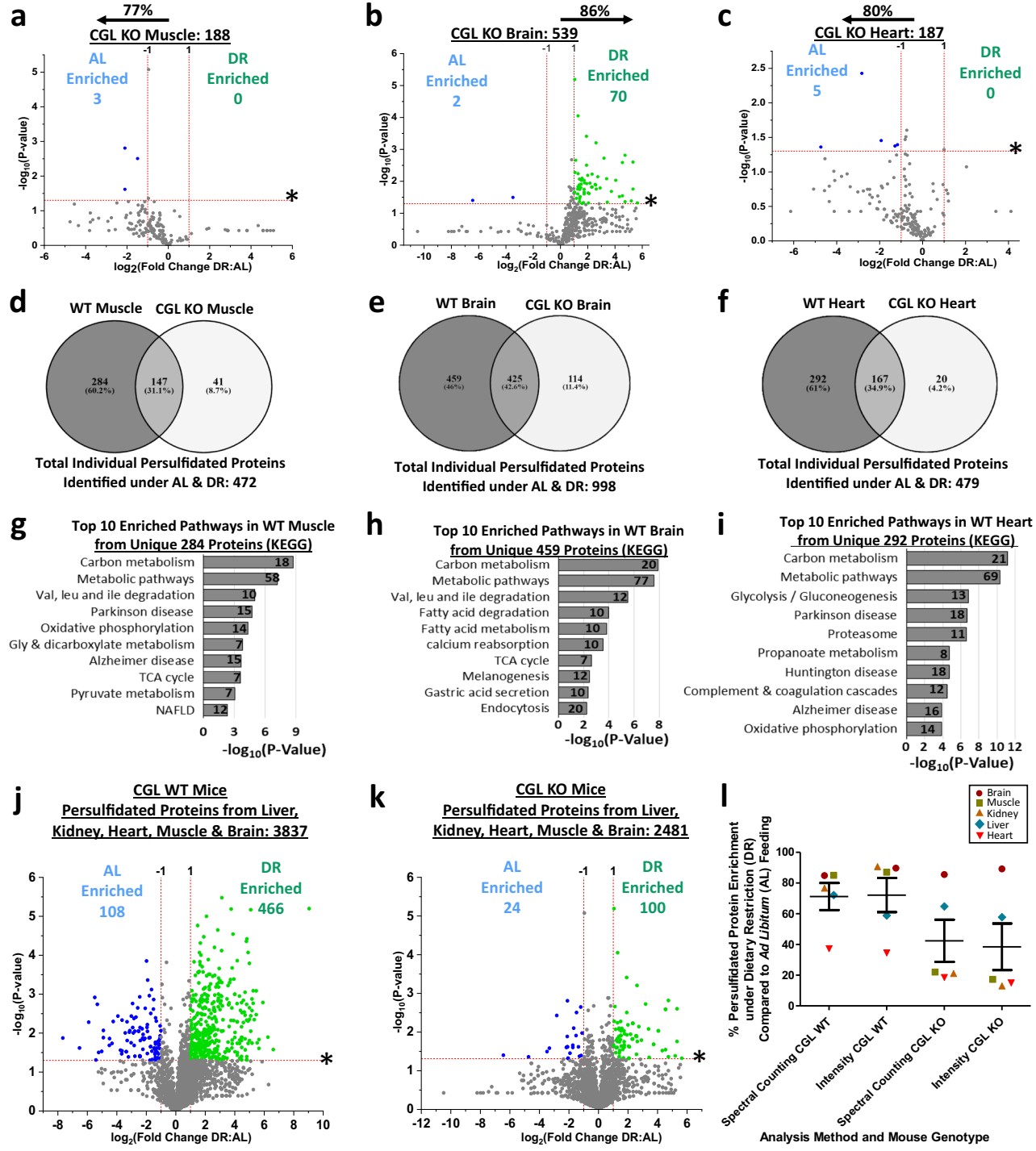

not expect increased muscle and brain persulfidation under DR. As both of these tissues fail to augment detectable $H_2S$ production capacity under DR, this suggests DR-mediated enhanced persulfidation may not be tissue autonomous and/or more complex than previously thought. The sources of $H_2S$ that enhance persulfidation in these weak-producing tissues may arise distally from liver or kidney, or from diet-modifiable $H_2S$ producing endothelial cells residing within the tissue[33]. While the idea of $H_2S$ being a circulating gasotransmitter in the blood is somewhat controversial[88], our data show these weak producing tissues still somehow augment their persulfidomes under DR. The failure to detect increased $H_2S$ production capacity or enriched

persulfidation in plasma may suggest DR does not induce systemic increases in sulfide availability, and local enrichment of persulfidation may be tissue autonomous via direct enzymatic persulfidation of proteins via CGL, CBS, and/or 3-MST or through local endothelial production. However, only plasma was assayed and not white and red blood cells, of which the latter act as a sink for $H_2S$ via exchange in the microcirculation[89]. Thus, we cannot entirely rule out DR-enhanced $H_2S$ in circulation as the mechanism for augmented persulfidation in marginally $H_2S$ producing tissues. Future studies are needed to test the requirement of direct protein–protein interactions for CGL, CBS, and 3-MST in DR-induced persulfidation changes.

**Fig. 8 CGL dependent and independent persulfidation and pathway enrichment in muscle, brain, and heart. a–c** Volcano plots showing differentially abundant persulfidated proteins in muscle (**a**), brain (**b**), and heart (**c**) from AL ($n = 3$ CGL KO mice/group) versus DR ($n = 3$ CGK KO mice/group). The $\log_2$(Fold Change DR:AL) X-axis displays the average fold change in spectral counts for each identified persulfidated protein while the $-\log_{10}$ Y-axis displays the calculated $P$ value from a two-sided Student's $t$ test when comparing the individual spectral count values for each identified persulfidated protein from AL versus DR fed mice. Blue (AL enriched) and green (DR enriched) dots and text indicate persulfidated proteins reaching both biological (*twofold enrichment*)- and statistical ($P < 0.05$)- thresholds. Gray dots indicate persulfidated proteins not reaching the criteria for both biological and statistical significance under either diet. The percentage and direction of proteins skewed toward one diet type is provided above the tissue label. **d–f** Shared and unshared persulfidated proteins between CGL WT and KO muscle (**d**), brain (**e**), and heart (**f**). **g–i** Pathway analysis of persulfidated proteins unique to CGL WT muscle (**g**), brain (**h**), and heart (**i**). Numbers indicate the proteins involved in that specific pathway. Significance for pathway enrichment plotted as the adjusted $-\log_{10}$ ($P$ value) and was auto-calculated via the g:Profiler g:SCS algorithm for KEGG database that utilizes multiple testing corrections. **j, k** Volcano plots and **l** dot plot re-imaging data from Figs. 2, 4, 7, and 8 showing differentially abundant persulfidated proteins in liver, kidney, muscle, brain, and heart from CGL WT AL versus CGL WT DR mice (**j**) and from CGL KO AL versus CGL KO DR mice (**k**) with the calculated $P$ values from a two-sided Student's $t$ test when comparing individual spectral count values for each identified persulfidated protein from AL versus DR fed mice. In (**l**), the mean $+/-$ SEM is plotted for % persulfidated protein enrichment under DR compared to AL feeding from each solid tissue in CGL WT and KO mice as determined through spectral counting or MS1 intensity. Source data are provided as a Source Data file. See also Supplementary Fig. 8.

Additionally, the absolute number of persulfidated proteins in the brain and their expansion under DR was surprising due to the purported sensitivity of the central nervous system (CNS) to H$_2$S-induced toxicity related to low or non-existent sulfide quinone oxidoreductase (SQR) expression[90,91]. SQR is a key mitochondrial enzyme catalyzing initial steps of H$_2$S oxidation, detoxification, and removal[92,93]. However, H$_2$S serves beneficial roles in the brain, as decreased H$_2$S or disruption of H$_2$S producing enzymes in the brain or neuronal cells are associated with aging-related Parkinson's, Huntington's, and Alzheimer's disease-like phenotypes[94], while controlled H$_2$S exposure protects against cerebral ischemia-reperfusion injury[95]. This suggests the lack of SQR enables H$_2$S that reaches or is produced in the CNS to be utilized primarily for protein persulfidation.

Higher SQR activity in liver, kidney, or heart potentially limits protein persulfidation via the oxidation and removal of H$_2$S. This may explain the decrease in persulfidation in heart upon DR, as H$_2$S may impart its beneficial cardiac functions[96,97], particularly under ischemia-reperfusion injury, via SQR-mediated mitochondrial oxidation of H$_2$S rather than persulfidation-centric mechanisms. Conversely, the decrease in cardiac persulfidation upon DR may be through potential depersulfidation of proteins via the thioredoxin (Trx) system[79,87]. The function and potential benefit of removing sulfhydryl groups on proteins in cardiac tissue upon DR is not well understood or described in our study. However, Trx mediated cleavage of cysteine persulfides results in H$_2$S release[87]. Thus, a potential hypothesis is that depersulfidation in the heart releases free H$_2$S and/or polysulfides for potential beneficial use elsewhere, perhaps via SQR for mitochondrial protection and appropriate changes in metabolism and fuel usage during ischemia reperfusion. Further examination into Trx depersulfidase activity during DR, and if it is stunted in tissues with enhanced persulfidome profiles and boosted in tissues with decreased persulfidome profiles, is warranted. This also offers an alternative explanation for the changes in tissue-specific persulfidomes as a function of DR that is mutually exclusive of H$_2$S production capacities.

Reactive oxygen and nitrogen species (RONS), much like H$_2$S, display hormetic dose responses and act as secondary messengers. Similarly, RONS production during DR through increased mitochondrial fatty acid oxidation is thought to drive many of the benefits of DR and DR-mimetics[82,98]. In light of the requirement for cysteine residue activation via oxidation or a redox thiol switch to occur prior to persulfidation by H$_2$S[43–45], it is appropriate to consider the interdependence of DR-mediated augmentation in RONS and H$_2$S for promoting changes in the persulfidome during DR. Likewise, non-dietary means to extend lifespan in the worm *Caenorhabditis elegans* triggered generation

of RONS and H$_2$S, with both factors responsible for stress resistance and slowed aging[84]. Thus, metabolic and stress cues leading to increased endogenous H$_2$S production also boost RONS production, and these RONS potentially prime cysteine residues for persulfidation. Additionally, enhanced H$_2$S production may prevent further irreversible oxidation of cysteine residues while under stressful conditions such as DR. This hypothesis is further supported by our observations showing basal persulfidation activity in the kidney under AL feeding is independent of CGL despite there being a large differential in H$_2$S production capacity in kidneys from AL fed CGL WT and KO mice, and it is not until DR is applied that concomitant enhanced persulfidation activity and H$_2$S production capacity via CGL is detected. These findings also bring to light the importance of oxygen tension during mammalian cell culture experiments being factored in as a variable for persulfidation assays, as the typical 20% O$_2$ cell cultures are exposed to, rather than 1–11% found in vivo[99], is hyperoxic and in itself may pose as a redox stressor impacting persulfidation readouts.

There are several biological-related limitations to our study. First, we examined only whole organ lysates, thus region-specific persulfidation enrichments, particularly in highly structured and region-specific organs such as the brain, are not reported. Second, the pathway enrichment analyses simply show involvement of the persulfidated proteins in specific pathways, but the direction of metabolic flux or pathway activation/silencing is not inferred. Third, our experimental animal setup may limit or influence the persulfidomes identified, as we only used male mice and under conditions of fitness-improving diets. Changing sex, variables related to circadian patterns and time of tissue collection, and diet lipid composition could expand or contract persulfidation[37,100,101]. Likewise, it was recently reported that enhanced H$_2$S production capacity under DR was only detected in mouse strains that gain longevity benefits under DR, so it would be interesting to study how persulfidation changes under DR in non-responding or negative-responding mouse strains[102].

There are also several technical-related limitations to our study. The first to address is the ratio of maleimide-biotin to protein added during the alkylation step in protein level BTA. Lower concentrations of maleimide-biotin result in selective labeling of the more highly reactive persulfidated cysteine groups versus the less reactive non-persulfidated cysteine thiol[43]. Increasing the ratio of maleimide-biotin to protein may underestimate persulfidated proteins identified by labeling both persulfidated cysteines and non-persulfidated cysteines on the same protein, thus preventing elution with DTT from the avidin columns. At the same time, this strategy offers more selective isolation of proteins with multiple persulfidation modifications. Here, we used a slightly

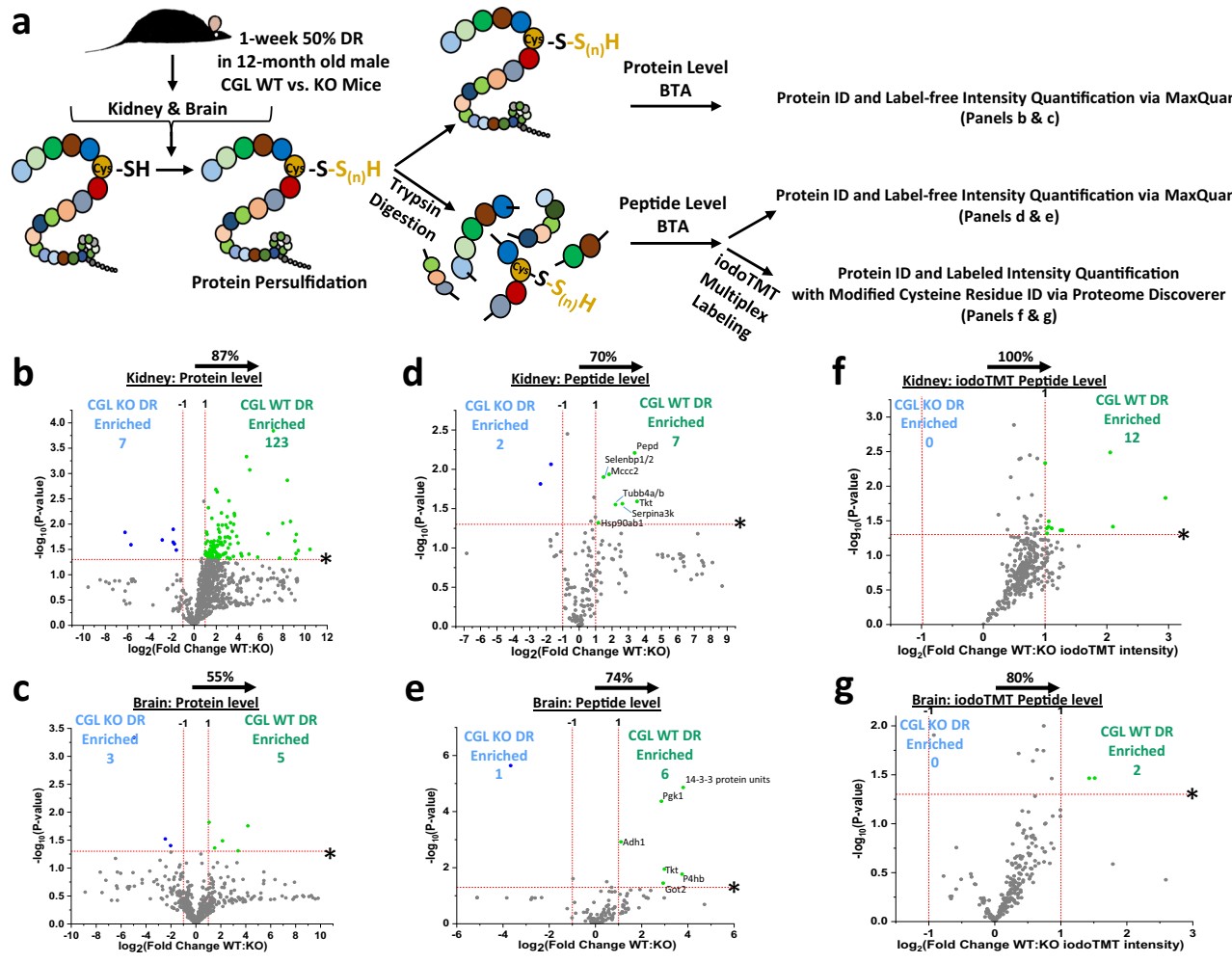

**Fig. 9 Orthogonal protein and peptide-based biotin thiol assay (BTA) approaches show CGL dependence for DR-induced persulfidation shifts.**
**a** Graphical presentation of experimental setup. Kidney and brain extracts from 12-month-old CGL WT and KO mice under 1 week 50% DR were subjected to protein level or peptide level BTA, LC-MS/MS, and analyzed via the given software platform ($n = 3$ mice/group). Furthermore, an additional peptide level BTA was subjected to iodoTMT labeling prior to LC-MS/MS for quantification and site-level analysis of cysteine residues ($n = 3$ mice/group). **b**–**g** Volcano plots showing CGL-dependent intensity shifts in DTT-eluted & unlabeled (**b**–**e**) or TCEP-eluted & iodoTMT labeled (**f**, **g**) persulfidated proteins. **b**, **c** Derived from unlabeled protein level BTA for kidney (**b**) and brain (**c**). **d**, **e** Derived from unlabeled peptide level BTA for kidney (**d**) and brain (**e**). **f**, **g** Derived from iodoTMT labeled peptide level BTA for kidney (**f**) and brain (**g**). The log₂(Fold Change WT:KO) X-axis displays the average fold change in intensity for each protein, and the −log₁₀ Y-axis displays the calculated $P$ value from a two-sided Student's $t$ test when comparing the individual intensity values for each protein from WT versus KO mice. The non-axial red dotted vertical lines highlight the biological significance threshold of +/− twofold change in intensity, while the non-axial red dotted horizontal line with asterisk highlights the statistical significance threshold of $P < 0.05$. Blue (KO enriched) and green (WT enriched) dots indicate proteins reaching both biological- and statistical-thresholds. The percentage and direction of proteins skewed toward CGL WT is provided above the tissue label. See also Supplementary Fig. 9.

higher ratio of maleimide-biotin to protein than utilized previously[43]. Thus, the latter scenario may hold true and altering maleimide-biotin concentrations during alkylation would give different persulfidome results. Second, similar to intramolecular cysteines providing challenges in protein level BTA, intermolecular disulfide bonds can result in potential false-positive identification due to non-persulfidated proteins co-captured with a persulfidated protein and elute during the reducing step[39,79]. However, it has been hypothesized that this issue is not expected to be a major source of false positives[39]. Nonetheless, it still has the potential to artificially expand or mask persulfidome shifts. For these reasons, we pursued predigested peptide level BTA in Fig. 9, in which we detected diet- and CGL-dependent enrichments in tissue-specific persulfidomes similar to our protein level BTA, thus validating and expanding upon the data generated from the latter method.

The third variable to address is the choice of alkylating agent. The use of maleimide derivatives to alkylate thiols and persulfides is well established[45,48,50,103,104]. Yet, a few studies indicate they also bind to sulfenic and sulfinic acids under specific experimental conditions[105,106], making way for potential false-positive persulfidation hits. However, these false positives may be minor as the interaction between maleimide derivatives and oxidized cysteine residues are not as stable under pH 7–8[106], which is optimal pH for thiol and persulfide alkylation with maleimide[107] and what we used in the current study. In addition, the reported low presence of oxidized cysteine species relative to thiols and persulfides limits the potential for false positives in the analysis of persulfides in the proteome[105]. Importantly, the concentration of maleimide derivatives used in the studies reporting interaction with oxidized cysteines ranged from 5 mM to 20 mM, which is 14- to 58-fold higher than what was used in the present study,

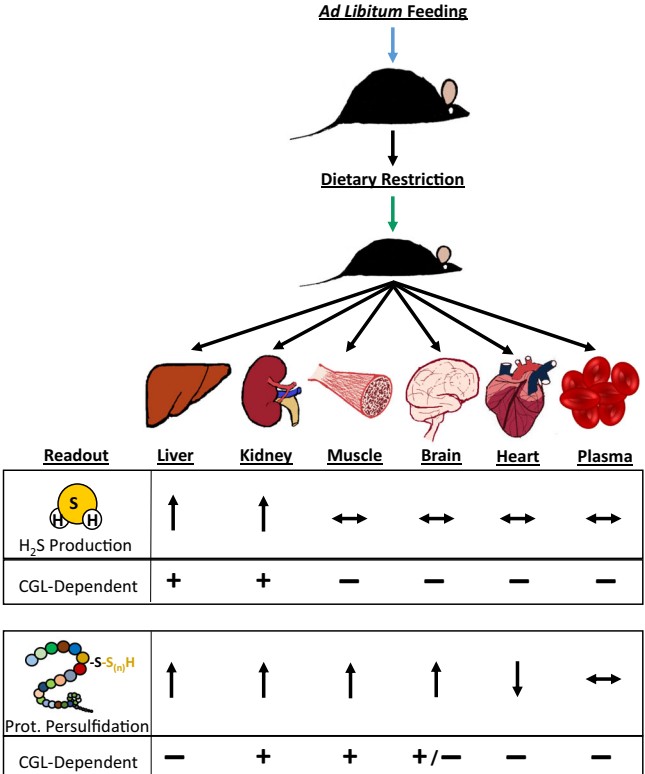

**Fig. 10 Model depicting the impact of dietary restriction (DR) on endogenous H₂S production and protein persulfidation in tissue-specific and CGL-dependent manners.** Arrows pointing up indicate an increase, arrows pointing down indicate a decrease, horizontal arrows indicate no change, and + or − relate to CGL dependence for the indicated DR-induced changes relative to ad libitum fed controls.

suggesting our lower concentration may avoid off-target labeling. Furthermore, in larger scale mammalian whole tissue cysteine modification studies, much like ours, maleimide derivatives preferentially alkylate and block R–SH groups and not R–SOH groups on proteins[51]. Given the enzymatic activity of CGL[108] is likely to only prompt alterations of cysteines with a persulfide and not sulfenic, sulfinic, or sulfonic modifications, the relatively minimal impact of oxidized cysteines giving false-positive results within our methodology is leveraged by our use of CGL KO mice, in which: (1) There is a reduction in total protein persulfidation numbers in all tissues from KO mice, and (2) There is an attenuation of DR induced persulfidation in kidney and brain of KO mice as detected in direct WT versus KO studies utilizing protein and peptide level BTA. Thus, while false positives are conceivable, the diet-induced and tissue-specific proteomic shifts detected in a CGL-dependent manner in this study most likely infer changes in persulfidation.

Thus, due to the dynamic rather than static nature of the persulfidome and the biological or technical experimental factors listed above, the generation of alternative persulfidome profiles is likely. Despite these limitations, our work provides a snapshot of these proteins and their persulfidation changes specific to tissue, diet, peptide site, and CGL status. Future work is needed to identify how these proteins are persulfidated; i.e., passive or targeted, commonalities or motifs in the peptide sequence that prompt for persulfidation, and the functional changes in protein stability, location, and activity induced by these modifications.

In summary, we reveal the global persulfidation profile alterations in major metabolic organs under ad libitum and dietary restriction. We show the importance of CGL enzyme for

protein persulfidation in multiple major organs. In addition to testing the necessity for CGL to shape baseline and diet-induced changes in persulfidomes, we also identified CGL as a target for persulfidation and its potential impact on increasing H₂S generating capacity. Ultimately, this study establishes a data resource for dietary restriction and H₂S fields while prompting the need to decipher downstream effects of these persulfidated proteins in metabolism, stress resistance, and longevity.

## Methods

**Animal husbandry and diet intervention**. All experiments were performed with approval of the Cleveland Clinic Institutional Animal Care and Use Committee, protocol # 2016–1778, and followed the National Institutes of Health Guide for the Care and Use of Laboratory Animals. In experiments utilizing male cystathionine gamma-lyase (CGL) wildtype (WT) and knockout (KO) mice, the mice were bred and weaned at 21–23 days of age and maintained under standard barrier housing in the Cleveland Clinic Lerner Research Institute on a 14-h light/10-h dark cycle, temperature between 20–23 °C, 30–70% relative humidity, and with initial ad libitum access to standard rodent food (Envigo #2918) and drinking water until the dietary intervention. Experimental cystathionine γ-lyase (CGL) mice were obtained from initial parental CGL Het x Het breeding to generate CGL WT and KO F1s, and then breeding CGL WT x WT or CGL KO x KO F1s to generate the littermate and/or age-matched F2 experimental animals. Mice were group-housed with 3–5 mice per cage. The CGL WT and KO mice were originally generated on a mixed 129/C57BL/6 background as described previously[25,27] and then subsequently rederived into pathogen-free C57BL/6 mice at Jackson Laboratories prior to colony establishment at Cleveland Clinic. When mice were 6 or 12 months old they were switched to the experimental AIN-93G-based diet (Research Diets D10012G-2V-Formula 1) and exposed to ad libitum access for several days to adapt to the new food and for monitoring intake. The experimental diet consists of 20% of calories from protein (casein-based containing ~2.9% methionine and 0.4% cysteine, with additional L-cystine supplemented to a final 1.5 g/1000 g final diet composition by Research Diets, Inc.), 64% of calories from carbohydrate, and 16% calories from fat, and importantly has 2x concentrations of mineral mix S10022G, vitamin mix V10037, and choline bitartrate to avoid potential micronutrient malnutrition during 50% dietary restriction. The powdered food mix was added in a 1:1 ratio (gram:mL) to a 2% agar (Sigma #A1296) solution in water before solidification to a semi-solid consistency that lessens the potential for food hoarding in group housing and increases the accuracy of food consumption measurement. Ad libitum food intake per cage was measured daily for up to 4 days to determine the correct amount to restrict to achieve 50% reduction in food intake. After randomly assigning ad libitum (AL) (n = 3–4/genotype/experiment) or diet restriction (DR) (n = 3–5/genotype/experiment) feeding to the cages, food intake, and body mass were measured over the 1-week intervention. AL fed mice were provided 24-h access to the diet and the DR mice fed their calculated allotment near the start of their dark phase at 7 pm to limit disturbances in circadian rhythms and feeding patterns between the two groups[52,53].

For aged mice experiments testing every-other-day (EOD) fasting, male C57BL/6 mice were obtained between 60–65 weeks of age (Stock no. 000664, Jackson Laboratories, Bar Harbor, ME) and group-housed (up to five same-sex mice per cage) in the Cleveland Clinic Lerner Research Institute Biological Resource Unit. The mice had ad libitum (AL) access to standard rodent chow (18.6% protein, 44.2% carbohydrate, and 6.2% fat; Teklad Global Rodent Diet #2918, Envigo, Madison, WI). At ~20 months of age, cages were randomly assigned to either EOD fasting (n = 5) or AL access (n = 5) to the standard rodent chow. The EOD fasting regimen consisted of repeated cycles of 24 h consecutive removal of food access with water always available (fast day) followed by 24-h access to food and water (fed day). The EOD fasting intervention proceeded for 2.5 months. To circumvent possible disturbances in circadian rhythms and feeding patterns in the Chow EOD group, the food was provided to or removed from the Chow EOD group just prior to the dark cycle onset at 7 pm.

After the noted dietary intervention periods, mice were euthanized in the late morning (between 9am–12pm) via isoflurane anesthesia overdose followed by cervical dislocation prior to collecting liver, kidney, heart, muscle (quadriceps), brain, and plasma. Plasma was collected via retro-orbital bleed and the blood immediately placed into lithium-heparin-coated tubes (Terumo #T-MLH). Tubes were centrifuged to separate RBCs from the plasma. Collected tissues were immediately placed in 1.5 mL centrifuge tubes, flash frozen in liquid nitrogen, and then stored in the dark at −80 °C until further analysis. Subsequent analysis of tissues was by researchers not involved in the live animal experiments and was initially done blinded in regards to individual animal identify then unblinded for downstream assays and data analysis for grouping purposes.

**Lead acetate/lead sulfide assay to detect enzymatic H₂S production**. Tissue H₂S production capacity was measured by the lead acetate/lead sulfide method[25,54]. Briefly, ~80 mg of tissue is first placed in 1.5 mL microcentrifuge tubes containing 250 μL of 1x passive lysis buffer (Promega) and homogenized, followed by multiple rounds of flash freezing/thawing using liquid nitrogen. After homogenization and

lysis, protein concentration was measured with bicinchoninic acid assay (BCA) kit (Bio-Rad) followed by normalization of proteins via additional 1x passive lysis buffer. Next, the lead acetate/lead sulfide assay is setup by initially preparing the reaction mixture of 10mM L-cysteine (Sigma #168149) and 1 mM pyridoxal phosphate (PLP) (Sigma #9255) in PBS, with 150 μL placed into each well of a 96-well plate. Hundred micrograms of protein from each tissue or 20 μL of plasma are added to each respective well, then the plate overlaid with lead acetate embedded filter paper and incubated at 37 °C until lead sulfide is detected for quantification using ImageJ (version: 1.51n) densitometry analysis via the IntDen function after subtracting background levels obtained from reaction mixture-only wells with no tissue or protein added. A similar approach was used to examine purified recombinant CGL-derived $H_2S$ production in vitro, with reactions run in 50 mM Tris (pH ~7.8) + 10 mM L-Cys, + 1 mM PLP, and +/− purified recombinant human CGL (Cayman Chemical Company #10329 +/− NaHS treatment in the given amounts and concentrations noted in the figure and figure legend.

**Mouse gene expression profiles**. RNA expression graphs of $H_2S$ producing and consuming proteins were generated utilizing the Mouse ENCODE project[55] with values extracted from the NCBI Mouse Gene Database at the following NCBI webpages: CGL: https://www.ncbi.nlm.nih.gov/gene/107869, CBS: https://www.ncbi.nlm.nih.gov/gene/12411, 3-MST: https://www.ncbi.nlm.nih.gov/gene/246221, and SQR https://www.ncbi.nlm.nih.gov/gene/59010.

**Protein level biotin thiol assay (BTA)**. The isolation and detection of persulfidated proteins were performed using an adaptation to the biotin thiol assay (BTA) as previously described by Gao, et al.[43]. To isolate persulfidated proteins, tissues were first lysed with RIPA lysis buffer (Thermo Fisher Scientific, #89900) containing protease inhibitor cocktail (Thermo Fisher Scientific, #78415). Protein concentration was determined via BCA kit (Bio-Rad) and all concentrations normalized. Next, 7 mg of protein were incubated with 343 μM Maleimide-PEG2-biotin (Thermo Fisher Scientific, #21901BID) in RIPA buffer in a total volume of 500 μL at pH 7.4 at room temperature for 30 min with agitation. Maleimide was chosen as the alkylating agent because it predominantly interacts with sulfhydryl groups (-SH and $-SS_nH$) of cysteines versus other cysteine modifications in non-reducing buffers at physiological pH of 7–7.5[50,103,104]. Precipitation of the proteins was done by adding 1 mL of 100% cold acetone at −20 °C for 30 min followed by centrifugation. Additional washes were performed with 1 mL of 75% cold acetone followed by centrifugation and removal of acetone. Proteins were then resuspended in 0.25 mL of suspension buffer (RIPA + 1% SDS, pH 7.5) followed by adding 0.75 mL of neutralization buffer (30 mM Tris, 1 mM EDTA, 150 mM NaCl, 0.5 % Triton X-100, pH 7.5). Prior to adding the alkylated proteins to the streptavidin-agarose resin containing spin columns, the columns (Pierce, Catalog no. 69705) were first washed with PBS two times, followed by adding 0.39 mL streptavidin-agarose resin (Thermo Scientific, #20347) to the column and washing the resin 1x with PBS. The resin was then equilibrated with a 1:4 suspension buffer:neutralization buffer washes four times prior to plugging the outlet of the column. Alkylated proteins were then added to the streptavidin-agarose resin containing spin columns and kept rotating overnight at 4 °C. After the overnight incubation, the protein resin mix in the columns was washed six times in 0.85 mL of wash buffer 1 (30 mM Tris, 1 mM EDTA, 150 mM NaCl, 0.5% Triton X-100, pH 7.5) followed by another six washes with 0.85 mL of wash buffer 2 (30 mM Tris, 1 mM EDTA, 600 mM NaCl, 0.5% Triton X-100, pH 7.5) and finally another three washes with 0.85 mL of wash buffer 3 (30 mM Tris, 1 mM EDTA, 100 mM NaCl, pH 7.5). All washes were done using the Vac-Man Laboratory Vacuum Manifold (Promega A7231). Resin with bound proteins was first incubated with freshly prepared 500 μL elution buffer (30 mM Tris, 1 mM EDTA, 100 mM NaCL, pH 7.5) without DTT for 30 min at 25 °C followed by centrifugation of the spin tube inside of a 2 mL collection tube at 2000 g for 2 min to collect the (−) DTT negative control eluate. The same spin column was then placed into a new 2 mL collection tube and 500 μL of freshly prepared elution buffer containing 20 mM DTT was added for 30 min at 25 °C followed by centrifugation at 2000 g for 2 min to elute persulfidated proteins. The addition of DTT in the final elution step selectively elutes persulfidated proteins, as it breaks the S–S bond but not the S–NM–Biotin bond at pH between 7–7.5[43,48,109], as can be seen with the –DTT control gel lanes having lower detectable proteins compared to the +DTT gel lanes (Supplementary Fig. 2b–f), supporting the effectiveness of the BTA method. Eluted proteins were run through Amicon Ultracel 10 K (Millipore, #UFC501096) to concentrate in a final volume of 50 μL. After obtaining these final eluates, subsequent gel electrophoresis analysis and mass spectrometry-based label-free protein identification is commenced to visualize, identify, and quantify the persulfidated proteins.

**Peptide level BTA**. Seven milligrams of proteins were extracted and alkylated with 343 μM Maleimide-PEG2- biotin as described above in the protein level BTA assay. Alkylated proteins were precipitated with ice-cold acetone, resuspended in denaturation buffer (8 M urea, 1 mM MgSO4 and 30 mM Tris–HCl, pH 7.5). After denaturation for 10 min at 90 °C proteins were diluted with seven volumes of dilution buffer (1 mM CaCl2, 100 mM NaCl, and 30 mM HEPES-NaOH pH 7.5), followed by incubating with trypsin (Promega, #V5111) (ratio of 1:50 W/W) overnight at 37 °C. After overnight digestion, trypsin was heat inactivated at 95 °C

for 10 min. Prior to adding the digested peptide to the streptavidin-agarose resin containing spin columns, the columns (Pierce, Catalog no. 69705) were first washed with PBS two times, followed by adding 0.400 mL streptavidin-agarose resin (Thermo Scientific, #20347) to the column and washing the resin 1x with PBS. The resin was then equilibrated with a 1:7 denaturation buffer: dilution buffer wash four times prior to plugging the outlet of the column and incubated at 4 °C with continuous rotation to mix for 18 h. After the overnight incubation, the protein resin mix in the columns was washed six times in 0.85 mL of wash buffer 1 (30 mM Hepes, 1 mM EDTA, 150 mM NaCl, 0.5% Triton X-100, 0.1% SDS, pH 7.5) followed by another six washes with 0.85 mL of wash buffer 2 (30 mM Hepes, 1 mM EDTA, 600 mM NaCl, 0.5% Triton X-100, 0.1% SDS, pH 7.5) and finally another three washes with 0.85 mL of wash buffer 3 (30 mM Hepes, 1 mM EDTA, 100 mM NaCl, pH 7.5). All washes were done using the Vac-Man Laboratory Vacuum Manifold (Promega A7231). Resin with bound peptide was eluted with or without DTT as described above in the protein level BTA, with the exception that no trypsin was added during the gel processing steps as the proteins were predigested prior to streptavidin capture.

**Peptide level iodoTMT-BTA**. Seven milligrams of proteins were extracted and alkylated with 343 μM Maleimide-PEG2- biotin as described above in the protein level BTA assay. Alkylated proteins were precipitated with ice-cold acetone, resuspended in denaturation buffer (8 M urea, 1 mM MgSO4 and 30 mM Tris–HCl, pH 7.5). After denaturation for 10 min at 90 °C proteins were diluted with seven volumes of dilution buffer (1 mM CaCl2, 100 mM NaCl, and 30 mM HEPES-NaOH pH 7.5), followed by incubating with trypsin (Promega, #V5111) (ratio of 1:50 W/W) overnight at 37 °C. After overnight digestion, trypsin was heat inactivated at 95 °C for 10 min. Prior to adding the digested peptide to the streptavidin-agarose resin containing spin columns, the columns (Pierce, Catalog no. 69705) were first washed with PBS two times, followed by adding 0.400 mL streptavidin-agarose resin (Thermo Scientific, #20347) to the column and washing the resin 1x with PBS. The resin was then equilibrated with a 1:7 denaturation buffer: dilution buffer washes four times prior to plugging the outlet of the column and incubated at 4 °C with continuous rotation for better mixing for 18 h. After the overnight incubation, the protein resin mix in the columns was washed six times in 0.85 mL of wash buffer 1 (30 mM Hepes, 1 mM EDTA, 150 mM NaCl, 0.5% Triton X-100, 0.1% SDS, pH 7.5) followed by another six washes with 0.85 mL of wash buffer 2 (30 mM Hepes, 1 mM EDTA, 600 mM NaCl, 0.5% Triton X-100, 0.1% SDS, pH 7.5) and finally another three washes with 0.85 mL of wash buffer 3 (30 mM Hepes, 1 mM EDTA, 100 mM NaCl, pH 7.5). All washes were done using the Vac-Man Laboratory Vacuum Manifold (Promega A7231). Resin with bound peptide was first incubated with freshly prepared 500 μL elution buffer (30 mM Hepes, 1 mM EDTA, 100 mM NaCl, 10 mM TCEP, 10 mM TEAB, pH 7.5) for 30 min at 25 °C followed by centrifugation of the spin tube inside of a 2 mL collection tube at 2000 g for 2 min to collect the eluate. A C18 column (Thermo #89870) was used to remove TCEP followed by eluting the peptides with 70% ACN. Eluted peptides were dried and suspended with 100 μl of 50 mM TEAB buffer. Thiol groups were alkylated and labeled with 2.95 mM iodoTMT (Thermo Scientific #90101) for 1 h in the dark. Reactions were then quenched by vortexing and incubating for 15 min at 37 °C in 20 mM DTT in the dark. Each of the six samples labeled with a unique iodoTMT tag was then mixed into a single tube and dried. The iodoTMT tags were iodoTMT[6]-126, iodoTMT[6]-127, iodoTMT[6]-128, iodoTMT[6]-129, iodoTMT[6]-130, and iodoTMT[6]-131. Finally, the combined labeled peptides were reconstituted with 1% Acetic Acid for further analyses via LC/MS.

**Gel electrophoresis and in-gel digestion**. 1.5 mm thick sodium dodecyl sulfate (SDS)-12% polyacrylamide gels were used to separate and visualize purified persulfidated proteins via staining or Western Blot. 0.75 mm thick one-dimensional (SDS)-12% polyacrylamide gels were used to separate persulfidated proteins for in-gel digestion and downstream HPLC-MS/MS analysis. For both of these processes, 11 μL of proteins from input, eluted –DTT samples, and +DTT samples, were mixed and denatured by boiling at 100 °C with 2 μL of 5x Laemmli loading buffer (Fisher Scientific, Catalog no. 39001) and loaded into each lane, with 7 μL of PageRuler Plus Prestained (Thermo Fisher #26619) protein ladder in the first lane. Gels were run at 135 volts for 2 h for imaging only and Western blot endpoints, and for 13–15 min for gels purposed for obtaining persulfidated proteins for in-gel digestion and HPLC-MS/MS analysis. Gels for imaging only were washed with ddH2O for 10 min, protein bands stained with colloidal blue staining kit (Thermo Fisher, #LC6025) for 3 h at room temperature, and followed by another 3 h of washing with ddH2O. Gels were then scanned (Epson Perfection V500 scanner) to obtain images. Proteins in gels used for Western Blotting were then transferred to polyvinylidene difluoride membranes (Whatman), blotted for GAPDH (Abcam #ab8245, 1:1000 in 5% milk) followed by horseradish peroxidase-conjugated goat anti-mouse secondary antibody (Invitrogen #62-6520, diluted 1:5000 in 5% milk) or α-tubulin (Abcam #ab4074, 1:1000 in 5% milk) followed by horseradish peroxidase-conjugated goat anti-rabbit IgG secondary antibody (Abcam #97051, 1:5000 in 5% milk) and visualized using SuperSignal West Femto Maximum Sensitivity Substrate (Thermo Scientific #34096) on an Amersham Imager 600 (General Electric). Gels used for in-gel digestion and downstream HPLC-MS/MS analysis were first cut for size and then washed for half an hour with ddH2O at room temperature. Protein bands were visualized by staining with gel code blue

stain reagent (Thermo Fisher scientific, #24590) for 1 h followed by washing with ddH₂O for one hour at room temperature. Gels were scanned by using the Epson Scanner. A scalpel was used to excise the entire band, and then the band was further cut into smaller bands of ~1 mm³. All of the smaller gel pieces from the original band were transferred to 0.2 mL of wash buffer (50% ethanol and 5% acetic acid) in a 1.5 mL tube and washed overnight at room temperature. The following day gel pieces were dehydrated with acetonitrile and dried in a SpeedVac for 6 min. Proteins in the dried bands were reduced with 0.1 mL of 65 mM dithiothreitol (DTT) (Fisher Scientific # BP172–5) at room temperature for 30 min, followed by alkylation for another 30 min at room temperature with 0.1 mL of 162 mM iodoacetamide (Fisher Scientific, #AC122270050). Gel pieces were dehydrated two times with 0.2 mL acetonitrile and rehydration with 0.2 mL of 100 mM ammonium bicarbonate. Then, 40 μL of 10 ng/μL trypsin (Promega, #V5111) was added into the dried gel pieces with an additional 20 μL of 50 mM ammonium bicarbonate added to completely cover the gel pieces and kept overnight at room temperature for tryptic digestion. 0.08 mL of extraction buffer (50% acetonitrile + 5% formic acid) was added twice for 10 min at room temperature each time to extract the digested peptides from the gel pieces followed by micro centrifuging for 10 s. Supernatants were combined and transferred into a new 0.5 mL centrifuge tube and dried at room temperature in a Speedvac for 3 h. Thirty microliters of 1% acetic acid was added into the dried peptides and transferred into an HPLC vial w/cap (Sun-Sri, #200 050 & 501 313) for further mass spectrometric analysis.

**HPLC-tandem mass spectrometric analysis.** Digested peptides were analyzed on a ThermoFisher Scientific UltiMate 3000 HPLC system (ThermoFisher Scientific, Bremen, Germany) interfaced with a ThermoFisher Scientific Orbitrap Fusion Lumos Tribrid mass spectrometer (Thermo Scientific, Bremen, Germany). Liquid chromatography was performed prior to MS/MS analysis for peptide separation. The HPLC column used is a Dionex 15 cm × 75 μm Acclaim Pepmap C18, 2 μm, 100 Å reversed-phase capillary chromatography column. Five microliter volumes of the peptide extract were injected and peptides eluted from the column by a 90-min acetonitrile/0.1% formic acid gradient at a flow rate of 0.30 μL/min and introduced to the source of the mass spectrometer on-line. Nano electrospray ion source was operated at 2.3 kV. Tissue extracted non-labeled protein digests were analyzed using the data-dependent multitask capability of the instrument acquiring full scan mass spectra using a Fourier Transform (FT) in the Orbitrap Fusion Lumos Tribrid mass spectrometer (Thermo Scientific, Bremen, Germany) to determine peptide molecular weights and collision-induced dissociation (CID) MS/MS product ion spectra with an ion-trap analyzer to further determine the amino acid sequence in successive instrument scans. The MS method used in this study was a data-dependent acquisition (DDA) with 3 s duty cycle. It includes one full scan at a resolution of 120,000 followed by as many MS/MS scans as possible on the most abundant ions in that full scan. Dynamic exclusion was enabled with a repeat count of 1 and ions within 10 ppm of the fragmented mass were excluded for 60 s. Data using the iodoTMT 6-plex labeled peptide samples were also collected using a TMT-MS2 method in the same Orbitrap Fusion Lumos Tribrid mass spectrometer (Thermo Scientific, Bremen, Germany). This was a data dependent acquisition method using higher-energy collisional dissociation (HCD) as the fragmentation method for MS/MS scans. It included one full scan followed by as many MS/MS scans as possible at collision energy 38 in a 3 s duty cycle on the most abundant ions in that full scan. The detector of MS full scans was the Orbitrap at a resolution of 120,000, and the detector of MS/MS scans was the Orbitrap at a resolution of 50,000. Dynamic exclusion was enabled with a repeat count of 1, and ions within 10 ppm of the fragmented mass were excluded for 60 s. Instrumental quality performance, aka quality control 1 (QC1), was periodically monitored via Pierce Retention Time Calibration Mixture (Pierce #88320) (Supplementary Data File 39, with raw data uploaded to Figshare at https://figshare.com/s/941a9d9bc9cd39b0dac6 and https://figshare.com/s/f613f80460bce4eb722d).

**Bioinformatics for peptide identification and quantification.** Mascot, Sequest, and Scaffold software packages were used for label-free quantitative and qualitative proteomics analysis. These software tools were used for protein identification by converting raw spectrometric data into protein IDs and for relative quantification of protein abundance via label-free spectral counting. For protein identification, three search engines were used including Mascot, Sequest which is bundled into Proteome Discoverer 1.4, and X!Tandem which is bundled into Scaffold 4.8.7. For the Mascot searches, primary raw MS/MS data were converted into its MGF (Mascot Generic File) format files by using Discoverer Daemon 1.4 software (Licensed under Cleveland Clinic Proteomics Core). The parameters selected for these searches include the following: Database: SwissProt (https://www.ebi.ac.uk/uniprot/), Taxonomy: Mus Musculus, Enzyme: Trypsin, Fixed modification: Carbamidomethyl(C), Variable modification: Oxidation of Methionine, Precursor Mass Tolerance: 10 ppm, Fragment Mass Tolerance: 0.8 Da, Peptide charge: 2+, 3+, and 4+, with the monoisotopic and decoy database present. Scaffold software (version Scaffold_4.8.7, Proteome Software Inc, Portland, OR)[110] was used to validate peptide and protein identifications and create comprehensive lists of target proteins. Protein and peptide identifications were validated using a decoy database strategy (reverse sequence of each protein for use as a decoy) and the decoy rate was set to zero, which may limit the total number of proteins identified but enables a higher confidence in protein identity for those that do meet all stringent criteria.

Other settings for Scaffold included only allowing positive IDs if they could be established at greater than 99.9% probability and contained at least two identified peptides by the Peptide Prophet algorithm[111,112] with Scaffold delta-mass correction. Proteins that contained similar peptides and could not be differentiated based on MS/MS analysis alone were grouped to satisfy the principles of parsimony. Additionally, to remove any false positive hits, we individually analyzed all the proteins identified to ensure each one contained at least one cysteine residue by using the mouse protein amino acid sequences from UniProt Knowledgebase protein database (Proteome_ID/Tax_ID: UP000000589/10090 [https://www.uniprot.org/proteomes/UP000000589]). Those that did not contain at least one cysteine residue, and thus could not be theoretically be persulfidated (~2–4% of our initial findings), were removed from our database and from further analysis. Peptide identifications were also required to exceed specific database search engine thresholds, including Mascot identifications required at least 40 ion score, Sequest identifications required at least XCorr (+1) is 1.5, XCorr (+2) is 2, XCorr (+3) is 2.25, and XCorr(+4) is 2.5, and X!Tandem identifications required at least 2.

Label-free spectral counting was used to determine relative differences in persulfidated protein abundance between AL versus DR fed groups within a specific tissue[57]. The spectral counts, defined as the total number of spectra identified for a specific protein[113], have integer values ranging from 0 for proteins below the level of detection up to 753. Theoretically, higher spectral counts for a given protein equates to higher abundance of that protein in a given tissue sample, where the number of spectra matched to peptides from a specific protein is used as a surrogate measure of protein abundance[114]. The average spectral counts for each protein in each tissue were calculated, and the ratio of average spectral counts amongst the control group (e.g., AL, WT) compared to the experimental group (e.g., DR, EOD, KO) were used to determine fold-change between the two groups. Spectral count values below the level of detection were originally automatically assigned a value of 0, however, these were changed to 0.1 for purposes of statistical analysis and plotting in log scale in volcano plots.

We also used MaxQuant version 1.6.14.0 with Perseus version 1.6.12.0 for further analysis and validation of raw data files for label-free intensity-based quantitation rather than spectral counting with the following settings. For variable modifications, we allowed acetyl (protein N-term) and methionine oxidation. As a fixed modification we selected carbamidomethylation on cysteine residues as all samples were treated with iodoacetamide and a maximum number of modifications allowed were 5. For the digestion mode, we used the enzymatic rule of trypsin/P with a maximum of two missed cleavages. We used Maxquant standard settings for instrument type setup. We used label-free quantification with the minimum ratio count of 2 and the normalization was set to none. We used the 17,053 reviewed proteins present in the mouse proteome downloaded from Uniprot on April 13, 2019, as a search FASTA file (https://www.ebi.ac.uk/uniprot/). We selected minimum peptide length for unspecified search as 8. Protein FDR was set at 0.01, minimum peptide length as 6 and maximum peptide mass as 5000 Da with minimum peptides were set as 2 and minimum unique peptides set as 1. For protein quantification we used unique and razor peptides and allowed oxidation (M) and acetyl (protein N-term) modifications. Intensity values below the level of detection or given a value of 0 were changed to 10,000 for purposes of statistical analysis and plotting in log scale in volcano plots.

MS raw data files of iodoTMT labeled peptides were analyzed by Proteome Discoverer version 2.4.1.15 (Thermo Fisher Scientific) with the following settings. We used *Mus musculus* in SwissProt version 2019 (Swiss Institute of Bioinformatics, Switzerland) FASTA file. For quantification method, we used iodoTMT6plex with the TMT reporter ion isotope distribution (lot number VH308730). For processing setup, we used integration tolerance as 20 ppm and integration method set to the most confident centroid. FTMS mass analyzer was used, and we selected MS1 precursor for the precursor selection section. For the enzyme name, we used trypsin (full) with a maximum missed cleavage site of 2. Precursor mass tolerance was set to 10 ppm and fragment mass tolerance was set to 0.2 Da. For dynamic modification, we allowed methionine oxidation (+15.995 Da). We selected static modification as iodoTMT6plex/+329.227 Da of cysteine. For consensus setup, we used intensity as reporter abundance. Validation mode was set to automatic (control peptide level error rate if possible). Target FDR (strict) for PSMs and target FDR (strict) for peptide were set at 0.01, while Target FDR (relaxed) for PSMs and target FDR (relaxed) for peptide set at 0.05. Peptide confidence was set to high with minimum peptide length allowed to 6. For confidence threshold, we used target FDR (strict) and target FDR (relaxed) at 0.01 and 0.05, respectively. Minimum PSM confidence level was set to high.

Raw proteomics-related data have been uploaded to the ProteomeXchange Consortium via the PRIDE partner repository with the following accession codes PXD022888 (Proteome Discoverer 2.4 based analysis), PXD022956 (MaxQuant 1.6.14.0 based analysis), and PXD022954 (Scaffold 4.8.7 based analysis).

**Biological function and pathway enrichment.** The online-based web server g:Profiler (https://biit.cs.ut.ee/gprofiler/gost)[59] was used for functional enrichment analysis of identified persulfidated proteins. Gene ID's or accession numbers of enriched proteins were used in the g:GOSt (Gene Group Functional Profiling) identifier tool to detect biological pathways from the KEGG database[60] (https://www.genome.jp/kegg/) significantly enriched with persulfidated proteins.

These significance values are given a threshold of $p < 0.05$ and were auto-calculated by the software's proprietary built-in g:SCS algorithm (https://biit.cs.ut.ee/gprofiler/page/docs) that utilizes multiple testing correction for $P$ values gained from pathway enrichment analysis. In setting the boundaries for significance determination, it corresponds to an experiment-wide threshold of $a = 0.05$, with the g:SCS threshold pre-calculated for list sizes up to 1000 accession number or gene ID terms and analytically approximates a threshold $t$ corresponding to the 5% upper quantile of randomly generated queries of that size. As per the g:Profiler description, all actual $P$ values resulting from the query are automatically transformed to corrected $p$ values by multiplying these to the ratio of the approximate threshold $t$ and the initial experiment-wide threshold $a = 0.05$ with consideration to the underlying gene sets annotated to terms from the KEGG database, and, therefore, gives a tighter threshold to significant results and is more conservative than the Benjamini–Hochberg False Discovery Rate (FDR)[59].

**Statistical analysis**. Statistical significance and data display was generated in Microsoft Excel (version 2013), GraphPad Prism (version 5), Origin (version 2018), and Venny 2.1.0 software. The differences between two diet groups within the same genotype in regards to $H_2S$ production or spectral counts for individual persulfidated proteins were analyzed by two-sided Student's $t$ test, with a level of significance set to $P < 0.05$. N values for the number of individual mice used are as follows (unless noted in the figure legend): (1) 6-month old CGL WT and KO mice: 4 mice for WT AL, 5 mice for WT DR, 3 mice for KO AL, and 3 mice for KO DR, (2) 12-month-old CGL WT and KO mice: 3 mice for WT DR, and 3 mice for KO DR, and (3) 20-month C57BL/6 mice: 5 mice for AL, and 5 mice for EOD fasting. Data were arranged in Microsoft Excel to show both relative fold-changes in average spectral counts for an individual protein, as well as the two-sided $t$ test $P$ value, and these data are found in the Supplementary Data files. Bar-graph data in Fig. 1 are displayed as means +/−SEM with $n$ values between 3 and 5 as indicated in the figure legend. Analysis of recombinant CGL $H_2S$ production was by one-way ANOVA with Dunnet's Multiple Comparison Test against the +CGL w/0 NaHS group after subtracting against the −CGL spot corresponding for that given dose of NaHS. Data are plotted as individual points with the mean and SEM shown. Origin software version 2018 (https://www.originlab.com/) was used to generate volcano plots to display differentially abundant persulfidated proteins in tissues from AL versus DR fed mice. Venny 2.1.0 software (https://bioinfogp.cnb.csic.es/tools/venny/) was used to generate Venn diagrams and determine differences and commonalities in persulfidated proteins between tissues, diets, and genotypes. Statistical analysis for pathway enrichment via g:SCS algorithm is described in the above section regarding the g:Profiler online functional enrichment analysis tool. The individual calculated/adjusted $P$ values and identities of proteins for each enriched pathway generated by g:Profiler is found in the Supplementary Data files. All measurements in this study were taken from distinct individual biological samples/animals and not the same sample measured repeatedly nor multiple samples taken from the same animal.

**Reporting summary**. Further information on research design is available in the Nature Research Reporting Summary linked to this article.

## Data availability

The authors declare the data supporting the findings of this study are available within the paper and the Supplemental Information file (containing Supplementary Figs. 1–9) and in the Supplementary Data files 1–39. Source data are provided with this paper. Furthermore, raw proteomics-related data have been uploaded to the ProteomeXchange Consortium via the PRIDE partner repository with the following dataset identifiers PXD022888 (Proteome Discoverer 2.4 based analysis), PXD022956 (MaxQuant 1.6.14.0 based analysis), and PXD022954 (Scaffold 4.8.7 based analysis). Proteomic instrumental quality control data (QC1) are available at Figshare (https://figshare.com/s/941a9d9bc9cd39b0dac6 and https://figshare.com/s/f613f80460bce4eb722d) and are summarized in Supplementary Data 39. RNA expression graphs of $H_2S$ producing and consuming proteins were generated utilizing the Mouse ENCODE project[55] with values extracted from the publically availably NCBI Mouse Gene Database at the following NCBI webpages: CGL (https://www.ncbi.nlm.nih.gov/gene/107869), CBS (https://www.ncbi.nlm.nih.gov/gene/12411), 3-MST (https://www.ncbi.nlm.nih.gov/gene/246221), and SQR (https://www.ncbi.nlm.nih.gov/gene/59010). SwissProt/UniProt data used in mass spectrometry analysis software tools is from the publically available UniProt database (https://www.uniprot.org/). Questions and requests for resources and data should be directed to and will be fulfilled by the corresponding author. Source data are provided with this paper.

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

## Acknowledgements

We thank Drs. Xing-Huang Gao and Maria Hatzoglou of Case Western Reserve University for technical insights regarding the BTA, and Paul Minkler from the Proteomics and Metabolomics Core at Cleveland Clinic for assistance in proteomics and LC-MS/MS analysis. This work was funded by NIH grants R00AG050777 and R01HL148352 to C.H., an NIH shared instrument grant 1S10OD023436-01 to B.W., an AFAR/Glenn Foundation Fellowship to Y.O.H., and a Discovery Grant from the Natural Sciences and Engineering Research Council of Canada to R.W.

## Author contributions

N.B. and C.H. conceptualized the project. N.B., Y.O.H., S.K., B.W., and C.H. designed experiments. N.B., C.L., Y.O.H., S.K., J.Y., L.L., B.W., and C.H. performed experiments and analyzed data. R.W. contributed valuable CGL KO mouse models. N.B. and C.H. wrote the paper with all authors editing and approving the manuscript.

## Competing interests

The authors declare no competing interests.
