## [Peer Review File · Nature Communications]

Reviewer #1 Comments to the author:

The present manuscript took as a starting point of investigation of the association between DR and H₂S production in CGL WT and KO mice. As a result, they found that DR and genetic manipulation of CGL levels affect H₂S production in a tissue specific manner. Then, they adapted a previously described proteomic approach (Gao, et al., Elife. 2015) to generate the first mouse tissue 'sulfhydrome' atlas perturbed with dietary restriction using the same mice models. Overall, the work is interesting and the experiments for such a large-scale proteomic analysis are properly designed. However, the scientific accuracy, as the work currently stands, may not meet high standards for this journal. In order to elevate this work to the level of Nat Commun, I would suggest that the authors consider and address the following points.

H₂S measurement

- Although lead acetate is a simple and widely used way for measuring H₂S production, it provides only semiquantitative data (ImageJ data in this case), and the sensitivity is quite low. Also, it seemed that the authors did not use a blank solution (e.g. PBS buffer made for the assay) for background correction. In terms of this, the results shown in Figure 1 is not very trustworthy.
- There is a lack of correlation between the H₂S production and SSH formation affected by DR. Specifically, a number of persulfidated proteins were enriched in brain and muscle under DR (Figure 2D), whereas DR had little impact on H₂S production in brain and muscle (Figure 1 G and H). The authors provide an explanation for such unexpected findings in the Discussion part. However, in line with the point above, this could be simply caused by inaccurate measurements.

Chemical labeling

- It is totally fine to apply NEM-biotin as the alkylating reagent (this warhead can also react with redox modifications, but this limitation can be avoided as the authors mentioned in the discussion part). The concentration of NEM-biotin used in this study was only 343 μM to achieve relatively selectivity of protein persulfides. However, if protein thiols cannot be fully alkylated or blocked, a large number of artifactual disulfide bonds might be formed across the proteome during sample handling. One can imagine the case shown in the following scheme, in which a non-'sulfhydrated' protein B could also be co-captured with a 'sulfhydrated' protein A using the BTA approach. The authors also discuss such a methodological limitation but claimed that false positive caused by such a limitation may be minor. Can they provide any experimental evidence to support this claim?

Identification and quantification of 'sulfdydrated' proteins

- The SSH profiling method used in this study was adapted from a previous report by Gao and colleagues. In a technical aspect, therefore, the novelty of this work was compromised. Moreover, as compared to its original version that enabled a site-level analysis, the present work by Bithi *et al.*, failed to do so.
- Though out of "fashion", DDA-based label-free quantification is still widely adopted in the field of proteomics, especially for measuring changes in protein expression across a number of biological samples. As compared to a routine/standard proteomic experiment, the BTA approach involves many more sample processing steps. It seemed to be extremely challenging, if not impossible, for such a large-scale study to obtain reproducible label-free quantification results. In principle, label free quantification based on MS1 intensity would be a better choice for DDA-based quantification. Why did the authors choose spectral counting strategy instead? If the former was not feasible to the authors, they should at least validate their quantification results using an orthogonal approach and/or demonstrate their instrumental performance by showing QC data obtained during the period of this project.

Bioinformatics

- In many cases, less than 10 hits could be enriched in individual KEGG pathways, posing a concern about the 'real' significance of statistics. In this regard, the calculated p-values might be subjected to multiple testing correction (Noble WS. *Nat Biotechnol.* 2009). Alternatively, the FDR rates for the KEGG enrichment analyses could be provided.

Minors:

- Indeed, sulfhydrylation is most widely used in literatures. However, in a chemical aspect, this term implies "hydration" and is inaccurate (Filipovic et al., *Chem Rev* 2018, 118: 1253-1337). The term "persulfidation" has been gaining wide acceptance (Bestetti et al., *Sci Adv* 2018, 4: eaar5770).
- Page 6, the authors claimed that "ex vivo addition of NaHS/H₂S may act as a reducing equivalent, similar to DTT, and remove the in vivo formed sulfhydrylation modifications prior to the BTA assay". This can be misleading as the effects of NaHS on the formation of persulfidation is believed to be dependent on its concentration. The authors should specify the experimental conditions.
- Page 12, "As NaHS itself gives an immediate positive lead sulfide result as it quickly releases H₂S upon placement in aqueous solutions" This is incorrect, as NaHS dissolves to give the HS⁻ anion rather than H₂S (HS⁻ is the predominant form of H₂S at physiological pH). Also, "NaHS is a H₂S donor" is a longstanding but wrong notion.
- Page 28, DIA should be changed to DDA.
- There are too many supplementary tables which are not easy to follow.
- Supplemental Data File 1 is not very informative, and should be removed.

Reviewer #2 (Remarks to the Author):

The study by Bithi et al, deals with the effects of dietary restriction on the sulfhydryl in various tissues. While the authors have generated several datasets pertaining to the sulfhydryl in different tissues, there are several concerns which cannot be set aside.

As the authors themselves acknowledge, there are some serious issues with the method utilized to detect persulfidation. It is well established that sulfenic acids (RSOH) readily react with thiol blocking reagents such as N-ethylmaleimide or iodoacetamide, giving a DTT cleavable adduct (Reisz et al., FEBS J, 2013). On the other hand, Michael-type addition product formed in the reaction of normal thiolate with maleimides could also easily undergo thiol exchange with DTT (particularly when present in large amounts) via retro Michael-type addition even under the conditions used by the authors (Fontaine et al., Bioconjug. Chem, 2015; Wu et al., Bioconjug. Chem, 2018). Finally, it is easy to imagine that biotinylated peptide could also be connected via disulfide bond to another peptide that will be released with DTT treatment. Thus, the approach not only measures R-SSH but also RSOH and RSSR, in addition to a high basal nonspecific release due to retro Michael-type thiol exchange, hence authors cannot claim that they measure changes in sulfhydrylation.

Besides, the authors do not perform the real quantitative proteomic approach. 1.25-fold increase is usually impossible to reach as significant even with approaches such as SILAC, and certainly not with $n=3$. The authors often compare the number of detected proteins. This is not a correct method for analysis, as differences in the number of identified peptides/proteins cannot be extrapolated to the level of modification. They could as well be a consequence of differential extraction efficiency from different tissues. The bioinformatics part is very basic. The cysteine proteome is well defined with the recent findings bringing the bioinformatics analysis to a much higher level (Xiao et al, Cell, 2020).

In addition to these major concerns, the method to detect H₂S is not sensitive or quantitative enough. The authors mention that not enough H₂S is produced in the brain upon dietary restriction, yet the enrichment of the sulfhydrylated proteins is among the highest there. Obviously, the liver produces higher H₂S in comparison to the brain, but the method (lead acetate-based method) is clearly not sensitive or quantitative enough.

A direct correlation between sulfhydrylation and CSE levels has been recently demonstrated in striatal mouse cell lines and rat brain (Zivanovic et al, Cell Metab, 2019) so there should be no surprise why the authors see the highest increase there. Neurons predominantly express CSE (Sbodio et al, Brit J Pharmacol, 2018).

Besides, the study does not provide novel conceptual advances over what is already known in literature. This study provides a large list of proteins persulfidated in response to dietary restriction using a method which is not rigorous enough, which does not significantly advance the field to warrant publication in the journal.

Reviewer #3 (Remarks to the Author):

This is a well-written paper describing experiments that further our understanding of the mechanisms underlying the effects of dietary restriction on health and longevity. The authors describe a unique approach that examines the sulfhydromes of various tissues in response to dietary restriction as well as a lack of a critical enzyme, cystathionase (gamma lyase). The results strengthen the idea that methionine metabolism plays a major role in the potential effects of dietary restriction on longevity. The data is novel and exciting especially to those that study the role of diet in health. The methods and analysis are thoughtful and thorough in terms of techniques to measure H₂S and sulfhydration.

The authors suggest that that H₂S signaling has only been recently recognized as proposed mechanism in lifespan extension. However, several papers published 15-20 years ago (that were not acknowledged) demonstrated that an upregulation of multiple components of the methionine metabolic pathway (that includes CGL and from which H₂S is derived) was likely a major contributor to enhanced stress resistance and longevity of long-living mice (Ames dwarf and Growth hormone receptor knockout mice) while suppressed methionine metabolism contributed to shortened lifespans exhibited by growth hormone transgenic mice. The point being that there were discoveries quite a while ago suggesting the importance of this pathway in longevity. Also, numerous reports indicate that 50% of the cysteine generated via the transsulfuration pathway is utilized in glutathione synthesis while a significant amount is metabolized into taurine. Thus, it has not yet been demonstrated that H₂S is the primary factor responsible for the beneficial effects of this pathway on health and longevity but it is indeed a contributor and warrants continued research. This paper is a productive step in that direction.

Transsulfuration operates in liver, kidney and brain thus it makes sense that these tissues were largely impacted by both DR and in the CGL mice.

In terms of experimental design, it's curious as to why they chose a one-week dietary restriction protocol to examine the changes to the sulfhydromes of various tissues. The impact of DR on many parameters that impact health and longevity adapt more slowly to the effects of reduced calories. Collection of tissues after two months of DR (a minimum standard in the aging field) would likely

have shown a much more dramatic and perhaps even a different picture than that obtained after one week. One other question not adequately described in the methods is how was food intake measured (individual or groups of mice?). Five grams of food per day is on the higher end. Many studies have indicated 3-4 grams of food are consumed per day. What was the methionine and cysteine content of the diets? There are many studies indicating that reduced methionine content of the diet extends health and lifespan. How do the authors address a 50% reduction in methionine in this study?

The statistics for the H₂S production and similar assays are reasonable.

Minor: there are two files labeled Supplemental Table 1 – should be liver and table 2 should be kidney.

Response to Reviewers' Comments NCOMMS-20-07544A

Dear Editor and Reviewers:

This document presents our full response to the Reviewers' Comments for the manuscript titled “**Dietary restriction transforms the mammalian protein sulfhydrylome in a tissue-specific and cystathionine γ -lyase-dependent manner**” (NCOMMS-20-07544A) by Nazmin Bithi, Christopher Link, Yoko O. Henderson, Suzie Kim, Jie Yang, Rui Wang, Belinda Willard, and Christopher Hine. We highly appreciate the feedback and constructive criticism from all three Reviewers, and we are very thankful to the Reviewers for carefully reading our manuscript. In this new revised manuscript, we have addressed to the best of our ability the biological, technical, and semantic issues raised by each Reviewer by the addition of new experimental animal model data, new technical proteomics approaches, new data analysis, and clarifications in the text. These new additions to the manuscript based on the reviewers' comments are provided in **BLUE** font in the text and figure legends. They are also described/highlighted in **BLUE** text in our response to the Reviews below.

(A) Response to Reviewer #1

A.1 Reviewer #1: The present manuscript took as a starting point of investigation of the association between DR and H₂S production in CGL WT and KO mice. As a result, they found that DR and genetic manipulation of CGL levels affect H₂S production in a tissue specific manner. Then, they adapted a previously described proteomic approach (Gao, et al., *Elife*. 2015) to generate the first mouse tissue ‘sulfhydrylome’ atlas perturbed with dietary restriction using the same mice models. Overall, the work is interesting and the experiments for such a large-scale proteomic analysis are properly designed. However, the scientific accuracy, as the work currently stands, may not meet high standards for this journal. In order to elevate this work to the level of *Nat Commun*, I would suggest that the authors consider and address the following points.

Our response: We thank Reviewer #1 for recognizing the novelty, design, and potential of this manuscript. We also thank Reviewer #1 for their constructive criticism of our work, in particular to the scientific accuracy, and for their suggestions and guidance to improve it. Have taken into account these suggestions and implemented new experimental models and approaches into the revised manuscript (in Blue text in the manuscript and figure legends) to elevate it to the level of *Nature Communications*. These are also outlined in our responses below.

A.2 Reviewer #1: Although lead acetate is a simple and widely used way for measuring H₂S production, it provides only semiquantitative data (ImageJ data in this case), and the sensitivity is quite low. Also, it seemed that the authors did not use a blank solution (e.g. PBS buffer made for the assay) for background correction. In terms of this, the results shown in Figure 1 is not very trustworthy.

Our response: The comments by Reviewer #1 are true in that the lead acetate method for measuring H₂S production gives only semiquantitative data and not absolute in terms of ppm of the H₂S gas in a sample. However, semiquantitative data generated like this and quantified via ImageJ for IntDen is very similar to other techniques widely used and accepted in biological research, such as Western Blots and DNA gels, for examining relative differences between a control group and an experimental groups. Our approach is optimal for measuring the relative differences in H₂S production capacity amongst individual samples and different groups all at one time (like what is shown in Figure 1), as equal exposure time of the lead acetate paper to the headspace of all the wells simultaneously is ensured (much like all bands in a Western Blot are equally exposed to a film when it is overlays over all bands at once). Again, while we agree that there are limitations to the lead acetate/lead sulfide assay in terms of obtaining absolute quantification, it is still widely used and accepted as a method that has selectivity and sensitivity for H₂S in the headspace/gas phase¹⁻⁹, and was recently used this year as the primary method to measure H₂S production capacity in liver derived cells in this same journal by Xu, Qing; et al. “HNF4 α regulates sulfur amino acid metabolism and confers sensitivity to methionine restriction in liver cancer” *Nature Communications* 11, 3978 (2020)¹⁰. Additionally, highlighting the sensitivity and selectivity of the lead acetate/lead sulfide methods, we previously published (Yang, J. et al. *Communications Biology* 2, 194 (2019))¹¹

that use of selected ion flow tube mass spectrometry (SIFT-MS)^{12,13} which is a quantitative, selective, and sensitive headspace analysis tool that can be used for H₂S measurement, validated our lead acetate/lead sulfide data and confirmed it is H₂S and not other volatile/reactive sulfide species generated from the reaction present in the gas phase detected in the lead acetate/lead sulfide method. However, we did not utilize SIFT-MS in the present paper for the main reason that each sample must be analyzed individually, thus making it difficult to ensure equal incubation time of each H₂S production capacity reaction at 37degC prior to sampling. As there are 36 individual biological samples analyzed in Figure 1B and C, it becomes apparent that analyzing that many samples via SIFT-MS would be technically challenging and produce results that may not be accurate.

Nonetheless, in the current work addressing the sensitivity of the assay, we have addressed part of Reviewer 1's concern regarding the use of the blank solutions (PBS buffer w/ PLP and L-cysteine) by now showing these reaction mixture wells/dots in:

A) The wells/dots labeled as (-) Tissue in **Figure 1B and C** and in the **Supplemental Data File 1: Uncropped Source Data for Figure 1**.

B) The wells/dots labeled as (-) CGL in **Figure 6, Supplemental Figure 6**, and in the **Supplemental Data File 1: Uncropped source Data for Figure 6**.

These reaction mixture wells were always taken into account even in our first version of the manuscript and data obtained from wells with tissues were normalized to these blank reaction mixture only wells. But at the request of Reviewer #1, we have now shown these background wells/spots and made it clearer in the Materials and Methods section and in the Figure legends that these reaction only background values were subtracted from the experimental sample values when plotting the graphs. In doing so, it is now more apparent that the H₂S production identified from tissues is above background, thus ensuring the sensitivity of the assay to detect H₂S production even in low producing tissues. Additionally, it should be highlighted that in Figure 6A and B, that our lead acetate/lead sulfide method is sensitive enough to detect a linear increase in H₂S production as a function of increasing CGL concentration (0, 1, 2.5, 5, and 10 µg per reaction) after a 2 hr incubation at 37°C. This data from updated **Figure 6A & B** supporting the sensitivity of the lead acetate/lead sulfide assay is shown below:

In addition to the *in vitro* assay utilizing purified CGL shown above, we previously published a study (Hine and Mitchell, Bio Protoc 7(13), 2017)¹⁴ that showed increasing liver protein lysate between 0 to 500 µg induces a linear H₂S production detection from the lead acetate/lead sulfide method up to 300 µg

[Redacted]

In our current assays, we utilized 100 µg of protein lysate/reaction, thus ensuring that we would stay in the linear range for detection. In summary, with the information and data provided here, we feel the experimental data obtained from the lead acetate/lead sulfide method is sensitive and appropriate for detecting relative changes in H₂S production capacity as a function of diet, genotype, and *in vitro* treatment.

A.3 Reviewer #1: There is a lack of correlation between the H₂S production and SSH formation affected by DR. Specifically, a number of persulfidated proteins were enriched in brain and muscle under DR (Figure 2D), whereas DR had little impact on H₂S production in brain and muscle (Figure 1 G and H). The authors provide an explanation for such unexpected findings in the Discussion part. However, in line with the point above, this could be simply caused by inaccurate measurements.

Our response: Reviewer #1 is correct in that there is a lack of correlation between H₂S production from weak H₂S producing organs such as muscle and brain with their enhanced persulfidation under DR. However, we do not think this is simply due to inaccurate measurements of H₂S production in the lead acetate/lead sulfide assay or its inability to detect relative differences induced by diet in these tissues. Again, as described in the response above, **we have now shown/described the reaction only wells/dots as background that is subtracted from the readings obtained from tissues**, and all values are above background for both ad libitum and DR feeding. Thus, H₂S production capacity does not necessarily have to increase with DR for enhanced persulfidation in select tissues, as we describe more in depth in the Discussion of this revised manuscript. We now hypothesize that persulfidation *in vivo* involves a complex series of modifications, events, and/or factors, and simply having more H₂S around does not necessarily mean increased persulfidation *in vivo*, as can be read in the updated Discussion.

For this reason and the new data shown below, we have now removed the data pertaining to basal ad libitum H₂S production capacity correlation with total numbers of persulfidated proteins in this revised manuscript. We have also added new data that brings to light the complexities of persulfidation *in vivo*. As seen in Figure 1B, CGL KO mice have decreased kidney H₂S production capacity under *ad libitum* feeding compared to WT mice. **However, in new data presented in Supplemental Figure 9D comparing head-to-head differences in protein persulfidation in these same kidneys using a new peptide-level biotin thiol assay combined with iodoTMT labeling, we observe there is not an enrichment for persulfidated proteins in WT mice, and there is slight enrichment in the KO kidneys under ad libitum feeding.** It is not until the mice are under dietary restriction the dependence for CGL for enhanced persulfidation is revealed, as shown in new Figure 9F. The data from these referenced figures from the revised manuscript are given below:

This suggests: 1) Basal persulfidation activity (at least in the kidney) under AL feeding or unstressed conditions is independent of CGL and/or CGL derived H₂S, and 2) Enhanced persulfidation in select tissues relies not only on CGL (or CBS, 3MST) derived H₂S but its coordination with other cysteine- and thiol- redox modifications induced by DR^{15,16} or other hormetic stressors^{17,18}. This is in line with the recent publication by Gao, *et al*¹⁸ highlighting a multi-step redox thiol switch from thiol glutathionylation to persulfidation.

[Redacted]

A.4 Reviewer #1: It is totally fine to apply NEM-biotin as the alkylating reagent (this warhead can also react with redox modifications, but this limitation can be avoided as the authors mentioned in the discussion part). The concentration of NEM-biotin used in this study was only 343 μ M to achieve relatively selectivity of protein persulfides. However, if protein thiols cannot be fully alkylated or blocked, a large number of artifactual disulfide bonds might be formed across the proteome during sample handling. One can imagine the case shown in the following scheme, in which a non-‘sulfdyated’ protein B could also be co-captured with a ‘sulfdyated’ protein A using the BTA approach. The authors also discuss such a methodological limitation but claimed that false positive caused by such a limitation may be minor. Can they provide any experimental evidence to support this claim?

Our response: We thank Reviewer #1 for agreeing that our use of NEM-biotin is appropriate as an alkylating agent, as well as their very helpful insights and suggestions for avoiding the caveats of protein level biotin thiol assay for detection of protein persulfides. **New experiments and data related to addressing these concerns are now found in the new Figure 9 and described here:** The Sulphydrome profiles generated thus far were performed on protein level pulldown/isolation and subsequent label-free quantification (LFQ). While suitable for protein persulfidation identification and showing relative shifts between groups, these methods are not without their drawbacks. Mainly, protein-level pulldown lends to potential false positive identification due to intermolecular disulfide bonds, thus allowing a non-persulfidated protein to be co-captured with a persulfidated protein and eluted during the final reducing step^{19,20}. Conversely, false negatives are possible due to NM-Biotin bound non-persulfidated intramolecular thiols (RS-NM-Biotin) on the same protein as a labeled persulfidated cysteine residue (RS-S_n-NM-Biotin) preventing final elution with a reducing agent²⁰. While these caveats are valid, it has been hypothesized that concentrations of disulfide bonds in intracellular proteins are low and not expected to be a major source of false positives in the approach utilized¹⁹. In the previous referenced paper by leaders in the redox biology/H₂S field (Filipovic, M.; Zivanovic, J.; Alvarez, B.; and Banerjee, R. Chemical Biology of H₂S Signaling through Persulfidation. Chemical Reviews, 2018, 118, 3, 1254-1337), they state “However, disulfides are not common in intracellular proteins, and the interference by disulfide-containing peptides/proteins in persulfide identification is expected to be low”, and later “concentrations of disulfide bonds in intracellular proteins is low and this is not expected to be a quantitatively major drawback of the method.” Nonetheless, we pursued an orthogonal approach as suggested by Reviewer 1 (**Figure 9A, Supplemental Figure 9A**) to avoid these limitations and further validate and expand our findings utilizing kidney and brain obtained from an independent cohort of 12-month old CGL WT and KO mice fed 1-week 50% DR.

Direct comparison between WT and KO kidney utilizing protein level BTA revealed the sulphydrome enriched in WT with 87% of all proteins having a higher LFQ intensity in WT over KO (**Figure 9B, Supplemental Figure 9B, and Supplemental Table 32**). Similarly, 123 proteins in WT met the biological- and statistical-significance LFQ thresholds, versus only 7 meeting similar thresholds in KO (**Figure 9B**). Direct comparison between WT and KO brains utilizing protein level BTA resulted in a near even split in total persulfidated proteins, with 55% skewed toward WT (**Figure 9C, Supplementary Figure 9C, and Supplemental Table 33**). These results are not surprising given: 1) The equal but opposite direction the kidney sulphydrome is shifted under DR in 6-month old CGL WT (**76% toward DR; Figure 2B**) and KO mice (**74% toward AL; Figure 7E**), and 2) The comparable brain sulphydrome enrichments in 6-month old CGL WT (**87% toward DR; Figure 2D**) and KO mice (**86% toward DR; Figure 8B**).

We next examined CGL dependence for DR-induced shifts utilizing **peptide level BTA (Figure 9A)**. Tissue lysates underwent trypsinization to digest proteins and produce smaller peptides prior to affinity purification, thus overcoming potential intermolecular and intramolecular cysteine-related issues experienced in protein level BTA²⁰. In kidney, 70% of peptides and resultant protein ID’s had higher LFQ intensities in WT over KO (**Figure 9D, Supplemental Figure 9B, and Supplemental Table 34**). Seven proteins in WT met the biological- and statistical-significance LFQ thresholds, versus only 2 meeting similar thresholds in KO (**Figure 9D**). In brain, 74% of the peptides and resultant protein ID’s were skewed toward WT (**Figure 9E, Supplemental Figure 9C, and Supplemental Table 35**). Six proteins in WT met the biological- and statistical-significance LFQ thresholds, versus only 1 meeting similar thresholds in KO (**Figure 9E**). Thus, peptide level pulldown showed CGL dependent kidney persulfidation enrichment under DR similar to protein level pulldown, while it

revealed CGL dependence for brain persulfidation enrichment under DR that was not detected by protein level BTA analyses.

Advancing the peptide level BTA, we applied **multiplex iodoacetyl isobaric tandem mass tag (iodoTMT) labeling of sulfhydryl groups after final tris(2-carboxyethyl)phosphine (TCEP) elution of persulfidated (Figure 9A, Supplemental Figure 9A)**. These steps and approaches advance upon non-labeled peptide BTA as TCEP reduction and subsequent iodoTMT labeling provide quantitative identification of persulfidated proteins and peptide site-level analysis for the modified cysteine residues^{18,21,22}. In both kidney (Figure 9F, and Supplemental Tables 36) and brain (Figure 9G, Supplemental Tables 37) total persulfidated peptides and resultant protein ID's had higher iodoTMT intensities in WT over KO, with 100% enrichment in WT kidney and 80% enrichment in WT brain. Twelve proteins in WT kidney and 2 proteins in WT brain met the biological- and statistical-significance for WT:KO iodoTMT intensity ratio, while no proteins reached similar thresholds in KO tissues (Figures 9F, G). *All of Figure 9 (completely new data) is given below:*

A.5 Reviewer 1: The SSH profiling method used in this study was adapted from a previous report by Gao and colleagues. In a technical aspect, therefore, the novelty of this work was compromised. Moreover, as compared to its original version that enabled a site-level analysis, the present work by Bithi et al., failed to do so.

Our response: Yes, Reviewer #1 is correct that the SSH profiling method used here was adapted from the Gao, et al paper²³, primarily due to those authors utilizing the same proteomics core and expertise at Cleveland Clinic that we have access to. So yes, from a purely technical perspective, the method (even though we adapted and optimized it here for *ex vivo* persulfidation detection from mammalian flash frozen tissues) is not entirely novel. However, the novelty aspect comes from the biological approaches and questions being answered to determine multi tissue-specific and CGL-dependent *in vivo* changes to the mammalian sulfhydrylome as a function of dietary

restriction at three different stages of age in mice: young adult (6-months), middle-aged adult (12-months) and advanced aged (20-months). Additionally, as described in our response to the previous comment (A.4 from Reviewer #1), in our revised manuscript we have now applied peptide level biotin thiol assay followed by iodoTMT multiplex labeling. Not only does this allow for labeled quantification of changes in persulfidation via mass spec (**Figure 9F, G**), but it provides the site-level analysis that Reviewer #1 is requesting. **Site-level analysis for the modified cysteine residues from peptide iodoTMT labeling are provided in the second tabs of Supplemental Tables 36 for DR kidney, Supplemental Tables 37 for DR brain, and Supplemental Tables 38 for AL kidney.**

A.6 Reviewer #1: Though out of “fashion”, DDA-based label-free quantification is still widely adopted in the field of proteomics, especially for measuring changes in protein expression across a number of biological samples. As compared to a routine/standard proteomic experiment, the BTA approach involves many more sample processing steps. It seemed to be extremely challenging, if not impossible, for such a large-scale study to obtain reproducible label free quantification results. In principle, label free quantification based on MS1 intensity would be a better choice for DDA-based quantification. Why did the authors choose spectral counting strategy instead?

Our response: We thank Reviewer #1 for providing insight into label-free quantification approaches. We chose spectral counting for DDA-based label-free quantification as it primarily affords relatively simple and faster sample preparation and data analysis in addition to its ability to quantify protein abundance with reasonable accuracy for most proteins over a dynamic range of 2 to 4 orders of magnitude and up to 60 orders of magnitude in specific cases²⁴⁻²⁶. In our revised version we have now performed several new experiments utilizing three independent experimental mouse cohorts differing in age, diet type, diet duration, and/or strain background along with several variations of the BTA (protein level and peptide level isolation, with the latter being done unlabeled and with iodoTMT labeling) to show the reproducibility of the sulfhydryme enrichments/shifts in our study.

The new additions not outlined previously in our response include sulfhydryme profiling on 20-month old male C57BL/6 mice that had been on an *ad libitum* diet (HFD) or a on an every-other-day (EOD) intermittent fasting diet for 2.5 months (**Figure 5 and shown below for ease of viewing**). Similar shifts in the sulfhydryme were obtained by the EOD fasting diet in these aged mice as what was detected in the younger 6-month mice under the 50% DR diet for 1 week. Thus, the reproducibility of our method, despite the numerous steps of the BTA, is attainable.

Additionally, for proof of principle that the spectral counting method provides similar conclusions as MS1 intensity, we provide the re-analysis of all the raw mass spec data that originally utilized spectral counting for showing relative tissue-specific changes in the sulfhydrome as a function of diet with now using their MS1 intensity on the proteomics software package name MaxQuant (version 1.6.14.0) and data analysis software Perseus (version 1.6.12.0). Please see the below side by side volcano plot comparisons between spectral counting results and MS1 intensity result to visualize the similarities.

Spectral Counting versus MS1 Intensity #1: Comparing label free Spectral Count (SC) and MS1 intensity-based mass spec quantification on 6-month old WT 50% DR and AL fed male mice (n = 4-5 mice/group).

Spectral Counting versus MS1 Intensity #2: Comparing label free Spectral Count (SC) and MS1 intensity-based mass spec quantification on 6-month old CGL KO 50% DR and AL fed male mice (n = 3 mice/group)

Spectral Counting versus MS1 Intensity #3: Comparing label free Spectral Count (SC) and MS1 intensity-based mass spec quantification on 20-month old WT EOD fasting and AL fed male mice (n = 5 mice/group)

A.7 Reviewer #1: If the former was not feasible to the authors, they should at least validate their quantification results using an orthogonal approach and/or demonstrate their instrumental performance by showing QC data obtained during the period of this project.

Our response: We thank reviewer #1 for suggesting the appropriate steps to take to validate our work and ensure the quality of our data. We have presently decided to proceed with the label free spectral count based data and analysis in the manuscript (Figures 2, 3, 4, 5, 7, and 8) rather than incorporate the MS1 as changing the data to MS1 based would require the need to re-synthesize all shared protein and pathway enrichment data panels in the figures. As the directionality and magnitude of the sulfhydryl shifts in all tissues were very similar when analyzed via spectral counting or MS1 (as shown in the above volcano plots showing side-by-side spectral counts plots versus to MS1 plots), we feel inclusion of the spectral count obtained data is suitable and provides the same narrative as the MS1 data, in addition to it being considered acceptable in the field of proteomics²⁴⁻²⁶. That being said, at the request of Reviewer #1, we have included instrumental performance QC1 data obtained during the

entire period of this project., which is described in the updated Materials and Methods section and physically located in **Supplemental Data File 2: Mass Spec QC1 data information** as well we uploaded all of the raw QC1 data to Figshare via: <https://figshare.com/s/941a9d9bc9cd39b0dac6> and <https://figshare.com/s/f613f80460bce4eb722d> and this information is also provided in the updated Material and Methods section. *However, if Reviewer #1 requires that we switch out all spectral count data for the MSI based data, we will kindly oblige.*

We would also like to highlight the new orthogonal approach taken in **Figure 9** to validate our quantification results and shifts in the sulfhydryl as a function of diet and CGL-status, in which we analyzed based on label free intensity quantification instead of spectral counting for both protein level and peptide level BTA (**Figure 9B-E**), and further advanced the study with the use of iodoTMT multiplex labeled quantification and site-level analysis (**Figure 9F, G**).

A.8 Reviewer #1: In many cases, less than 10 hits could be enriched in individual KEGG pathways, posing a concern about the ‘real’ significance of statistics. In this regard, the calculated p-values might be subjected to multiple testing correction (Noble WS. Nat Biotechnol. 2009). Alternatively, the FDR rates for the KEGG enrichment analyses could be provided.

Our response: We would like to emphasize that the KEGG pathway analysis was done by the online-based server g:Profiler (<https://biit.cs.ut.ee/gprofiler/gost>).²⁷ This server/tool does functional enrichment analysis of identified persulfidated proteins by using gene ID’s or accession numbers in the g:GOST (Gene Group Functional Profiling) identifier tool to detect biological pathways from the KEGG database²⁸ significantly enriched with persulfidated proteins. These significance values were auto-calculated by the software’s proprietary g:SCS algorithm that already utilizes multiple testing correction for p-values gained from pathway enrichment analysis and given a threshold of $p < 0.05$. We decided to use the g:SCS algorithm because g:SCS is more conservative than Benjamini-Hochberg False Discovery Rate (FDR)²⁷. Moreover, g:GOST reports the adjusted enrichment P-values (which are shown in the figure). In setting the boundaries for significance determination, it corresponds to an experiment-wide threshold of $\alpha=0.05$, with the g:SCS threshold pre-calculated for list sizes up to 1000 accession number or gene ID terms and analytically approximates a threshold t corresponding to the 5% upper quantile of randomly generated queries of that size. As per the g:Profiler description, all actual p-values resulting from the query are automatically transformed to corrected p-values by multiplying these to the ratio of the approximate threshold t and the initial experiment-wide threshold $\alpha=0.05$ with consideration to the underlying gene sets annotated to terms of each organism from the KEGG database, and therefore gives a tighter threshold to significant result. We have now added more of the description of g:Profiler and its mathematical approaches in the Materials and Methods section in the revised manuscript.

Minors:

A.9 Reviewer #1: Indeed, sulfhydrylation is most widely used in literatures. However, in a chemical aspect, this term implies “hydration” and is inaccurate (Filipovic et al., Chem Rev 2018, 118: 12531337). The term “persulfidation” has been gaining wide acceptance (Bestetti et al., Sci Adv 2018, 4: eaar5770). **Our response:** In our revised manuscript we have changed “sulfhydrylation” to “persulfidation”.

A.10 Reviewer #1: Page 6, the authors claimed that “ex vivo addition of NaHS/H₂S may act as a reducing equivalent, similar to DTT, and remove the in vivo formed sulfhydrylation modifications prior to the BTA assay”. This can be misleading as the effects of NaHS on the formation of persulfidation is believed to be dependent on its concentration. The authors should specify the experimental conditions. **Our response:** We have added the concentration of NaHS (1 mM) to page 6 and in the figure legend for Supplemental Figure 2.

A.11 Reviewer #1: Page 12, “As NaHS itself gives an immediate positive lead sulfide result as it quickly releases H₂S upon placement in aqueous solutions” This is incorrect, as NaHS dissolves to give the HS⁻ anion rather than H₂S (HS⁻ is the predominant form of H₂S at physiological pH). Also, “NaHS is a H₂S donor” is a longstanding but wrong notion. **Our response:** We made the suggested change in our revised manuscript, as on page 13, it now says “As NaHS itself gives an immediate positive lead sulfide result as it quickly releases HS⁻ upon placement in

aqueous solutions at physiological pH". We have also removed "NaHS is a H₂S donor" wording from the manuscript.

A.12 Reviewer #1: Page 28, DIA should be changed to DDA.

Our response: We changed "DIA" to "DDA" on page 31.

A.13 Reviewer #1: There are too many supplementary tables which are not easy to follow.

Our response: We apologize for the sheer number of supplementary tables. However, we feel it is in the best interest for transparency and accessibility of science to include the information contained in the tables, and it is also a requirement for us to add in these source data that was used to make the volcano plots as per the publisher's guidelines. Also, we feel that they make our work an easy to utilize tool and resource for those not experts in proteomics or akin to proteomics analysis software, as one does not have to go and download the raw proteomic files that we've deposited into the ProteomeXchange Consortium in order to see if their protein of interest is persulfidated, and if so, which tissues and in what direction relative to diet and genotype. That being said, we have gone and gotten rid of the tables as PDFs and kept them as Excel files for ease of search and manipulation by the end user.

A.14 Reviewer #1: Supplemental Data File 1 is not very informative, and should be removed.

Our response: We removed the old Supplemental Data File 1, as the raw proteomics-related data have been uploaded to the ProteomeXchange Consortium via the PRIDE partner repository with the following dataset identifier information: Project accession PXD022888 with Project DOI 10.6019/PXD022888 (from Proteome Discoverer 2.4 based analysis), Project accession PXD022956 (from MaxQuant 1.6.14.0 based analysis), and Project accession PXD022954 with Project DOI 10.6019/PXD022954 (from Scaffold 4.8.7 based analysis).

(B) Response to Reviewer #2

B.1 Reviewer #2: The study by Bithi et al, deals with the effects of dietary restriction on the sulfhydryme in various tissues. While the authors have generated several datasets pertaining to the sulfhydryme in different tissues, there are several concerns which cannot be set aside.

Our response: We thank Reviewer #2 for critically reading and providing constructive criticism to our work. We have taken these comments into consideration in revising our manuscript with additional animal models, experimental proteomics approaches, and clarifications in the text. We hope these additions and revisions, principally in **new Figures 5 and 9**, help address the concerns of Reviewer #2.

B.2 Reviewer #2: As the authors themselves acknowledge, there are some serious issues with the method utilized to detect persulfidation. It is well established that sulfenic acids (RSOH) readily react with thiol blocking reagents such as N-ethylmaleimide or iodoacetamide, giving a DTT cleavable adduct (Reisz et al., FEBS J, 2013). On the other hand, Michael-type addition product formed in the reaction of normal thiolate with maleimides could also easily undergo thiol exchange with DTT (particularly when present in large amounts) via retro Michael-type addition even under the conditions used by the authors (Fontaine et al., Bioconjug. Chem, 2015; Wu et al., Bioconjug. Chem, 2018). Finally, it is easy to imagine that biotinylated peptide could also be connected via disulfide bond to another peptide that will be released with DTT treatment. Thus, the approach not only measures R-SSH but also RSOH and RSSR, in addition to a high basal nonspecific release due to retro Michael-type thiol exchange, hence authors cannot claim that they measure changes in sulfhydrylation.

Our response: Thank you to Reviewer #2 for bringing forth the limitations of our work and allowing us to address these either experimentally and/or verbally. As Reviewer #1 has stated “It is totally fine to apply NEM-biotin as the alkylating reagent”, the use of maleimide derivatives to alkylate thiols and persulfides is well established in the field ^{18,19,29-34}. Yet, as Reviewer #2 has pointed out, a few studies have indicated they also bind to sulfenic and sulfinic acids under specific experimental conditions ^{35,36}, making way for potential false positive persulfidation hits. While we do not deny the validity of this concern nor the work in Reisz, et al., FEBS J, 2013, it should be noted that these false positives may be minor as the interaction between maleimide derivatives and oxidized cysteine residues are not as stable under pH 7-8 ³⁶, which is optimal pH for thiol and persulfide alkylation with maleimide ³⁷ and what we used in the current study. In addition, the reported low presence of oxidized cysteine species relative to thiols and persulfides limits the potential for false positives in the analysis of persulfides in the proteome ³⁵. Importantly, the concentration of maleimide derivatives used in the studies reporting interaction with oxidized cysteines (such as sulfenic and sulfinic acids) ranged from 5 mM to 20 mM, which is 14- to 58-fold higher than what was used in our present study, suggesting our lower concentration may avoid off target labeling. Furthermore, in larger scale mammalian whole tissue cysteine modification studies, much like ours, it was shown maleimide derivatives preferentially alkylate and block –SH groups and not –SOH groups on proteins ³⁸. So, again, we do not question the work cited by Reviewer #2 in regards to the Reisz, et al. FEBS J, 2013 article, however, we do disagree that it is well established that sulfenic and sulfinic acids readily react with thiol blocking reagents such as NEM and that our results should not be invalidated, as the interactions of NEM with sulfenic and sulfinic acids may occur under very specific in vitro conditions. In support that it is not well established, we would also like to respectfully point out that the same authors in the Reisz, et al., FEBS J, 2013 article stated in a subsequent article two years later in 2015 that: “Even with the wide use of high resolution instrumentation, distinguishing these two modifications on intact proteins remains a challenge as, for example, a mass accuracy of < 1 ppm would be needed to distinguish between these in a 20 kDa protein. Control experiments in this case could include 1) treatment with reducing agents (e.g., DTT or TCEP) to obtain the reduced thiol form of the analyzed protein and/or 2) reaction with thiol-blocking reagents such as IAM or NEM ³⁹⁻⁴¹. Both are critical as they allow for specific distinction of persulfides against sulfinic acid, noting that the latter cannot be reduced by DTT and is also inert to IAM and NEM treatment ³⁵” by Baez, Reisz, and Furdui “Mass Spectrometry in Studies of Protein Thiol Chemistry and Signaling: Opportunities and Caveats” Free Radic Biol Med 2015; 80:191-211 ⁴². Thus, we feel that if the same authors are advocating for the use of DTT and NEM due to their specificity towards persulfides, then their original findings should not be held up as well established nor disqualify our current work.

On an experimental response to Reviewer #2's comments about the specificity of NEM to block and/or alkylate persulfides, we would like to point out that given the enzymatic activity of CGL⁴³ is likely to only prompt alterations of cysteines with a persulfide and not sulfenic, sulfinic, or sulfonic modifications, the relative minimal impact of oxidized cysteines giving false positive results within our methodology is leveraged by our use of CGL KO mice, in which: 1) There is a reduction in total protein persulfidation numbers in all tissues from KO mice, and 2) There is attenuation of DR induced persulfidation in kidney and brain of KO mice as detected in direct WT versus KO studies utilizing protein and peptide level BTA. Thus, the fact that reduced protein persulfidation levels were observed in tissues from CGL KO mice compared to the wild-type mice verifies that the protein level and the peptide level BTA approaches used indeed detect protein persulfide species, similar to what was seen in CBS and CGL deletion yeast strains in Doka, et al.²⁰ Thus, while false positives are conceivable, the diet-induced and tissue-specific proteomic shifts detected in a CGL-dependent manner in this study infer changes in persulfidation. Our new data addressing the specificity of the method utilizing CGL KO mice as the experimental variable are shown in Figure 9 and Supplemental Figure 9, with head to head comparisons from unlabeled protein level and peptide level biotin thiol assays as well as iodoTMT labeled peptide level biotin thiol assay in CGL WT and KO kidney and brain. The new figure 9 is given below for ease of viewing:

Other concerns Reviewer #2 has raised in comment B.2 that are addressed in Figure 9 include: 1) “Michael-type addition product formed in the reaction of normal thiolate with maleimides could also easily undergo thiol exchange with DTT”, and 2) “biotinylated peptide could also be connected via disulfide bond to another peptide that will be released with DTT treatment”. In regards to the use of DTT, we have substituted DTT with the non-thiol containing TCEP for the final elution step in Figures 9F and G in the peptide level biotin thiol assay, thus avoiding a potential thiol exchange. The results obtained with TCEP parallel those obtained with using DTT, thus strengthening and validating the data obtained throughout the manuscript for the diet-, tissue-, and CGL-dependent enrichments of the sulfhydryl. In regards to disulfide connected proteins giving false positives

during the elution stages, as we had described in the response to Reviewer #1: The Sulphydrome profiles generated thus far were performed on protein level pulldown and subsequent label-free quantification (LFQ). While suitable for protein persulfidation identification and showing relative shifts between groups, these methods are not without their drawbacks. Mainly, protein-level pulldown leads to potential false positive identification due to intermolecular disulfide bonds, thus allowing a non-persulfidated protein to be co-captured with a persulfidated protein and eluted during the final reducing step^{19,20}. Conversely, false negatives are possible due to NM-Biotin bound non-persulfidated intramolecular thiols (RS-NM-Biotin) on the same protein as a labeled persulfidated cysteine residue (RS-S_n-NM-Biotin) preventing final elution with a reducing agent²⁰. While these caveats are valid, it has been hypothesized that concentrations of disulfide bonds in intracellular proteins are low and not expected to be a major source of false positives in the approach utilized¹⁹. In the previous referenced paper by leaders in the redox biology/H₂S field (Filipovic, M.; Zivanovic, J.; Alvarez, B.; and Banerjee, R. Chemical Biology of H₂S Signaling through Persulfidation. Chemical Reviews, 2018, 118, 3, 1254-1337), they state “However, disulfides are not common in intracellular proteins, and the interference by disulfide-containing peptides/proteins in persulfide identification is expected to be low”, and later “concentrations of disulfide bonds intracellular proteins is low and this is not expected to be a quantitatively major drawback of the method.” Nonetheless, we pursued an orthogonal approach (**Figure 9A, Supplemental Figure 9A**) to avoid these limitations and further validate and expand our findings utilizing kidney and brain obtained from an independent cohort of 12-month old CGL WT and KO mice fed 1-week 50% DR.

Direct comparison between WT and KO kidney utilizing protein level BTA revealed the sulphydrome enriched in WT with 87% of all proteins having a higher LFQ intensity in WT over KO (**Figure 9B, Supplemental Figure 9B, and Supplemental Table 32**). Similarly, 123 proteins in WT met the biological- and statistical-significance LFQ thresholds, versus only 7 meeting similar thresholds in KO (**Figure 9B**). Direct comparison between WT and KO brains utilizing protein level BTA resulted in a near even split in total persulfidated proteins, with 55% skewed toward WT (**Figure 9C, Supplementary Figure 9C, and Supplemental Table 33**). These results are not surprising given: 1) The equal but opposite direction the kidney sulphydrome is shifted under DR in 6-month old CGL WT (**76% toward DR; Figure 2B**) and KO mice (**74% toward AL; Figure 7E**), and 2) The comparable brain sulphydrome enrichments in 6-month old CGL WT (**87% toward DR; Figure 2D**) and KO mice (**86% toward DR; Figure 8B**).

We next examined CGL dependence for DR-induced shifts utilizing **peptide level BTA (Figure 9A)**. Tissue lysates underwent trypsinization to digest proteins and produce smaller peptides prior to affinity purification, thus overcoming potential intermolecular and intramolecular cysteine-related issues experienced in protein level BTA²⁰. In kidney, 70% of peptides and resultant protein ID's had higher LFQ intensities in WT over KO (**Figure 9D, Supplemental Figure 9B, and Supplemental Table 34**). Seven proteins in WT met the biological- and statistical-significance LFQ thresholds, versus only 2 meeting similar thresholds in KO (**Figure 9D**). In brain, 74% of the peptides and resultant protein ID's were skewed toward WT (**Figure 9E, Supplemental Figure 9C, and Supplemental Table 35**). Six proteins in WT met the biological- and statistical-significance LFQ thresholds, versus only 1 meeting similar thresholds in KO (**Figure 9E**). Thus, peptide level pulldown showed CGL dependent kidney persulfidation enrichment under DR similar to protein level pulldown, while it revealed CGL dependence for brain persulfidation enrichment under DR that was not detected by protein level BTA analyses.

Advancing the peptide level BTA, we applied **multiplex iodoacetyl isobaric tandem mass tag (iodoTMT) labeling of sulfhydryl groups after final tris(2-carboxyethyl)phosphine (TCEP) elution of persulfidated (Figure 9A, Supplemental Figure 9A)**. These steps and approaches advance upon non-labeled peptide BTA as TCEP reduction and subsequent iodoTMT labeling provide quantitative identification of persulfidated proteins and peptide site-level analysis for the modified cysteine residues^{18,21,22}. In both kidney (**Figure 9F, and Supplemental Tables 36**) and brain (**Figure 9G, Supplemental Tables 37**) total persulfidated peptides and resultant protein ID's had higher iodoTMT intensities in WT over KO, with 100% enrichment in WT kidney and 80% enrichment in WT brain. Twelve proteins in WT kidney and 2 proteins in WT brain met the biological- and statistical-significance for WT:KO iodoTMT intensity ratio, while no proteins reached similar thresholds in KO tissues (**Figures 9F, G**).

B.3 Reviewer #2: Besides, the authors do not perform the real quantitative proteomic approach. 1.25-fold increase is usually impossible to reach as significant even with approaches such as SILAC, and certainly not with n=3.

Our response: Thank you for the comments and suggestions. However, we feel Reviewer #2 may be incorrect as to what we set our biological significance threshold to for spectral count differences between groups. We did not set this to 1.25-fold, but a minimum of 2-fold. We have tried to make this clearer in the Materials and Methods section as well elsewhere in the text and figure legends. We chose spectral counting for the majority of the label-free quantification as it primarily affords relatively simple and faster sample preparation and data analysis in addition to its ability to quantify protein abundance with reasonable accuracy for most proteins over a dynamic range of 2 to 4 orders of magnitude and up to 60 orders of magnitude in specific cases²⁴⁻²⁶. In our revised version we have now performed several new experiments utilizing three independent experimental mouse cohorts differing in age, diet type, diet duration, and/or strain background along with several variations of the biotin thiol assay (protein level and peptide level isolation, with the latter being done unlabeled and with iodoTMT labeling for a quantitative and site-level proteomic approach) to show the reproducibility of the sulfhydryme enrichments/shifts our study. Again, the head to head quantitative proteomic approach using iodoTMT labeling is found in Figure 9F and G, as well as Supplemental Figure 9D.

In regards to biological N-values, yes, Reviewer #2 is correct for the CGL KO *ad libitum* versus the CGL KO diet restriction those initial experiments were performed with 3 mice per group. However, in other experimental set ups (50% DR in WT mice and EOD in WT mice) we used n=4 to 5 in each group. In this revised manuscript, we have performed a completely new independent test regarding the reproducibility and validity of our results. The new additions not outlined previously in our response to Reviewer #2 includes sulfhydryme profiling on 20-month old male C57BL/6 mice that had been on either an *ad libitum* diet (HFD) or a on an every-other-day (EOD) intermittent fasting diet for 2.5 months (**Figure 5 and shown below for ease of viewing**).

As can be seen from Figure 5 above, similar shifts in the sulfhydryme were obtained by the EOD fasting diet in these aged mice as what was detected in the younger 6-month mice under the 50% DR diet for 1 week. Thus, the reproducibility of our method, despite the numerous steps of the BTA, is attainable. Also, we would also like to point out that the use of n =3 for both cell based and tissue based assays was deemed sufficient by the Filipovic group in their article describing differences in tissue persulfidation between WT and CGL KO mice⁴⁴

B.4 Reviewer 2: The authors often compare the number of detected proteins. This is not a correct method for analysis, as differences in the number of identified peptides/proteins cannot be extrapolated to the level of modification. They could as well be a consequence of differential extraction efficiency from different tissues. The bioinformatics part is very basic. The cysteine proteome is well defined with the recent findings bringing the bioinformatics analysis to a much higher level (Xiao et al, Cell, 2020).

Our response: We thank Reviewer #2 for bringing up concerns in regards to our inclusion of comparisons between the absolute numbers of tissues in WT and KO tissues. While we do state in each tissue the absolute number of persulfidated proteins identified, we do not make any statements as to the level of modification for said proteins (*aka* the ratio of persulfidated vs non-persulfidated). This is a completely different form of analysis and most likely could not be completed on a multi-tissue/multi-diet/multi-genotype global sulphydrome project that we have presented here. Nonetheless, we have added additional statistical analysis to our work, with stating the % shift in the sulphydrome for each tissue studied under the following head to head comparison conditions: WT AL vs. WT DR; KO AL vs. KO DR; WT AL vs WT EOD; WT DR vs KO DR; WT AL vs KO AL. Thus, it is now easier to compare how the sulphydrome shifts or is enriched comparing between groups in a specific tissue. As all of these head to head comparison conditions were completed at the same time within experiment, there is no reason to believe extraction efficiency or technical variables would be the cause for these differences. Additionally, equal amounts of protein lysates are input into the each individual BTA (as can be seen from the stained gel images in the supplemental figures), so even if there are extraction differences between different tissue types, this does not impact the amount of protein going into each assay. In addition, while the Xiao et al, Cell, 2020 paper is a tour de force and landmark paper in the field of redox biology, it does not address persulfidation/sulphydrome analysis. We also believe the presentation of our data is accessible and understandable for a wide scientific audience and not just those with expertise in bioinformatics and proteomic fields.

Additionally, we have performed the head to head iodoTMT Multiplex labeled quantitative approach in Figure 9 in brain and kidney from CGL WT and KO mice under DR, as well as in Supplemental Figure 9 for kidney from CGL WT and KO mice under ad libitum feeding. This allows us to quantitatively detect differences between WT and KO based on the intensity of the iodoTMT label. In addition, to enhance the bioinformatics analysis to a higher level, this labeling approach provides the site-level analysis **for the modified cysteine residues and is provided in the second tabs of Supplemental Tables 36 for DR kidney, Supplemental Tables 37 for DR brain, and Supplemental Tables 38 for AL kidney.**

B.5 Reviewer #2: In addition to these major concerns, the method to detect H₂S is not sensitive or quantitative enough. The authors mention that not enough H₂S is produced in the brain upon dietary restriction, yet the enrichment of the sulphydrated proteins is among the highest there. Obviously, the liver produces higher H₂S in comparison to the brain, but the method (lead acetate-based method) is clearly not sensitive or quantitative enough.

Our response: Thank Reviewer #2 for bringing up this issue. As we already provide an explanation to the similar issue brought up by Reviewer #1, we will address it similarly here. Yes, the lead acetate method for measuring H₂S production gives only semiquantitative data and not absolute in terms of ppm of the H₂S gas in a sample. However, semiquantitative data generated like this and quantified via ImageJ for IntDen is very similar to other techniques widely used and accepted in biological research, such as Western Blots and DNA gels, for examining relative differences between a control group and an experimental groups. Our approach is optimal for measuring the relative differences in H₂S production capacity amongst individual samples and different groups all at one time (like what is shown in Figure 1), as equal exposure time of the lead acetate paper to the headspace of all the wells simultaneously is ensured (much like all bands in a Western Blot are equally exposed to a film when it is overlays over all bands at once). Again, while we agree that there are limitations to the lead acetate/lead sulfide assay in terms of obtaining absolute quantification, it is still widely used and accepted as a method that has selectivity and sensitivity for H₂S in the headspace/gas phase¹⁻⁹, and was recently used this year as the primary method to measure H₂S production capacity in liver derived cells in this same journal by Xu, Qing; et al. “HNF4α regulates sulfur amino acid metabolism and confers sensitivity to methionine restriction in liver cancer” Nature

Communications 11, 3978 (2020)¹⁰. Additionally, highlighting the sensitivity and selectivity of the lead acetate/lead sulfide methods, we previously published (Yang, J. et al. Communications Biology 2, 194 (2019))¹¹ that use of selected ion flow tube mass spectrometry (SIFT-MS)^{12,13} which is a quantitative, selective, and sensitive headspace analysis tool that can be used for H₂S measurement, validated our lead acetate/lead sulfide data and confirmed it is H₂S and not other volatile/reactive sulfide species generated from the reaction present in the gas phase detected in the lead acetate/lead sulfide method. However, we did not utilize SIFT-MS in the present paper for the main reason that each sample must be analyzed individually, thus making it difficult to ensure equal incubation time of each H₂S production capacity reaction at 37degC prior to sampling. As there are 36 individual biological samples analyzed in Figure 1B and C, it becomes apparent that analyzing that many samples via SIFT-MS would be technically challenging and produce results that may not be accurate.

Nonetheless, in the current work addressing the sensitivity of the assay, we have addressed these concerns by now showing the background reaction mixture wells/dots to indicate we are detected above background in:

A) The wells/dots labeled as (-) Tissue in **Figure 1B and C** and in the **Supplemental Data File 1: Uncropped Source Data for Figure 1**.

B) The wells/dots labeled as (-) CGL in **Figure 6, Supplemental Figure 6, and in the Supplemental Data File 1: Uncropped source Data for Figure 6**.

These reaction mixture wells were always taken into account even in our first version of the manuscript and data obtained from wells with tissues were normalized to these blank reaction mixture only wells. We have now shown these background wells/spots and made it clearer in the Materials and Methods section and in the Figure legends that these reaction only background values were subtracted from the experimental sample values when plotting the graphs. In doing so, it is now more apparent that the H₂S production identified from tissues is above background, thus ensuring the sensitivity of the assay to detect H₂S production even in low producing tissues. Additionally, it should be highlighted that in Figure 6A and B, that our lead acetate/lead sulfide method is sensitive enough to detect a linear increase in H₂S production as a function of increasing CGL concentration (0, 1, 2.5, 5, and 10 µg per reaction) after a 2 hr incubation at 37^oC. This data from updated **Figure 6A & B** supporting the sensitivity of the lead acetate/lead sulfide assay is shown to the right:

In addition to the *in vitro* assay utilizing purified CGL shown above, we previously published a study (Hine and Mitchell, Bio Protoc 7(13), 2020)¹⁴ that showed increasing liver protein lysate between 0 to 500 µg induces a linear H₂S production detection from the lead acetate/lead sulfide method up to 300 µg

[Redacted]

In our current assays, we utilized 100 µg of protein lysate/reaction, thus ensuring that we would stay in the linear range for detection. In summary, with the information and data provided here, we feel the experimental data obtained from the lead acetate/lead sulfide method is sensitive and appropriate for detecting relative changes in H₂S production capacity as a function of diet, genotype, and *in vitro* treatment.

B.6 Reviewer #2: A direct correlation between sulfhydration and CSE levels has been recently demonstrated in striatal mouse cell lines and rat brain (Zivanovic et al, Cell Metab, 2019) so there should be no surprise why the authors see the highest increase there. Neurons predominantly express CSE (Sbodio et al, Brit J Pharmacol, 2018).

Our response: In the revised manuscript, we have refined our hypothesis that H₂S production capacity does not necessarily have to increase with DR for enhanced persulfidation in select tissues, as we describe more in depth in the Discussion of this revised manuscript. We now hypothesize that persulfidation *in vivo* involves a complex series of modifications, events, and/or factors, and simply having more H₂S or CGL does not necessarily mean increased persulfidation *in vivo*, as can be read in the updated Discussion. **For this reason and the new data shown below, we have now removed the correlative data pertaining to basal *ad libitum* H₂S production capacity with total numbers of persulfidated proteins in this revised manuscript.** We have also added new data that brings to light the complexities of persulfidation *in vivo*. As seen in Figure 1B, CGL KO mice have decreased kidney H₂S production capacity under *ad libitum* feeding compared to WT mice. **However, in new data presented in Supplemental Figure 9D comparing head-to-head differences in protein persulfidation in these same kidneys using a new peptide-level biotin thiol assay combined with iodoTMT labeling, we observe there is not an enrichment for persulfidated proteins in WT mice, and if anything there is slight enrichment in the KO kidneys under *ad libitum* feeding. It is not until the mice are under dietary restriction the dependence for CGL for enhanced persulfidation is revealed, as shown in new Figure 9F.** The data from these referenced figures from the revised manuscript are given below:

This suggests: 1) Basal persulfidation activity (at least in the kidney) under AL feeding or unstressed conditions is independent of CGL and/or CGL derived H₂S, and 2) Enhanced persulfidation in select tissues relies not only on CGL (or CBS, 3MST) derived H₂S but its coordination with other cysteine- and thiol- redox modifications induced by DR^{15,16} or other hormetic stressors as suggested by others^{17,18}. This is in line with the recent publication by Gao, *et al*¹⁸ highlighting a multi-step redox thiol switch from thiol glutathionylation to persulfidation.

[Redacted]

B.7 Reviewer #2: Besides, the study does not provide novel conceptual advances over what is already known in literature. This study provides a large list of proteins persulfidated in response to dietary restriction using a method which is not rigorous enough, which does not significantly advance the field to warrant publication in the journal.

Our response: While we appreciate the reviewer's feedback, we respectfully disagree as to the novel conceptual advances our work provides. Here, we found CGL dependent alterations in H₂S production capacity and sulfhydryl enrichment induced by DR to vary based on tissue, summarized in Figure 10. Utilizing three independent experimental mouse cohorts differing in age, diet type, diet duration, and/or strain background along with several variations of the BTA, we found DR enhanced the sulfhydryls of liver, kidney, muscle, and brain while contracted those of heart. Dependence for CGL in diet induced sulfhydryl expansion was most prominent in kidney and muscle, and to a certain extent in brain. While this is not the first study to recognize relative changes in mammalian protein persulfidation as a function of CGL or diet^{44,45}, it is the first to reveal individual identities and enrichment shifts of these proteins, their biological pathway involvement, sites of modification, and the interdependence of tissue-type, diet, and CGL to impact persulfidation profiles.

We feel the inclusion of our new data sets pertaining to late life initiated long-term dietary restriction in the form of every other day fasting in 20 month old mice and 1-week 50% dietary restriction in 1 year old CGL WT and KO mice enhance the novel conceptual advances, as no other paper has looked at these specific ages, diet-types, and applied in CGL WT and KO mice. Additionally, the use of iodoTMT labeled peptide level BTA as seen in Figure 9 and Supplementary Figure 9 highlight that CGL is required for sulfhydryl enrichment under dietary restriction, but not so much under unstressed *ad libitum* feeding conditions. No prior publications have addressed this issue *in vivo* nor have they deciphered the contribution CGL provides for tissue sulfhydryls under stressed and non-stressed conditions *in vivo*.

(C) Response to Reviewer #3

C.1 Reviewer #3: This is a well-written paper describing experiments that further our understanding of the mechanisms underlying the effects of dietary restriction on health and longevity. The authors describe a unique approach that examines the sulfhydromes of various tissues in response to dietary restriction as well as a lack of a critical enzyme, cystathionase (gamma lyase). The results strengthen the idea that methionine metabolism plays a major role in the potential effects of dietary restriction on longevity. The data is novel and exciting especially to those that study the role of diet in health. The methods and analysis are thoughtful and thorough in terms of techniques to measure H₂S and sulfhydrylation.

Our response: We thank Reviewer #3 for their careful and critical reading of our manuscript, as well as their constructive criticism for ways to improve our manuscript.

C.2 Reviewer #3: The authors suggest that that H₂S signaling has only been recently recognized as proposed mechanism in lifespan extension. However, several papers published 15-20 years ago (that were not acknowledged) demonstrated that an upregulation of multiple components of the methionine metabolic pathway (that includes CGL and from which H₂S is derived) was likely a major contributor to enhanced stress resistance and longevity of long-living mice (Ames dwarf and Growth hormone receptor knockout mice) while suppressed methionine metabolism contributed to shortened lifespans exhibited by growth hormone transgenic mice. The point being that there were discoveries quite a while ago suggesting the importance of this pathway in longevity. Also, numerous reports indicate that 50% of the cysteine generated via the transsulfuration pathway is utilized in glutathione synthesis while a significant amount is metabolized into taurine. Thus, it has not yet been demonstrated that H₂S is the primary factor responsible for the beneficial effects of this pathway on health and longevity but it is indeed a contributor and warrants continued research. This paper is a productive step in that direction. Transsulfuration operates in liver, kidney and brain thus it makes sense that these tissues were largely impacted by both DR and in the CGL mice.

Our response: We thank Reviewer #3 for highlighting the need for us to include some of the landmark studies related to sulfur amino acid metabolism and longevity models. We have now added the following text (highlighted here) in the introduction: “A common molecular phenomenon amongst several dietary and endocrine models of longevity is the altered metabolism of sulfur containing amino acids methionine and cysteine, which entails flux through transsulfuration¹³⁻¹⁵ and increased production capacity and/or bioavailability of hydrogen sulfide (H₂S) gas¹⁶. H₂S and its HS⁻ anion and S²⁻ ion, herein referred simply as H₂S, is historically classified as an environmental and occupational hazard⁴⁶⁻⁴⁸.”

This new addition adds the following historical papers from 15-20 years ago:

13 Uthus, E. O. & Brown-Borg, H. M. Methionine flux to transsulfuration is enhanced in the long living Ames dwarf mouse. *Mech Ageing Dev* 127, 444-450, doi:10.1016/j.mad.2006.01.001 (2006).

14 Brown-Borg, H. M., Rakoczy, S. G. & Uthus, E. O. Growth hormone alters methionine and glutathione metabolism in Ames dwarf mice. *Mech Ageing Dev* 126, 389-398, doi:10.1016/j.mad.2004.09.005 (2005).

15 Uthus, E. O. & Brown-Borg, H. M. Altered methionine metabolism in long living Ames dwarf mice. *Exp Gerontol* 38, 491-498 (2003).

Additionally, we have included references to others' work in regards to longevity models and alterations in transsulfuration and CGL expression:

23 Miller, D. L. & Roth, M. B. Hydrogen sulfide increases thermotolerance and lifespan in *Caenorhabditis elegans*. *Proc Natl Acad Sci U S A* 104, 20618-20622, doi:10.1073/pnas.0710191104 (2007).

36 Kabil, H., Kabil, O., Banerjee, R., Harshman, L. G. & Pletcher, S. D. Increased transsulfuration mediates longevity and dietary restriction in *Drosophila*. *Proc Natl Acad Sci U S A* 108, 16831-16836, doi:10.1073/pnas.1102008108 (2011).

38 Tyshkovskiy, A. et al. Identification and Application of Gene Expression Signatures Associated with Lifespan Extension. *Cell Metab* 30, 573-593 e578, doi:10.1016/j.cmet.2019.06.018 (2019).

If Reviewer #3 wishes additional specific papers to be referenced, we will happily oblige. Additionally, we are thankful for Reviewer #3's insights into the importance of studying H₂S as it relates to longevity. While out of the scope of this paper, we thought we would mention that we currently have a lifespan study ongoing with CGL WT and KO mice under *Ad libitum* and 35-45% DR started at 6 months of age. As expected, the WT mice under DR have increased median and maximum lifespans, however, CGL KO mice fail to gain many of the lifespan and healthspan extending benefits as the WT mice under DR.

C.3 Reviewer #3: In terms of experimental design, it's curious as to why they chose a one-week dietary restriction protocol to examine the changes to the sulfhydromes of various tissues. The impact of DR on many parameters that impact health and longevity adapt more slowly to the effects of reduced calories. Collection of tissues after two months of DR (a minimum standard in the aging field) would likely have shown a much more dramatic and perhaps even a different picture than that obtained after one week. One other question not adequately described in the methods is how was food intake measured (individual or groups of mice?). Five grams of food per day is on the higher end. Many studies have indicated 3-4 grams of food are consumed per day. What was the methionine and cysteine content of the diets? There are many studies indicating that reduced methionine content of the diet extends health and lifespan. How do the authors address a 50% reduction in methionine in this study?

Our response: We chose 1-week of 50% DR because we had previously shown it to be sufficient at enhancing H₂S production capacity in the liver and kidney and deriving stress resistance benefits in a CGL dependent manner⁴⁹. Thus, we hypothesized that this same amount and duration of DR would elicit changes in the sulfhydrome.

In consideration to Reviewer #3's comments, we agree 1 week of 50% DR tested in the young adult mice is most likely not long enough to extend longevity nor represent a tool for probing resiliency and adaptation at older ages. The impact of DR on longevity parameters sometimes arise more latently to the effects of reduced calories and/or timed feeding. Previously, Zivanovic, *et al.* showed that persulfidation in rat brain and heart tissues decreased with advanced age at 12-24 months, and that long-term daily 30% DR in mice between 2 months to 20 months of age rescued ageing-related persulfidation loss in the liver⁴⁴. However, studies examining the effects of long-term daily DR on older humans are somewhat difficult to execute due to patient adherence^{50,51} and need for medical oversight to ensure caloric intake guidelines are safely followed⁵²⁻⁵⁶. Given these limitations, the relatively implementable yet effective DR regimen of intermittent fasting has gained substantial attention as an alternative method to continuous, chronic DR^{57,58}. There are several types of intermittent fasting regimens, with every-other-day (EOD) fasting being one of the most popular^{59,60}. In EOD fasting, animals and/or participants abstain from food for a day (i.e., 24 hr) followed by food intake *ad libitum* (AL) for the next 24 hr. This method is feasible in older adult humans⁶¹, and in aged rodents augments lifespan⁶²⁻⁶⁶ and improves metabolic flexibility⁶³.

Therefore, we utilized late-life initiated EOD fasting for 2.5 months in 20 month old male C57BL/6 mice to test if this more translatable form of DR is capable of altering tissue-specific sulfhydromes in aged subjects (Figure 5A). Similar to our findings in young adult mice, EOD fasting augmented protein persulfidation in liver (Figure 5B, Supplemental Figure 5A, Supplemental Table 16), kidney (Figure 5C, Supplemental Figure 5B, Supplemental Table 17), muscle (Figure 5D, Supplemental Figure 5C, Supplemental Table 18), and brain (Figure 5E, Supplemental Figure 5D, Supplemental Table 19), while it decreased persulfidation in the heart (Figure 5F, Supplemental Figure 5E, Supplemental Table 20). Overall numbers of persulfidated proteins identified in each tissue from these aged mice were on the same order to what was found in the young adult mice. While the shift in protein persulfidation enrichment under 2.5 months EOD fasting in aged mice for kidney (Figure 5C) and muscle (Figure 5D) was not as robust as that found under 1-week 50% DR in young

adult mice, a greater number of proteins meeting the biological- and statistical-thresholds were found in the EOD fasting group. Thus, despite being completely different forms and durations of DR, in addition to being testing on mice of different ages and strain background/source, long-term EOD fasting and short term 50% DR alter tissue-specific sulfhydromes in an analogous manner. Figure 5 has been copied below for ease of viewing this new data:

Figure 5: Late-life initiated Every Other Day (EOD) fasting for 2.5 months in aged mice transforms tissue-specific sulfhydromes similar to short-term DR in young adult mice. (A) Graphical presentation of the overarching experimental setup. 20-month old male C57BL/6 mice were placed on *ad libitum* (AL) or Every Other Day (EOD) fasting for 2.5-months prior to tissue collection. Tissues were analyzed for protein persulfidation/sulphydrome profiles via the biotin thiol (BTA) assay. (B-F) Volcano plots showing differentially abundant persulfidated proteins in liver (B), kidney (C), quadriceps muscle (D), whole brain (E), and heart (F) from AL fed (n = 5 mice/group) versus 2.5 months of EOD fasting (n = 5 mice/group) 20-month old C57BL/6 male mice. The $\log_2(\text{Fold Change DR:AL})$ X-axis displays the average fold change in spectral counts for each identified persulfidated protein while the $-\log_{10}$ Y-axis displays the calculated p-value when comparing the individual spectral count values for each identified persulfidated protein in a specific tissue from AL versus EOD fed mice. The non-axial red dotted vertical lines highlight the biological significance threshold of ± 2 -fold change in spectral counts between EOD versus AL, while the non-axial red dotted horizontal line with asterisk highlights our statistical significance threshold of $p < 0.05$. The number of total persulfidated proteins identified in each tissue are given next to the tissue name, while blue (AL enriched) and green (EOD enriched) colored dots and text indicate persulfidated proteins reaching both biological- and statistical- thresholds. Gray color dots indicate persulfidated proteins not reaching the criteria for both biological and statistical significance for enrichment under either diet. The percentage and direction of proteins skewed toward one diet type is provided above the tissue label.

Additional comments and concerns by Reviewer #3 are related to the methods we utilized for dietary restriction implementation, sulfur amino acid composition of the diet, and group housing of mice. We have updated these in the materials and methods section with the goal of making them more clear and repeatable. We have copied and highlighted the pertinent parts below:

“When mice were 6-months or 12-months old they were switched to the experimental AIN-93G-based diet (Research Diets D10012G-2V-Formula 1) and exposed to *ad libitum* access for several days to adapt to the new food and for monitoring intake. The experimental diet consists of 20% of calories from protein (casein-based containing approximately 2.9% methionine and 0.4% cysteine, with additional L-cystine supplemented to a final 1.5 g/1,000 g final diet composition by Research Diets, Inc.), 64% of calories from carbohydrate, and 16% calories from fat, and importantly has 2x concentrations of mineral mix S10022G, vitamin mix V10037, and choline bitartrate to avoid potential micronutrient malnutrition during 50% dietary restriction. The powdered food mix was added in a 1:1 ratio (gram:mL) to a 2% agar (Sigma #A1296) solution in water before solidification to a semi-solid consistency that lessens the potential for food hoarding in group housing and increases accuracy of food consumption measurement. *Ad libitum* food intake per cage was measured daily for up to four days to determine the correct amount to restrict to achieve 50% reduction in food intake. After randomly assigning *ad libitum* (AL) (n = 3-4/genotype/experiment) or diet restriction (DR) (n = 3-5/genotype/experiment) feeding to the cages, food intake and body mass were measured over the 1-week intervention. AL fed mice were provided 24-hour access to the diet and the DR mice fed their calculated allotment near the start of their dark phase at 7pm to limit disturbances in circadian rhythms and feeding patterns between the two groups^{67,68}.

For aged mice experiments testing every-other-day (EOD) fasting, male C57BL/6 mice were obtained between 60–65 weeks of age (Stock No: 000664, Jackson Laboratories, Bar Harbor, ME) and group-housed (up to 5 same-sex mice per cage) in the Cleveland Clinic Lerner Research Institute Biological Resource Unit. The mice had *ad libitum* (AL) access to standard rodent chow (18.6% protein, 44.2% carbohydrate, and 6.2% fat; Teklad Global Rodent Diet #2918, Envigo, Madison, WI). At approximately 20 months of age, cages were randomly assigned to either EOD fasting (n = 5) or AL access (n = 5) to the standard rodent chow. The EOD fasting regimen consisted of repeated cycles of 24-hr consecutive removal of food access with water always available (fast day) followed by 24-hr access to food and water (fed day). The EOD fasting intervention proceeded for 2.5 months. To circumvent possible disturbances in circadian rhythms and feeding patterns in the Chow EOD group, the food was provided to or removed from the Chow EOD group just prior to the dark cycle onset at 7pm.”

In regards to amount of food intake, the 5 gram totals are for our custom Research Diets, Inc lab-made diets that we produce into a semi-solid agar, thus the wet weight of these diets is more than standard animal facility dry food pellets, thus the reason for the increased weight of food intake.

C.4 Reviewer #3: The statistics for the H2S production and similar assays are reasonable. Our response: We thank Reviewer #3 for their comment on the statistical analysis of the paper.

C.5 Reviewer #3: Minor: there are two files labeled Supplemental Table 1 – should be liver and table 2 should be kidney. Our response: We thank Reviewer #3 for catching this typographical error. This has now been corrected, along with re-formatting of all Tables into modifiable Excel documents instead of PDFs.

References for Response to Reviewers

- 1 Cha, J. H., Kim, D. H., Choi, S. J., Koo, W. T. & Kim, I. D. Sub-Parts-per-Million Hydrogen Sulfide Colorimetric Sensor: Lead Acetate Anchored Nanofibers toward Halitosis Diagnosis. *Anal Chem* **90**, 8769-8775, doi:10.1021/acs.analchem.8b01273 (2018).
- 2 Lim, S. H., Feng, L., Kemling, J. W., Musto, C. J. & Suslick, K. S. An optoelectronic nose for the detection of toxic gases. *Nat Chem* **1**, 562-567, doi:10.1038/nchem.360 (2009).
- 3 Yoshida, S. *et al.* Role of dietary amino acid balance in diet restriction-mediated lifespan extension, renoprotection, and muscle weakness in aged mice. *Aging Cell*, e12796, doi:10.1111/accel.12796 (2018).
- 4 Wang, X. *et al.* Hydrogen Sulfide (H₂S) Releasing Capacity of Isothiocyanates from *Moringa oleifera* Lam. *Molecules* **23**, doi:10.3390/molecules23112809 (2018).
- 5 Zhang, Y. & Weiner, J. H. A simple semi-quantitative in vivo method using H₂S detection to monitor sulfide metabolizing enzymes. *Biotechniques* **57**, 208-210, doi:10.2144/000114218 (2014).
- 6 Ugliano, M. & Henschke, P. A. Comparison of three methods for accurate quantification of hydrogen sulfide during fermentation. *Anal Chim Acta* **660**, 87-91, doi:10.1016/j.aca.2009.09.049 (2010).
- 7 Longchamp, A. *et al.* Amino Acid Restriction Triggers Angiogenesis via GCN2/ATF4 Regulation of VEGF and H₂S Production. *Cell* **173**, 117-129 e114, doi:10.1016/j.cell.2018.03.001 (2018).
- 8 Li, H. *et al.* The interaction of estrogen and CSE/H₂S pathway in the development of atherosclerosis. *Am J Physiol Heart Circ Physiol* **312**, H406-H414, doi:10.1152/ajpheart.00245.2016 (2017).
- 9 Shatalin, K., Shatalina, E., Mironov, A. & Nudler, E. H₂S: a universal defense against antibiotics in bacteria. *Science* **334**, 986-990, doi:10.1126/science.1209855 (2011).
- 10 Xu, Q. *et al.* HNF4 α regulates sulfur amino acid metabolism and confers sensitivity to methionine restriction in liver cancer. *Nat Commun* **11**, 3978, doi:10.1038/s41467-020-17818-w (2020).
- 11 Yang, J. *et al.* Non-enzymatic hydrogen sulfide production from cysteine in blood is catalyzed by iron and vitamin B6. *Communications Biology* **2**, 194, doi:10.1038/s42003-019-0431-5 (2019).
- 12 Navaneethan, U. *et al.* Volatile organic compounds in bile can diagnose malignant biliary strictures in the setting of pancreatic cancer: a preliminary observation. *Gastrointest Endosc* **80**, 1038-1045, doi:10.1016/j.gie.2014.04.016 (2014).
- 13 Spanel, P. & Smith, D. Quantification of hydrogen sulphide in humid air by selected ion flow tube mass spectrometry. *Rapid Commun Mass Spectrom* **14**, 1136-1140, doi:10.1002/1097-0231(20000715)14:13<1136::AID-RCM998>3.0.CO;2-R (2000).
- 14 Hine, C. & Mitchell, J. R. Endpoint or Kinetic Measurement of Hydrogen Sulfide Production Capacity in Tissue Extracts. *Bio Protoc* **7**, doi:10.21769/BioProtoc.2382 (2017).
- 15 Ristow, M. & Schmeisser, S. Extending life span by increasing oxidative stress. *Free Radic Biol Med* **51**, 327-336, doi:S0891-5849(11)00312-1 [pii] 10.1016/j.freeradbiomed.2011.05.010 (2011).
- 16 Schulz, T. J. *et al.* Glucose restriction extends *Caenorhabditis elegans* life span by inducing mitochondrial respiration and increasing oxidative stress. *Cell Metab* **6**, 280-293, doi:S1550-4131(07)00256-2 [pii] 10.1016/j.cmet.2007.08.011 (2007).
- 17 Wei, Y. & Kenyon, C. Roles for ROS and hydrogen sulfide in the longevity response to germline loss in *Caenorhabditis elegans*. *Proc Natl Acad Sci U S A* **113**, E2832-2841, doi:10.1073/pnas.1524727113 (2016).
- 18 Gao, X. H. *et al.* Discovery of a Redox Thiol Switch: Implications for Cellular Energy Metabolism. *Mol Cell Proteomics* **19**, 852-870, doi:10.1074/mcp.RA119.001910 (2020).
- 19 Filipovic, M. R., Zivanovic, J., Alvarez, B. & Banerjee, R. Chemical Biology of H₂S Signaling through Persulfidation. *Chem Rev* **118**, 1253-1337, doi:10.1021/acs.chemrev.7b00205 (2018).
- 20 Doka, E. *et al.* A novel persulfide detection method reveals protein persulfide- and polysulfide-reducing functions of thioredoxin and glutathione systems. *Sci Adv* **2**, e1500968, doi:10.1126/sciadv.1500968 (2016).
- 21 McAlister, G. C. *et al.* Increasing the multiplexing capacity of TMTs using reporter ion isotopologues with isobaric masses. *Anal Chem* **84**, 7469-7478, doi:10.1021/ac301572t (2012).

- 22 Getz, E. B., Xiao, M., Chakrabarty, T., Cooke, R. & Selvin, P. R. A comparison between the sulfhydryl reductants tris(2-carboxyethyl)phosphine and dithiothreitol for use in protein biochemistry. *Anal Biochem* **273**, 73-80, doi:10.1006/abio.1999.4203 (1999).
- 23 Gao, X. H. *et al.* Quantitative H₂S-mediated protein sulfhydration reveals metabolic reprogramming during the integrated stress response. *Elife* **4**, e10067, doi:10.7554/eLife.10067 (2015).
- 24 Al Shweiki, M. R. *et al.* Assessment of Label-Free Quantification in Discovery Proteomics and Impact of Technological Factors and Natural Variability of Protein Abundance. *J Proteome Res* **16**, 1410-1424, doi:10.1021/acs.jproteome.6b00645 (2017).
- 25 Asara, J. M., Christofk, H. R., Freemark, L. M. & Cantley, L. C. A label-free quantification method by MS/MS TIC compared to SILAC and spectral counting in a proteomics screen. *Proteomics* **8**, 994-999, doi:10.1002/pmic.200700426 (2008).
- 26 Arike, L. & Peil, L. Spectral counting label-free proteomics. *Methods Mol Biol* **1156**, 213-222, doi:10.1007/978-1-4939-0685-7_14 (2014).
- 27 Raudvere, U. *et al.* g:Profiler: a web server for functional enrichment analysis and conversions of gene lists (2019 update). *Nucleic Acids Res* **47**, W191-W198, doi:10.1093/nar/gkz369 (2019).
- 28 Kanehisa, M., Sato, Y., Kawashima, M., Furumichi, M. & Tanabe, M. KEGG as a reference resource for gene and protein annotation. *Nucleic Acids Res* **44**, D457-462, doi:10.1093/nar/gkv1070 (2016).
- 29 Sen, N. *et al.* Hydrogen sulfide-linked sulfhydration of NF- κ B mediates its antiapoptotic actions. *Mol Cell* **45**, 13-24, doi:10.1016/j.molcel.2011.10.021 (2012).
- 30 Chouchani, E. T. *et al.* Mitochondrial ROS regulate thermogenic energy expenditure and sulfenylation of UCP1. *Nature* **532**, 112-116, doi:10.1038/nature17399 (2016).
- 31 Li, R. & Kast, J. Biotin Switch Assays for Quantitation of Reversible Cysteine Oxidation. *Methods Enzymol* **585**, 269-284, doi:10.1016/bs.mie.2016.10.006 (2017).
- 32 Cuevasanta, E. *et al.* Reaction of Hydrogen Sulfide with Disulfide and Sulfenic Acid to Form the Strongly Nucleophilic Persulfide. *J Biol Chem* **290**, 26866-26880, doi:10.1074/jbc.M115.672816 (2015).
- 33 Alcock, L. J., Perkins, M. V. & Chalker, J. M. Chemical methods for mapping cysteine oxidation. *Chem Soc Rev* **47**, 231-268, doi:10.1039/c7cs00607a (2018).
- 34 Yang, J., Carroll, K. S. & Liebler, D. C. The Expanding Landscape of the Thiol Redox Proteome. *Mol Cell Proteomics* **15**, 1-11, doi:10.1074/mcp.O115.056051 (2016).
- 35 Reisz, J. A., Bechtold, E., King, S. B., Poole, L. B. & Furdui, C. M. Thiol-blocking electrophiles interfere with labeling and detection of protein sulfenic acids. *FEBS J* **280**, 6150-6161, doi:10.1111/febs.12535 (2013).
- 36 Kuo, Y. H. *et al.* Profiling Protein S-Sulfination with Maleimide-Linked Probes. *Chembiochem* **18**, 2028-2032, doi:10.1002/cbic.201700137 (2017).
- 37 Nanda, J. S. & Lorsch, J. R. Labeling of a protein with fluorophores using maleimide derivitization. *Methods Enzymol* **536**, 79-86, doi:10.1016/B978-0-12-420070-8.00007-6 (2014).
- 38 Saurin, A. T., Neubert, H., Brennan, J. P. & Eaton, P. Widespread sulfenic acid formation in tissues in response to hydrogen peroxide. *Proc Natl Acad Sci U S A* **101**, 17982-17987, doi:10.1073/pnas.0404762101 (2004).
- 39 Pan, J. & Carroll, K. S. Persulfide reactivity in the detection of protein s-sulfhydration. *ACS Chem Biol* **8**, 1110-1116, doi:10.1021/cb4001052 (2013).
- 40 Ollagnier-de-Choudens, S. *et al.* Mechanistic studies of the SufS-SufE cysteine desulfurase: evidence for sulfur transfer from SufS to SufE. *FEBS Lett* **555**, 263-267, doi:10.1016/s0014-5793(03)01244-4 (2003).
- 41 Matthies, A., Nimtz, M. & Leimkuhler, S. Molybdenum cofactor biosynthesis in humans: identification of a persulfide group in the rhodanese-like domain of MOCS3 by mass spectrometry. *Biochemistry* **44**, 7912-7920, doi:10.1021/bi0503448 (2005).
- 42 Baez, N. O., Reisz, J. A. & Furdui, C. M. Mass spectrometry in studies of protein thiol chemistry and signaling: opportunities and caveats. *Free Radic Biol Med* **80**, 191-211, doi:10.1016/j.freeradbiomed.2014.09.016 (2015).
- 43 Singh, S. & Banerjee, R. PLP-dependent H₂S biogenesis. *Biochim Biophys Acta* **1814**, 1518-1527, doi:10.1016/j.bbapap.2011.02.004 (2011).
- 44 Zivanovic, J. *et al.* Selective Persulfide Detection Reveals Evolutionarily Conserved Antiaging Effects of S-Sulfhydration. *Cell Metab* **31**, 207, doi:10.1016/j.cmet.2019.12.001 (2020).
- 45 Wedmann, R. *et al.* Improved tag-switch method reveals that thioredoxin acts as depersulfidase and controls the intracellular levels of protein persulfidation. *Chem Sci* **7**, 3414-3426, doi:10.1039/c5sc04818d (2016).

- 46 Wu, N., Du, X., Wang, D. & Hao, F. Myocardial and lung injuries induced by hydrogen sulfide and the effectiveness of oxygen therapy in rats. *Clin Toxicol (Phila)* 49, 161-166, doi:10.3109/15563650.2011.565419 (2011).
- 47 Szabo, C. et al. Regulation of Mitochondrial Bioenergetic Function by Hydrogen Sulfide. Part I. Biochemical and Physiological Mechanisms. *Br J Pharmacol*, doi:10.1111/bph.12369 (2013).
- 48 Hendrickson, R. G., Chang, A. & Hamilton, R. J. Co-worker fatalities from hydrogen sulfide. *Am J Ind Med* 45, 346-350, doi:10.1002/ajim.10355 (2004).
- 49 Hine, C. et al. Endogenous hydrogen sulfide production is essential for dietary restriction benefits. *Cell* 160, 132-144, doi:10.1016/j.cell.2014.11.048 (2015).
- 50 Lemstra, M., Bird, Y., Nwankwo, C., Rogers, M. & Moraros, J. Weight loss intervention adherence and factors promoting adherence: a meta-analysis. *Patient Prefer Adherence* 10, 1547-1559, doi:10.2147/PPA.S103649 (2016).
- 51 Stewart, T. M. et al. Comprehensive Assessment of Long-term Effects of Reducing Intake of Energy Phase 2 (CALERIE Phase 2) screening and recruitment: methods and results. *Contemp Clin Trials* 34, 10-20, doi:10.1016/j.cct.2012.08.011 (2013).
- 52 Walford, R. L., Mock, D., Verdery, R. & MacCallum, T. Calorie restriction in biosphere 2: alterations in physiologic, hematologic, hormonal, and biochemical parameters in humans restricted for a 2-year period. *J Gerontol A Biol Sci Med Sci* 57, B211-224, doi:10.1093/gerona/57.6.b211 (2002).
- 53 Racette, S. B. et al. One year of caloric restriction in humans: feasibility and effects on body composition and abdominal adipose tissue. *J Gerontol A Biol Sci Med Sci* 61, 943-950, doi:10.1093/gerona/61.9.943 (2006).
- 54 Heilbronn, L. K. et al. Effect of 6-month calorie restriction on biomarkers of longevity, metabolic adaptation, and oxidative stress in overweight individuals: a randomized controlled trial. *JAMA* 295, 1539-1548, doi:10.1001/jama.295.13.1539 (2006).
- 55 Das, S. K. et al. Body-composition changes in the Comprehensive Assessment of Long-term Effects of Reducing Intake of Energy (CALERIE)-2 study: a 2-y randomized controlled trial of calorie restriction in nonobese humans. *Am J Clin Nutr* 105, 913-927, doi:10.3945/ajcn.116.137232 (2017).
- 56 Das, S. K. et al. Long-term effects of 2 energy-restricted diets differing in glycemic load on dietary adherence, body composition, and metabolism in CALERIE: a 1-y randomized controlled trial. *Am J Clin Nutr* 85, 1023-1030, doi:10.1093/ajcn/85.4.1023 (2007).
- 57 Headland, M., Clifton, P. M., Carter, S. & Keogh, J. B. Weight-Loss Outcomes: A Systematic Review and Meta-Analysis of Intermittent Energy Restriction Trials Lasting a Minimum of 6 Months. *Nutrients* 8, doi:10.3390/nu8060354 (2016).
- 58 Cioffi, I. et al. Intermittent versus continuous energy restriction on weight loss and cardiometabolic outcomes: a systematic review and meta-analysis of randomized controlled trials. *J Transl Med* 16, 371, doi:10.1186/s12967-018-1748-4 (2018).
- 59 Rynders, C. A. et al. Effectiveness of Intermittent Fasting and Time-Restricted Feeding Compared to Continuous Energy Restriction for Weight Loss. *Nutrients* 11, doi:10.3390/nu11102442 (2019).
- 60 Varady, K. A. & Hellerstein, M. K. Alternate-day fasting and chronic disease prevention: a review of human and animal trials. *Am J Clin Nutr* 86, 7-13, doi:10.1093/ajcn/86.1.7 (2007).
- 61 Catenacci, V. A. et al. A randomized pilot study comparing zero-calorie alternate-day fasting to daily caloric restriction in adults with obesity. *Obesity (Silver Spring)* 24, 1874-1883, doi:10.1002/oby.21581 (2016).
- 62 Talan, M. I. & Ingram, D. K. Effect of intermittent feeding on thermoregulatory abilities of young and aged C57BL/6J mice. *Arch Gerontol Geriatr* 4, 251-259, doi:10.1016/0167-4943(85)90007-x (1985).
- 63 Xie, K. et al. Every-other-day feeding extends lifespan but fails to delay many symptoms of aging in mice. *Nat Commun* 8, 155, doi:10.1038/s41467-017-00178-3 (2017).
- 64 Goodrick, C. L., Ingram, D. K., Reynolds, M. A., Freeman, J. R. & Cider, N. Effects of intermittent feeding upon body weight and lifespan in inbred mice: interaction of genotype and age. *Mech Ageing Dev* 55, 69-87, doi:10.1016/0047-6374(90)90107-q (1990).
- 65 Goodrick, C. L., Ingram, D. K., Reynolds, M. A., Freeman, J. R. & Cider, N. L. Differential effects of intermittent feeding and voluntary exercise on body weight and lifespan in adult rats. *J Gerontol* 38, 36-45, doi:10.1093/geronj/38.1.36 (1983).
- 66 Goodrick, C. L., Ingram, D. K., Reynolds, M. A., Freeman, J. R. & Cider, N. L. Effects of intermittent feeding upon growth and life span in rats. *Gerontology* 28, 233-241, doi:10.1159/000212538 (1982).

- 67 Acosta-Rodriguez, V. A., de Groot, M. H. M., Rijo-Ferreira, F., Green, C. B. & Takahashi, J. S. Mice under Caloric Restriction Self-Impose a Temporal Restriction of Food Intake as Revealed by an Automated Feeder System. *Cell Metab* 26, 267-277 e262, doi:10.1016/j.cmet.2017.06.007 (2017).
- 68 Froy, O., Chapnik, N. & Miskin, R. Effect of intermittent fasting on circadian rhythms in mice depends on feeding time. *Mech Ageing Dev* 130, 154-160, doi:10.1016/j.mad.2008.10.006 (2009).

REVIEWERS' COMMENTS

Reviewer #1 (Remarks to the Author):

The manuscript by Hine and colleagues has been much improved upon revision. I believe the study will be a valuable resource for the H2S field. Most of my queries have been addressed satisfactorily by the authors and some arguments made by the authors have also been taken. Therefore, in the aspect of redox proteomics, I am pleased to recommend publication in Nat Commun upon addressing the following concern and suggestion.

1. The literatures used to argue the validity of using SC approach were 'old' and, apparently, would not be acknowledged by the current proteomic community. Since the authors are able to re-analyze the RAW data with MS1-based approach using MaxQuant as described in the revised METHODS (I do appreciate the hard work by the authors), I initially see NO excuse not to incorporate the new analyses in the manuscript. However, I do realize that by doing so all bioinformatics and most figures need to be re-done or re-made, which are absolutely not trivial works. I also understand that such revisions may not change the major conclusions of this study. In this regard, it is not clear whether the differentially regulated SSH proteins identified by these two approaches are mostly the same, although side-by-side comparisons have been provided in the rebuttal letter (p7-9).

2. The term "sulfhydrome" should also be changed to "persulfidome"

Reviewer #2 (Remarks to the Author):

The authors have addressed the concerns and revisions adequately. I have no further comments.

Reviewer #3 (Remarks to the Author):

The authors adequately addressed the concerns raised in a thorough manner.

Response to Reviewers' Comments NCOMMS-20-07544B

Dear Reviewers:

This document presents our full response to the Reviewers' Comments for the manuscript titled "**Dietary restriction transforms the mammalian protein persulfidome in a tissue-specific and cystathionine γ -lyase-dependent manner**" (NCOMMS-20-07544B) by Nazmin Bithi, Christopher Link, Yoko O. Henderson, Suzie Kim, Jie Yang, Ling Li, Rui Wang, Belinda Willard, and Christopher Hine. We highly appreciate the feedback and constructive criticism from all three Reviewers, and we are very thankful to the Reviewers for carefully reading our manuscript. In this new revised manuscript, we have addressed to the best of our ability the biological, technical, and/or semantic issues raised by each Reviewer in the manuscript and associated files.

(A) Response to Reviewer #1

2nd Revision

A.1 Reviewer #1: The manuscript by Hine and colleagues has been much improved upon revision. I believe the study will be a valuable resource for the H2S field. Most of my queries have been addressed satisfactorily by the authors and some arguments made by the authors have also been taken. Therefore, in the aspect of redox proteomics, I am pleased to recommend publication in Nat Commun upon addressing the following concern and suggestion

Our response: We thank Reviewer #1 for providing constructive criticism and comments throughout the review process and helping us improve our manuscript.

A.2 Reviewer #1: The literatures used to argue the validity of using SC approach were 'old' and, apparently, would not be acknowledged by the current proteomic community. Since the authors are able to re-analyze the RAW data with MS1-based approach using MaxQuant as described in the revised METHODS (I do appreciate the hard work by the authors), I initially see NO excuse not to incorporate the new analyses in the manuscript. However, I do realize that by doing so all bioinformatics and most figures need to be re-done or re-made, which are absolutely not trivial works. I also understand that such revisions may not change the major conclusions of this study. In this regard, it is not clear whether the differentially regulated SSH proteins identified by these two approaches are mostly the same, although side-by-side comparisons have been provided in the rebuttal letter (p7-9).

Our response: We appreciate Reviewer 1's comments and suggestions regarding the MS1-based analysis approaches and their importance to be incorporated into the current study. Thus, we have now supplied all MS1-based analysis in parallel with spectral counting analysis in which spectral counting was originally used. This new presentation of MS1-based analysis is included in all Supplementary Data files as the second tab in the Excel sheets (with the first tab containing the respective spectral counting data), and includes numerical data as well as the corresponding graphs/plots (volcano plots, Venn diagrams, etc) so they can be easily understood and compared to the spectral count data, which now also has the corresponding graphs/plots placed alongside the numerical data in the Supplementary Data files. By doing so, we believe this has two benefits: 1) Readers can now easily access data from both methods (spectral count based and MS-1 intensity based) and see how their numerical data fits into the visual data, and 2) Comparisons between the two methods can be made with similar outcomes observed, emphasizing the reported results in the narrative as robust and rigorously obtained.

In addition to now including all MS-1 data in the Supplemental Data files, we have also included a new figure in the main manuscript (Figure 8I) that shows the mean \pm SEM plotted for % persulfidated protein enrichment under DR compared to AL feeding from each solid tissue in CGL WT and KO mice as determined through spectral counting and MS1 intensity. Importantly, this new figure shows similar results were obtained via MS1 intensity analysis compared to spectral counting analysis of persulfidated proteins from the five solid tissues. Thus, both label-free quantification mass spectrometry analysis techniques utilized ultimately produced similar results and highlighted the tissue specificity and dependence on CGL for transforming persulfidomes under DR.

We have also now included a statement introducing and outlining our use of spectral counting and MS1 intensity in the results section for Figure 2, which directs the readers on how to access either data sets in the Supplemental Data files and to see the side-by-side comparisons in the Peer Review File.

A.3 Reviewer #1: The term “sulphydrome” should also be changed to “persulfidome”

Our response: We have now replaced the term “sulphydrome” both in the title and in the body of the manuscript with the term “persulfidome”.

(B) Response to Reviewer #2

2nd Revision

B.1 Reviewer #2: The authors have addressed the concerns and revisions adequately. I have no further comments.

Our response: We thank Reviewer #2 for their comments and for guiding us in improving our manuscript.

(C) Response to Reviewer #3

2nd Revision

C.1 Reviewer #3: The authors adequately addressed the concerns raised in a thorough manner.

Our response: We thank Reviewer #3 for their comments and for guiding us in improving our manuscript.